# QUADRATIC MODELS FOR UNDERSTANDING CATAPULT DYNAMICS OF NEURAL NETWORKS

**Libin Zhu**[1,2], **Chaoyue Liu**[2], **Adityanarayanan Radhakrishnan**[3], **Mikhail Belkin**[1,2]

[1]Department of Computer Science, UC San Diego
[2]Halıcıoğlu Data Science Institute, UC San Diego
[3]Harvard & Broad Institute of MIT and Harvard
[2]{libinzhu,ch1212,mbelkin}@ucsd.edu
[3]aradha@mit.edu

## ABSTRACT

While neural networks can be approximated by linear models as their width increases, certain properties of wide neural networks cannot be captured by linear models. In this work we show that recently proposed Neural Quadratic Models can exhibit the "catapult phase" (Lewkowycz et al., 2020) that arises when training such models with large learning rates. We then empirically show that the behaviour of neural quadratic models parallels that of neural networks in generalization, especially in the catapult phase regime. Our analysis further demonstrates that quadratic models can be an effective tool for analysis of neural networks.

## 1 INTRODUCTION

A recent remarkable finding on neural networks, originating from Jacot et al. (2018) and termed as the "transition to linearity" (Liu et al., 2020), is that, as network width goes to infinity, such models become linear functions in the parameter space. Thus, a linear (in parameters) model can be built to accurately approximate wide neural networks under certain conditions. While this finding has helped improve our understanding of trained neural networks (Du et al., 2019; Nichani et al., 2021; Zou & Gu, 2019; Montanari & Zhong, 2020; Ji & Telgarsky, 2019; Chizat et al., 2019), not all properties of finite width neural networks can be understood in terms of linear models, as is shown in several recent works (Yang & Hu, 2020; Ortiz-Jiménez et al., 2021; Long, 2021; Fort et al., 2020). In this work, we show that properties of finitely wide neural networks in optimization and generalization that cannot be captured by linear models are, in fact, manifested in quadratic models.

The training dynamics of linear models with respect to the choice of the learning rates[1] are well-understood (Polyak, 1987). Indeed, such models exhibit *linear* training dynamics, i.e., there exists a critical learning rate, $\eta_{\text{crit}}$, such that the loss converges monotonically if and only if the learning rate is smaller than $\eta_{\text{crit}}$ (see Figure 1a).

Recent work Lee et al. (2019) showed that the training dynamics of a wide neural network $f(\mathbf{w}; \boldsymbol{x})$ can be accurately approximated by that of a linear model $f_{\text{lin}}(\mathbf{w}; \boldsymbol{x})$:

$$f_{\text{lin}}(\mathbf{w}; \boldsymbol{x}) = f(\mathbf{w}_0; \boldsymbol{x}) + (\mathbf{w} - \mathbf{w}_0)^T \nabla f(\mathbf{w}_0; \boldsymbol{x}), \tag{1}$$

where $\nabla f(\mathbf{w}_0; \boldsymbol{x})$ denotes the gradient[2] of $f$ with respect to trainable parameters $\mathbf{w}$ at an initial point $\mathbf{w}_0$ and input sample $\boldsymbol{x}$. This approximation holds for learning rates less than $\eta_{\text{crit}} \approx 2/\|\nabla f(\mathbf{w}_0; \boldsymbol{x})\|^2$, when the width is sufficiently large.

However, the training dynamics of finite width neural networks, $f$, can sharply differ from those of linear models when using large learning rates. A striking non-linear property of wide neural

---

[1]Unless stated otherwise, we always consider the setting where models are trained with squared loss using gradient descent.

[2]For non-differentiable functions, e.g. neural networks with ReLU activation functions, we define the gradient based on the update rule used in practice. Similarly, we use $H_f$ to denote the second derivative of $f$ in Eq. (2).

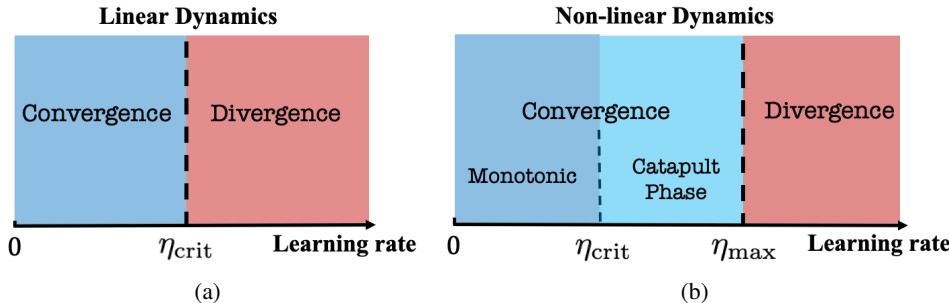

Figure 1: **Optimization dynamics for linear and non-linear models based on choice of learning rate.** (**a**) Linear models either converge monotonically if learning rate is less than $\eta_{\text{crit}}$ and diverge otherwise. (**b**) Unlike linear models, *finitely wide neural networks* and *NQMs Eq. (2) (or general quadratic models Eq. (3))* can additionally observe a catapult phase when $\eta_{\text{crit}} < \eta < \eta_{\text{max}}$.

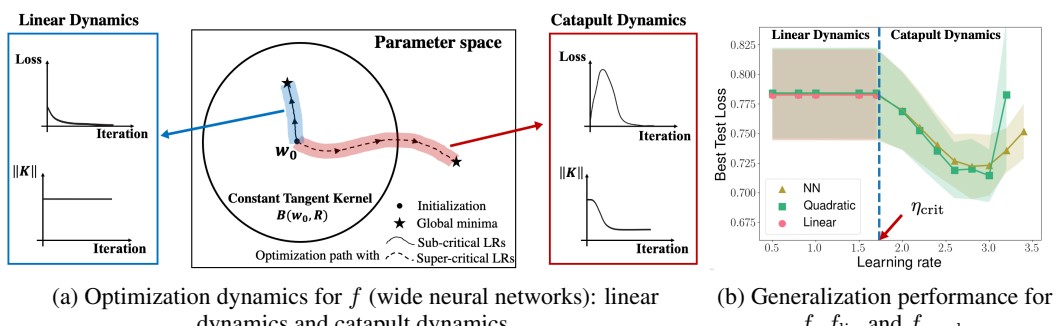

(a) Optimization dynamics for $f$ (wide neural networks): linear dynamics and catapult dynamics.

(b) Generalization performance for $f$, $f_{\text{lin}}$ and $f_{\text{quad}}$.

Figure 2: (**a**) **Optimization dynamics of wide neural networks with sub-critical and super-critical learning rates.** With sub-critical learning rates ($0 < \eta < \eta_{\text{crit}}$), the tangent kernel of wide neural networks is nearly constant during training, and the loss decreases monotonically. The whole optimization path is contained in the ball $B(\mathbf{w}_0, R) := \{\mathbf{w} : \|\mathbf{w} - \mathbf{w}_0\| \leq R\}$ with a finite radius $R$. With super-critical learning rates ($\eta_{\text{crit}} < \eta < \eta_{\text{max}}$), the catapult phase happens: the loss first increases and then decreases, along with a decrease of the norm of the tangent kernel . The optimization path goes beyond the finite radius ball. (**b**) **Test loss of $f_{\text{quad}}$, $f$ and $f_{\text{lin}}$ plotted against different learning rates.** With sub-critical learning rates, all three models have nearly identical test loss for any sub-critical learning rate. With super-critical learning rates, $f$ and $f_{\text{quad}}$ have smaller best test loss than the one with sub-critical learning rates. Experimental details are in Appendix N.5.

networks discovered in Lewkowycz et al. (2020) is that when the learning rate is larger than $\eta_{\text{crit}}$ but smaller than a certain maximum learning rate, $\eta_{\text{max}}$, gradient descent still converges but experiences a "catapult phase." Specifically, the loss initially grows exponentially and then decreases after reaching a large value, along with the decrease of the norm of tangent kernel (see Figure 2a), and therefore, such training dynamics are *non-linear* (see Figure 1b).

As linear models cannot exhibit such a catapult phase, under what models and conditions does this phenomenon arise? The work of Lewkowycz et al. (2020) first observed the catapult phase phenomenon in finite width neural networks and analyzed this phenomenon for a two-layer linear neural network. However, a theoretical understanding of this phenomenon for general non-linear neural networks remains open. In this work, we utilize a quadratic model as a tool to shed light on the optimization and generalization discrepancies between finite and infinite width neural networks. We define *Neural Quadratic Model (NQM)* by the second order Taylor series expansion of $f(\mathbf{w}; \boldsymbol{x})$ around the point $\mathbf{w}_0$:

$$\textbf{NQM}: \quad f_{\text{quad}}(\mathbf{w}) = f(\mathbf{w}_0) + (\mathbf{w} - \mathbf{w}_0)^T \nabla f(\mathbf{w}_0) + \frac{1}{2}(\mathbf{w} - \mathbf{w}_0)^T H_f(\mathbf{w}_0)(\mathbf{w} - \mathbf{w}_0). \quad (2)$$

Here in the notation we suppress the dependence on the input data $\boldsymbol{x}$, and $H_f(\mathbf{w}_0)$ is the Hessian of $f$ with respect to $\mathbf{w}$ evaluated at $\mathbf{w}_0$. Note that $f_{\text{quad}}(\mathbf{w}) = f_{\text{lin}}(\mathbf{w}) + \frac{1}{2}(\mathbf{w} - \mathbf{w}_0)^T H_f(\mathbf{w}_0)(\mathbf{w} - \mathbf{w}_0)$.

Indeed, we note that NQMs are contained in a more general class of quadratic models:

**General Quadratic Model :** $\qquad g(\mathbf{w}; \boldsymbol{x}) = \mathbf{w}^T \phi(\boldsymbol{x}) + \frac{1}{2}\gamma\mathbf{w}^T\Sigma(\boldsymbol{x})\mathbf{w}, \qquad\qquad (3)$

where $\mathbf{w}$ are trainable parameters and $\boldsymbol{x}$ is input data. We discuss the optimization dynamics of such general quadratic models in Section 3.3 and show empirically that they exhibit the catapult phase phenomenon in Appendix N.4. Note that the two-layer linear network analyzed in Lewkowycz et al. (2020) is a special case of Eq. (3), when $\phi(\boldsymbol{x}) = 0$ (See Appendix M).

**Main Contributions.** We prove that NQMs, $f_{\text{quad}}$, which approximate shallow fully-connected ReLU activated neural networks, exhibit catapult phase dynamics. Specifically, we analyze the optimization dynamics of $f_{\text{quad}}$ by deriving the evolution of $f_{\text{quad}}$ and the tangent kernel during gradient descent with squared loss, for a single training example and multiple uni-dimensional training examples. We identify three learning rate regimes yielding different optimization dynamics for $f_{\text{quad}}$, which are (1) converging monotonically (linear dynamics); (2) converging via a catapult phase (catapult dynamics); and (3) diverging. We provide a number of experimental results corroborating our theoretical analysis (See Section 3).

We then empirically show that NQMs, for the architectures of shallow (see Figure 2b as an example) and deep networks, have better test performances when catapult dynamics happens. While this was observed for some synthetic examples of neural networks in Lewkowycz et al. (2020), we systematically demonstrate the improved generalization of NQMs across a range of experimental settings. Namely, we consider fully-connected and convolutional neural networks with ReLU and other activation functions trained with GD/SGD on multiple vision, speech and text datatsets (See Section 4).

To the best of our knowledge, our work is the first to analyze the non-linear wide neural networks in the catapult regime through the perspective of the quadratic approximation. While NQMs (or quadratic models) were proposed and analyzed in Roberts et al. (2022), our work focuses on the properties of NQMs in the large learning rate regime, which has not been discussed in Roberts et al. (2022). Similarly, the following related works did not study catapult dynamics. Huang & Yau (2020) analyzed higher order approximations to neural networks under gradient flow (infinitesimal learning rates). Bai & Lee (2019) studied different quadratic models with randomized second order terms and Zhang et al. (2019) considered the loss in the quadratic form, where no catapult phase happens. A recent work showed the existence of the catapult phase in two-layer, homogenous networks Meltzer & Liu (2023).

**Discontinuity in dynamics transition.** In the ball $B(\mathbf{w}_0, R) := \{\mathbf{w} : \|\mathbf{w} - \mathbf{w}_0\| \leq R\}$ with constant radius $R > 0$, the transition to linearity of a wide neural network (with linear output layer) is continuous in the network width $m$. That is, the deviation from the network function to its linear approximation within the ball can be continuously controlled by the Hessian of the network function, i.e. $H_f$, which scales with $m$ (Liu et al., 2020):

$$\|f(\mathbf{w}) - f_{\text{lin}}(\mathbf{w})\| \leq \sup_{\mathbf{w} \in B(\mathbf{w}_0, R)} \|H_f(\mathbf{w})\|R^2 = \tilde{O}(1/\sqrt{m}). \qquad (4)$$

Using the inequality from Eq. (4), we obtain $\|f_{\text{quad}} - f_{\text{lin}}\| = \tilde{O}(1/\sqrt{m})$, hence $f_{\text{quad}}$ transitions to linearity continuously as well in $B(\mathbf{w}_0, R)^3$. Given the continuous nature of the transition to linearity, one may expect that the transition from non-linear dynamics to linear dynamics for $f$ and $f_{\text{quad}}$ is continuous in $m$ as well. Namely, one would expect that the domain of catapult dynamics, $[\eta_{\text{crit}}, \eta_{\text{max}}]$, shrinks and ultimately converges to a single point, i.e., $\eta_{\text{crit}} = \eta_{\text{max}}$, as $m$ goes to infinity, with non-linear dynamics turning into linear dynamics. However, as shown both analytically and empirically, the transition is *not* continuous, for both network functions $f$ and NQMs $f_{\text{quad}}$, since the domain of the catapult dynamics can be independent of the width $m$ (or $\gamma$). Additionally, the length of the optimization path of $f$ in catapult dynamics grows with $m$ since otherwise, the optimization path could be contained in a ball with a constant radius independent of $m$, in which $f$ can be approximated by $f_{\text{lin}}$. Since the optimization of $f_{\text{lin}}$ diverges in catapult dynamics, by the approximation, the optimization of $f$ diverges as well, which contradicts the fact that the optimization of $f$ can converge in catapult dynamics (See Figure 2a).

---

[3]For general quadratic models in Eq. (3), the transition to linearity is continuously controlled by $\gamma$.

## 2 NOTATION AND PRELIMINARY

We use bold lowercase letters to denote vectors and capital letters to denote matrices. We denote the set $\{1, 2, \cdots, n\}$ by $[n]$. We use $\|\cdot\|$ to denote the Euclidean norm for vectors and the spectral norm for matrices. We use $\odot$ to denote element-wise multiplication (Hadamard product) for vectors. We use $\lambda_{\max}(A)$ and $\lambda_{\min}(A)$ to denote the largest and smallest eigenvalue of a matrix $A$, respectively.

Given a model $f(\mathbf{w}; \boldsymbol{x})$, where $\boldsymbol{x}$ is input data and $\mathbf{w}$ are model parameters, we use $\nabla_{\mathbf{w}} f$ to represent the partial first derivative $\partial f(\mathbf{w}; \boldsymbol{x})/\partial \mathbf{w}$. When clear from context, we let $\nabla f := \nabla_{\mathbf{w}} f$ for ease of notation. We use $H_f$ and $H_{\mathcal{L}}$ to denote the Hessian (second derivative matrix) of the function $f(\mathbf{w}; \boldsymbol{x})$ and the loss $\mathcal{L}(\mathbf{w})$ with respect to parameters $\mathbf{w}$, respectively.

In the paper, we consider the following supervised learning task: given training data $\{(\boldsymbol{x}_i, y_i)\}_{i=1}^n$ with data $\boldsymbol{x}_i \in \mathbb{R}^d$ and labels $y_i \in \mathbb{R}$ for $i \in [n]$, we minimize the empirical risk with the squared loss $\mathcal{L}(\mathbf{w}) = \frac{1}{2} \sum_{i=1}^n (f(\mathbf{w}; \boldsymbol{x}_i) - y_i)^2$. Here $f(\mathbf{w}; \cdot)$ is a parametric family of models, e.g., a neural network or a kernel machine, with parameters $\mathbf{w} \in \mathbb{R}^p$. We use full-batch gradient descent to minimize the loss, and we denote trainable parameters $\mathbf{w}$ at iteration $t$ by $\mathbf{w}(t)$. With constant step size (learning rate) $\eta$, the update rule for the parameters is:

$$\mathbf{w}(t+1) = \mathbf{w}(t) - \eta \frac{d\mathcal{L}(\mathbf{w})}{d\mathbf{w}}(t), \ \ \forall t \geq 0.$$

**Definition 1** (Tangent Kernel). *The tangent kernel $K(\mathbf{w}; \cdot, \cdot)$ of $f(\mathbf{w}; \cdot)$ is defined as*

$$K(\mathbf{w}; \boldsymbol{x}, \boldsymbol{z}) = \langle \nabla f(\mathbf{w}; \boldsymbol{x}), \nabla f(\mathbf{w}; \boldsymbol{z}) \rangle, \quad \forall \boldsymbol{x}, \boldsymbol{z} \in \mathbb{R}^d. \tag{5}$$

In the context of the optimization problem with $n$ training examples, the tangent kernel matrix $K \in \mathbb{R}^{n \times n}$ satisfies $K_{i,j}(\mathbf{w}) = K(\mathbf{w}; \boldsymbol{x}_i, \boldsymbol{x}_j)$, $i, j \in [n]$. The critical learning rate for optimization is given as follows.

**Definition 2** (Critical learning rate). *With an initialization of parameters $\mathbf{w}_0$, the critical learning rate of $f(\mathbf{w}; \cdot)$ is defined as*

$$\eta_{\mathrm{crit}} := 2/\lambda_{\max}(H_{\mathcal{L}}(\mathbf{w}_0)). \tag{6}$$

*A learning rate $\eta$ is said to be sub-critical if $0 < \eta < \eta_{\mathrm{crit}}$ or super-critical if $\eta_{\mathrm{crit}} < \eta < \eta_{\max}$. Here $\eta_{\max}$ is the maximum leaning rate such that the optimization of $\mathcal{L}(\mathbf{w})$ initialized at $\mathbf{w}_0$ can converge.*

**Dynamics for Linear models.** When $f$ is linear in $\mathbf{w}$, the gradient, $\nabla f$, and tangent kernel are constant: $K(\mathbf{w}(t)) = K(\mathbf{w}_0)$. Therefore, gradient descent dynamics are:

$$F(\mathbf{w}(t+1)) - \mathbf{y} = (I - \eta K(\mathbf{w}_0))(F(\mathbf{w}(t)) - \mathbf{y}), \quad \forall t \geq 0, \tag{7}$$

where $F(\mathbf{w}_0) = [f_1(\mathbf{w}_0), ..., f_n(\mathbf{w}_0)]^T$ with $f_i(\mathbf{w}_0) = f(\mathbf{w}_0; \boldsymbol{x}_i)$.

Noting that $H_{\mathcal{L}}(\mathbf{w}_0) = \nabla F(\mathbf{w}_0)^T \nabla F(\mathbf{w}_0)$ and that tangent kernel $K(\mathbf{w}_0) = \nabla F(\mathbf{w}_0) \nabla F(\mathbf{w}_0)^T$ share the same positive eigenvalues, we have $\lambda_{\max}(H_{\mathcal{L}}(\mathbf{w}_0)) = \lambda_{\max}(K(\mathbf{w}_0))$, and hence,

$$\eta_{\mathrm{crit}} = 2/\lambda_{\max}(K(\mathbf{w}_0)). \tag{8}$$

Therefore, from Eq. (7), if $0 < \eta < \eta_{\mathrm{crit}}$, the loss $\mathcal{L}$ decreases monotonically and if $\eta > \eta_{\mathrm{crit}}$, the loss $\mathcal{L}$ keeps increasing. Note that the critical and maximum learning rates are equal in this setting.

## 3 OPTIMIZATION DYNAMICS IN NEURAL QUADRATIC MODELS

In this section, we analyze the gradient descent dynamics of the NQM corresponding to a two-layer fully-connected neural network. We show that, unlike a linear model, the NQM exhibits a catapult dynamics: the loss increases at the early stage of training then decreases afterwards. We further show that the top eigenvalues of the tangent kernel typically become smaller as a consequence of the catapult.

**Neural Quadratic Model (NQM).** Consider the NQM that approximates the following two-layer neural network:

$$f(\mathbf{u}, \mathbf{v}; \boldsymbol{x}) = \frac{1}{\sqrt{m}} \sum_{i=1}^m v_i \sigma\left(\frac{1}{\sqrt{d}} \mathbf{u}_i^T \boldsymbol{x}\right), \tag{9}$$

where $\mathbf{u}_i \in \mathbb{R}^d$, $v_i \in \mathbb{R}$ for $i \in [m]$ are trainable parameters, $\boldsymbol{x} \in \mathbb{R}^d$ is the input, and $\sigma(\cdot)$ is the ReLU activation function. We initialize $\mathbf{u}_i \sim \mathcal{N}(0, I_d)$ and $v_i \in \text{Unif}[\{-1, 1\}]$ for each $i$ independently. Letting $g(\mathbf{u}, \mathbf{v}; \boldsymbol{x}) := f_{\text{quad}}(\mathbf{u}, \mathbf{v}; \boldsymbol{x})$, this NQM has the following expression (See the full derivation in Appendix A):

$$g(\mathbf{u}, \mathbf{v}; \boldsymbol{x}) = f(\mathbf{u}_0, \mathbf{v}_0; \boldsymbol{x}) + \frac{1}{\sqrt{md}} \sum_{i=1}^{m} v_{0,i}(\mathbf{u}_i - \mathbf{u}_{0,i})^T \boldsymbol{x} \mathbb{1}_{\left\{\mathbf{u}_{0,i}^T \boldsymbol{x} \geq 0\right\}} + \frac{1}{\sqrt{md}} \sum_{i=1}^{m} (v_i - v_{0,i})\sigma\left(\mathbf{u}_{0,i}^T \boldsymbol{x}\right)$$

$$+ \frac{1}{\sqrt{md}} \sum_{i=1}^{m} (v_i - v_{0,i})(\mathbf{u}_i - \mathbf{u}_{0,i})^T \boldsymbol{x} \mathbb{1}_{\left\{\mathbf{u}_{0,i}^T \boldsymbol{x} \geq 0\right\}}. \tag{10}$$

Given training data $\{\boldsymbol{x}_i, y_i\}_{i=1}^n$, we minimize the empirical risk with the squared loss $\mathcal{L}(\mathbf{w}) = \frac{1}{2} \sum_{i=1}^{n} (g(\mathbf{w}; \boldsymbol{x}_i) - y_i)^2$ using GD with constant learning rate $\eta$. Throughout this section, we denote $g(\mathbf{u}(t), \mathbf{v}(t); \boldsymbol{x})$ by $g(t)$ and its tangent kernel $K(\mathbf{u}(t), \mathbf{v}(t))$ by $K(t)$, where $t$ is the iteration of GD. We assume $\|\boldsymbol{x}_i\| = O(1)$ and $|y_i| = O(1)$ for $i \in [n]$, and we assume the width of $f$ is much larger than the input dimension $d$ and the data size $n$, i.e., $m \gg \max\{d, n\}$. Hence, $d$ and $n$ can be regarded as small constants. In the whole paper, we use the big-O and small-o notation with respect to the width $m$. Below, we start with the single training example case, which already showcases the non-linear dynamics of NQMs.

## 3.1 CATAPULT DYNAMICS WITH A SINGLE TRAINING EXAMPLE

In this subsection, we consider training dynamics of NQM Eq. (10) with a single training example $(\boldsymbol{x}, y)$ where $\boldsymbol{x} \in \mathbb{R}^d$ and $y \in \mathbb{R}$. In this case, the tangent kernel matrix $K$ reduces to a scalar, and we denote $K$ by $\lambda$ to distinguish it from a matrix.

By gradient descent with step size $\eta$, the updates for $g(t) - y$ and $\lambda(t)$, which we refer to as dynamics equations, can be derived as follows (see the derivation in Appendix B.1):

**Dynamics equations.**

$$g(t+1) - y = \left(1 - \eta\lambda(t) + \underbrace{\frac{\|\boldsymbol{x}\|^2}{md}\eta^2(g(t) - y)g(t)}_{R_g(t)}\right)(g(t) - y) := \mu(t)(g(t) - y), \tag{11}$$

$$\lambda(t+1) = \lambda(t) - \underbrace{\eta\frac{\|\boldsymbol{x}\|^2}{md}(g(t) - y)^2\left(4\frac{g(t)}{g(t) - y} - \eta\lambda(t)\right)}_{R_\lambda(t)}, \quad \forall t \geq 0. \tag{12}$$

Note that as the loss is given by $\mathcal{L}(t) = \frac{1}{2}(g(t) - y)^2$, to understand convergence, it suffices to analyze the dynamics equations above. Compared to the linear dynamics Eq. (7), this non-linear dynamics has extra terms $R_g(t)$ and $R_\lambda(t)$, which are induced by the non-linear term in the NQM. We will see that the convergence of gradient descent depends on the scale and sign of $R_g(t)$ and $R_\lambda(t)$. For example, for constant learning rate that is slightly larger than $\eta_{\text{crit}}$ (which would result in divergence for linear models), $R_\lambda(t)$ stays positive during training, resulting in both monotonic decrease of tangent kernel $\lambda$ and the loss.

As $\lambda(t) = \lambda_0 - \sum_{\tau=0}^{t-1} R_\lambda(\tau)$, to track the scale of $|\mu(t)|$, we will focus on the scale and sign of $R_g(t)$ and $R_\lambda(t)$ in the following analysis. For the scale of $\lambda_0$, which is non-negative by Definition 1, we can show that with high probability over random initialization, $|\lambda_0| = \Theta(1)$ (see Appendix I). And $|g(0)| = O(1)$ with high probability as well Du et al. (2018). Therefore the following discussion is with high probability over random initialization. We start by establishing monotonic convergence for sub-critical learning rates.

**Monotonic convergence: sub-critical learning rates ($\eta < 2/\lambda_0 = \eta_{\text{crit}}$).** The key observation is that when $|g(t)| = O(1)$, and $\lambda(t) = \Theta(1)$, $|R_g(t)|$ and $|R_\lambda(t)|$ are of the order $o(1)$. Then, the dynamics equations approximately reduce to the ones of linear dynamics:

$$g(t+1) - y = (1 - \eta\lambda(t) + o(1))(g(t) - y),$$
$$\lambda(t+1) = \lambda(t) + o(1).$$

Note that at initialization, the output satisfies $|g(0)| = O(1)$, and we have shown $\lambda_0 = \Theta(1)$. With the choice of $\eta$, we have for all $t \geq 0$, $|\mu(t)| = |1 - \eta\lambda(t) + o(1)| < 1$; hence, $|g(t) - y|$ decreases monotonically. The cumulative change on the tangent kernel will be $o(1)$, i.e., $\sum_t |R_\lambda(t)| = o(1)$, since for all $t$, $|R_\lambda(t)| = O(1/m)$ and the loss decreases exponentially hence $\sum |R_\lambda(t)| = O(1/m) \cdot \log O(1) = o(1)$. See Appendix C for a detailed discussion.

**Catapult convergence: super-critical learning rates ($\eta_{\mathrm{crit}} = 2/\lambda_0 < \eta < 4/\lambda_0 = \eta_{\max}$).** The training dynamics are given by the following theorem.

**Theorem 1** (Catapult dynamics on a single training example). *Consider training the NQM Eq. (10) with squared loss on a single training example by GD. With a super-critical learning rate $\eta \in \left[\frac{2+\epsilon}{\lambda_0}, \frac{4-\epsilon}{\lambda_0}\right]$ where $\epsilon = \Theta\left(\frac{\log m}{\sqrt{m}}\right)$, the catapult happens: with high probability over random initialization, the loss increases to the order of $\Omega\left(\frac{m(\eta\lambda_0 - 2)^2}{\log m}\right)$ then decreases to $O(1)$.*

*Proof of Theorem 1.* We use the following transformation of the variables to simplify notations.

$$u(t) = \frac{\|\boldsymbol{x}\|^2}{md}\eta^2(g(t) - y)^2, \quad w(t) = \frac{\|\boldsymbol{x}\|^2}{md}\eta^2(g(t) - y)y, \quad v(t) = \eta\lambda(t).$$

Then the Eq. (11) and Eq. (12) are reduced to

$$u(t+1) = (1 - v(t) + u(t) + w(t))^2 u(t) := \kappa(t)u(t), \tag{13}$$
$$v(t+1) = v(t) - u(t)(4 - v(t)) - 4w(t). \tag{14}$$

At initialization, since $|g(0)| = O(1)$, we have $u(0) = O\left(\frac{1}{m}\right)$ and $w(0) = O\left(\frac{1}{m}\right)$. Note that by definition, for all $t \geq 0$, $u(t) \geq 0$ and we have $v(t) \geq 0$ since $\lambda(t)$ is the tangent kernel for a single training example.

In the following, we will analyze the above dynamical equations. To make the analysis more understandable, we separate the dynamics during training into increasing phase and decreasing phase. We denote $\delta := (\eta - \eta_{\mathrm{crit}})\lambda_0 = \eta\lambda_0 - 2$.

**Increasing phase.** In this phase, $|u(t)|$ increases exponentially from $O\left(\frac{1}{m}\right)$ to $\Theta\left(\frac{\delta^2}{\log m}\right)$ and $|v(t) - v(0)| = O\left(\frac{\delta}{\log m}\right)$. This can be shown by the following lemma.

**Lemma 1.** *For $T > 0$ such that $\sup_{t\in[0,T]} u(t) = O\left(\frac{\delta^2}{\log m}\right)$, $u(t)$ increases exponentially with $\inf_{t\in[0,T]} \kappa(t) \geq \left(1 + \delta - O\left(\frac{\delta}{\log m}\right)\right)^2 > 1$ and $\sup_{t\in[0,T]} |v(t) - v(0)| = O\left(\frac{\delta}{\log m}\right)$.*

*Proof.* See the proof in Section D. $\square$

After the increasing phase, based on the order of $u(t)$ we can infer the order of loss is $\Theta\left(\frac{m\delta^2}{\log m}\right)$.

**Decreasing phase.** When $u(t)$ is sufficiently large, $v(t)$ will have non-negligible decrease which leads to the decreasing of $\kappa(t)$, hence in turn making $u(t)$ decrease as well. Consequently, we have:

**Lemma 2.** *There exists $T^* > 0$ such that $u(T^*) = O\left(\frac{1}{m}\right)$.*

*Proof.* See the proof in Section E. $\square$

Then accordingly, the loss is of the order $O(1)$. $\square$

Once the loss decreases to the order of $O(1)$, the catapult finishes and we in general have $\eta < 2/\lambda(t)$ as $|\mu(t)| = |1 - \eta\lambda(t) + R_{\boldsymbol{g}}(t)| < 1$ where $|R_{\boldsymbol{g}}(t)| = O(\mathcal{L}(t)/m) = O(1/m)$. Therefore the training dynamics fall into linear dynamics, and we can use the same analysis for sub-critical learning rates for the remaining training dynamics. The stableness of the steady-state equilibria of dynamical equations can be guaranteed by the following:

**Theorem 2.** *For dynamical equations Eq. (11) and (12), the stable steady-state equilibria satisfy $g(t) = y$ (i.e.,loss is 0), and $\lambda(t) \in [\epsilon, 2/\eta - \epsilon]$ with $\epsilon = \Theta(\log m/\sqrt{m})$.*

**Divergence** ($\eta > \eta_{\max} = 4/\lambda_0$). Initially, it follows the same dynamics with that in the increasing phase in catapult convergence: $|g(t) - y|$ increases exponentially as $|\mu(t)| > 1$ and the $\lambda(t)$ almost does not change as $R_\lambda(t)$ is small. However, note that $R_\lambda(t) > 0$ since 1) $g(t)/(g(t)-y) \approx 1$ when $g(t)$ becomes large and 2) $\eta > 4/\lambda(t)$. Therefore, $\lambda(t)$ keeps increasing during training, which consequently leads to the divergence of the optimization. See Appendix G for a detailed discussion.

### 3.2 CATAPULT DYNAMICS WITH MULTIPLE TRAINING EXAMPLES

In this subsection we show the catapult phase will happen for NQMs Eq. (9) with multiple training examples. We assume unidimensional input data, which is common in the literature and simplifies the analysis for neural networks (see for example Williams et al. (2019); Savarese et al. (2019)).

**Assumption 1.** *The input dimension $d = 1$ and not all $x_i$ is 0, i.e., $\sum |x_i| > 0$.*

We similarly analyze the dynamics equations with different learning rates for multiple training examples (see the derivation of Eq. (22) and (23) in Appendix) which are update equations of $\mathbf{g}(t) - \mathbf{y}$ and $K(t)$. And similarly, we show there are three training dynamics: monotonic convergence, catapult convergence and divergence.

In the analysis, we consider the training dynamics projected to two orthogonal eigenvectors of the tangent kernel, i.e., $\boldsymbol{p}_1$ and $\boldsymbol{p}_2$, and we show with different learning rates, the catapult phase can occur only in the direction of $\boldsymbol{p}_1$, or occur in both directions. We consider the case where $2/\lambda_2(0) < 4/\lambda_1(0)$ hence the catapult can occur in both directions. The analysis for the other case can be directly obtained from our results. We denote the loss projected to $\boldsymbol{p}_i$ by $\Pi_i \mathcal{L} := \frac{1}{2} \langle \mathbf{g} - \mathbf{y}, \boldsymbol{p}_i \rangle^2$ for $i = 1, 2$. We have $\Pi_i \mathcal{L}(0) = O(1)$ with high probability over random initialization of weights.

We formulate the result for the catapult dynamics, which happens when training with super-critical learning rates, into the following theorem, and defer the proof of it and the full discussion of training dynamics to Appendix K.

**Theorem 3** (Catapult dynamics on multiple training examples). *Supposing Assumption 1 holds, consider training the NQM Eq. (10) with squared loss on multiple training examples by GD. Then, with high probability over random initialization we have*

1. *with $\eta \in \left[\frac{2+\epsilon}{\lambda_1(0)}, \frac{2-\epsilon}{\lambda_2(0)}\right]$, the catapult only occurs in eigendirection $\boldsymbol{p}_1$: $\Pi_1 \mathcal{L}$ increases to the order of $\Omega\left(\frac{m(\eta\lambda_1(0)-2)^2}{\log m}\right)$ then decreases to $O(1)$;*

2. *with $\eta \in \left[\frac{2+\epsilon}{\lambda_2(0)}, \frac{4-\epsilon}{\lambda_1(0)}\right]$, the catapult occurs in both eigendirections $\boldsymbol{p}_1$ and $\boldsymbol{p}_2$: $\Pi_i \mathcal{L}$ for $i = 1, 2$ increases to the order of $\Omega\left(\frac{m(\eta\lambda_i(0)-2)^2}{\log m}\right)$ then decreases to $O(1)$,*

*where $\epsilon = \Theta\left(\frac{\log m}{\sqrt{m}}\right)$.*

We verify the results for multiple training examples via the experiments in Figure 3.

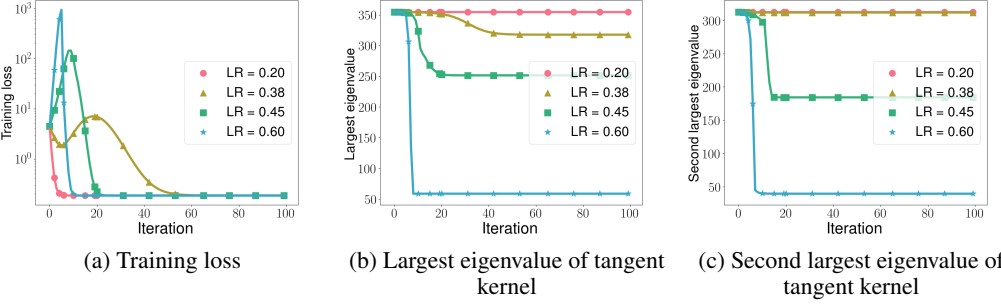

(a) Training loss     (b) Largest eigenvalue of tangent kernel     (c) Second largest eigenvalue of tangent kernel

Figure 3: **Training dynamics of NQMs for multiple examples case with different learning rates.** By our analysis, two critical values are $2/\lambda_1(0) = 0.37$ and $2/\lambda_2(0) = 0.39$. When $\eta < 0.37$, linear dynamics dominate hence the kernel is nearly constant; when $0.37 < \eta < 0.39$, the catapult phase happens in $\boldsymbol{p}_1$ and only $\lambda_1(t)$ decreases; when $0.39 < \eta < \eta_{\max}$, the catapult phase happens in $\boldsymbol{p}_1$ and $\boldsymbol{p}_2$ hence both $\lambda_1(t)$ and $\lambda_2(t)$ decreases. The experiment details can be found in Appendix N.1.

### 3.3 CONNECTION TO GENERAL QUADRATIC MODELS AND WIDE NEURAL NETWORKS

**General quadratic models.** As mentioned in the introduction, NQMs are contained in a general class of quadratic models of the form given in Eq. (3). Additionally, we show that the two-layer linear neural network analyzed in Lewkowycz et al. (2020) is a special case of Eq. (3), and we provide a more general condition for such models to have catapult dynamics in Appendix M. Furthermore, we empirically observe that a broader class of quadratic models $g$ can have catapult dynamics simply by letting $\phi(\boldsymbol{x})$ and $\Sigma$ be random and assigning a small value to $\gamma$ (See Appendix N.4).

**Wide neural networks.** We have seen that NQMs, with fixed Hessian, exhibit the catapult phase phenomenon. Therefore, the change in the Hessian of wide neural networks during training is not required to produce the catapult phase. We will discuss the high-level idea of analyzing the catapult phase for a general NQM with large learning rates, and empirically show that this idea applies to neural networks. We train an NQM Eq. (2) $f_{\text{quad}}$ on $n$ data points $\{(\boldsymbol{x}_i, y_i)\}_{i=1}^n \in \mathbb{R}^d \times \mathbb{R}$ with GD. The dynamics equations take the following form:

$$\mathbf{f}_{\text{quad}}(t+1) - \mathbf{y} = \left( I - \eta K(t) + \underbrace{\frac{1}{2}\eta^2 G(t)\nabla\mathbf{f}_{\text{quad}}(t)^T}_{R_{\mathbf{f}_{\text{quad}}}(t)} \right) (\mathbf{f}_{\text{quad}}(t) - \mathbf{y}), \tag{15}$$

$$K(t+1) = K(t) - \underbrace{\frac{1}{4}\eta \left( 4G(t)\nabla\mathbf{f}_{\text{quad}}(t)^T - \eta G(t)G(t)^T \right)}_{R_K(t)}, \tag{16}$$

where $G_{i,:}(t) = (\mathbf{f}_{\text{quad}}(t) - \mathbf{y})^T \nabla\mathbf{f}_{\text{quad}}(t) H_f(\boldsymbol{x}_i) \in \mathbb{R}^m$ for $i \in [n]$.

In our analysis for $f_{\text{quad}}$ which approximates two-layer networks in Section 3.2, we show that catapult dynamics occur in the top eigenspace of the tangent kernel. Specifically, we analyze the dynamics equations confined to the top eigendirection of the tangent kernel $\boldsymbol{p}_1$ (i.e, $\Pi_1\mathcal{L}$ and $\lambda_1(t)$). We show that $\boldsymbol{p}_1^T R_{\mathbf{f}_{\text{quad}}}\boldsymbol{p}_1$ and $\boldsymbol{p}_1^T R_K \boldsymbol{p}_1$ scale with the loss and remain positive when the loss becomes large, therefore $\boldsymbol{p}_1^T K\boldsymbol{p}_1$ (i.e., $\lambda_{\max}(K)$) as well as the loss will be driven down, and consequently we yield catapult convergence.

We empirically verify catapults indeed happen in the top eigenspace of the tangent kernel for additional NQMs and wide neural networks in Appendix N.3. Furthermore, a similar behaviour of top eigenvalues of the tangent kernel with the one for NQMs is observed for wide neural networks when training with different learning rates (See Figure 5 in Appendix N).

## 4 QUADRATIC MODELS PARALLEL NEURAL NETWORKS IN GENERALIZATION

In this section, we empirically compare the test performance of three different models considered in this paper upon varying learning rate. In particular, we consider (1) the NQM, $f_{\text{quad}}$; (2) corresponding neural networks, $f$; and (3) the linear model, $f_{\text{lin}}$.

We implement our experiments on 3 vision datasets: CIFAR-2 (a 2-class subset of CIFAR-10 (Krizhevsky et al., 2009)), MNIST (LeCun et al., 1998), and SVHN (The Street View House Numbers) (Netzer et al., 2011), 1 speech dataset: Free Spoken Digit dataset (FSDD) (Jakobovski) and 1 text dataset: AG NEWS (Gulli).

In all experiments, we train the models by minimizing the squared loss using standard GD/SGD with constant learning rate $\eta$. We report the best test loss achieved during the training process with each learning rate. Experimental details can be found in Appendix N.5. We also report the best test accuracy in Appendix N.6. For networks with 3 layers, see Appendix N.7. From the experimental results, we observe the following:

**Sub-critical learning rates.** In accordance with our theoretical analyses, we observe that all three models have nearly identical test loss for any sub-critical learning rate. Specifically, note that as the width $m$ increases, $f$ and $f_{\text{quad}}$ will transition to linearity in the ball $B(\mathbf{w}_0, R)$:

$$\|f - f_{\text{lin}}\| = \tilde{O}(1/\sqrt{m}), \quad \|f_{\text{quad}} - f_{\text{lin}}\| = \tilde{O}(1/\sqrt{m}),$$

where $R > 0$ is a constant which is large enough to contain the optimization path with respect to sub-critical learning rates. Thus, the generalization performance of these three models will be similar when $m$ is large, as shown in Figure 4.

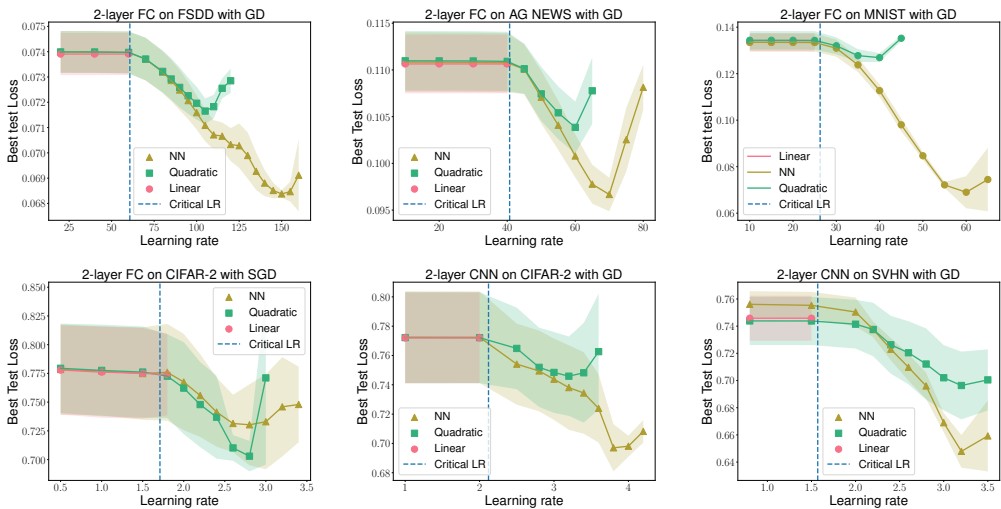

Figure 4: **Best test loss plotted against different learning rates for $f(\mathbf{w})$, $f_{\text{lin}}(\mathbf{w})$ and $f_{\text{quad}}(\mathbf{w})$ across a variety of datasets and network architectures.**

**Super-critical learning rates.** The best test loss of both $f(\mathbf{w})$ and $f_{\text{quad}}(\mathbf{w})$ is consistently smaller than the one with sub-critical learning rates, and decreases for an increasing learning rate in a range of values beyond $\eta_{\text{crit}}$, which was observed for wide neural networks in Lewkowycz et al. (2020).

As discussed in the introduction, with super-critical learning rates, both $f_{\text{quad}}$ and $f$ can be observed to have catapult phase, while the loss of $f_{\text{lin}}$ diverges. Together with the similar behaviour of $f_{\text{quad}}$ and $f$ in generalization with super-critical learning rates, we believe NQMs are a better model to understand $f$ in training and testing dynamics, than the linear approximation $f_{\text{lin}}$.

In Figure 4 we report the results for networks with ReLU activation function. We also implement the experiments using networks with Tanh and Swish (Ramachandran et al., 2017) activation functions, and observe the same phenomena in generalization for $f$, $f_{\text{lin}}$ and $f_{\text{quad}}$ (See Appendix N.8).

## 5 SUMMARY AND DISCUSSION

**Summary.** In this paper, we use quadratic models as a tool to better understand optimization and generalization properties of finite width neural networks trained using large learning rates. Notably, we prove that quadratic models exhibit properties of neural networks such as the catapult phase that cannot be explained using linear models, which importantly includes linear approximations to neural networks given by the neural tangent kernel. Interestingly, we show empirically that quadratic models mimic the generalization properties of neural networks when trained with large learning rate, and that such models perform better than linearized neural networks.

**Future directions.** As quadratic models are more analytically tractable than finite width neural networks, these models open further avenues for understanding the good performance of finite width networks in practice. In particular, one interesting direction of future work is to understand the change in the kernel corresponding to a trained quadratic model. As we showed, training a quadratic model with large learning rate causes a decrease in the eigenvalues of the neural tangent kernel, and it would be interesting to understand the properties of this changed kernel that correspond with improved generalization. Indeed, prior work Long (2021) has analyzed the properties of the "after kernel" corresponding to finite width neural networks, and it would be interesting to observe whether similar properties hold for the kernel corresponding to trained quadratic models.

Another interesting avenue of research is to understand whether quadratic models can be used for representation learning. Indeed, prior work Yang & Hu (2020) argues that networks in the neural tangent kernel regime do not learn useful representations of data through training. As quadratic models trained with large learning rate can already exhibit nonlinear dynamics and better capture generalization properties of finite width networks, it would be interesting to understand whether such models learn useful representations of data as well.

## ACKNOWLEDGEMENTS

We thank Boris Hanin, Daniel A. Roberts and Sho Yaida for the discussion about quadratic models and catapults. A.R. is funded by the George F. Carrier fellowship at Harvard School of Engineering and Applied Sciences. We are grateful for the support from the National Science Foundation (NSF) and the Simons Foundation for the Collaboration on the Theoretical Foundations of Deep Learning (https://deepfoundations.ai/) through awards DMS-2031883 and #814639 and the TILOS institute (NSF CCF-2112665). This work used NVIDIA V100 GPUs NVLINK and HDR IB (Expanse GPU) at SDSC Dell Cluster through allocation TG-CIS220009 and also, Delta system at the National Center for Supercomputing Applications through allocation bbjr-delta-gpu from the Advanced Cyberinfrastructure Coordination Ecosystem: Services & Support (ACCESS) program, which is supported by National Science Foundation grants #2138259, #2138286, #2138307, #2137603, and #2138296.

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

## APPENDIX

## A    DERIVATION OF NQM

We will derive the NQM that approximate the two-layer fully connected ReLU activated neural networks based on Eq. (2).

The first derivative of $f$ can be computed by:

$$\frac{\partial f}{\partial \mathbf{u}_i} = \frac{1}{\sqrt{md}} v_i \mathbb{1}_{\{\mathbf{u}_i^T \boldsymbol{x} \geq 0\}} \boldsymbol{x}^T, \quad \frac{\partial f}{\partial v_i} = \frac{1}{\sqrt{m}} \sigma\left(\frac{1}{\sqrt{d}} \mathbf{u}_i^T \boldsymbol{x}\right), \quad \forall i \in [m].$$

And each entry of the Hessian of $f$, i.e., $H_f$, can be computed by

$$\frac{\partial^2 f}{\partial \mathbf{u}_i^2} = \mathbf{0}, \quad \frac{\partial^2 f}{\partial v_i^2} = 0, \quad \frac{\partial^2 f}{\partial \mathbf{u}_i v_i} = \frac{1}{\sqrt{md}} \mathbb{1}_{\{\mathbf{u}_i^T \boldsymbol{x} \geq 0\}} \boldsymbol{x}^T, \quad \forall i \in [m].$$

Now we get $f_{\text{quad}}$ taking the following form

$$\mathbf{NQM}: f_{\text{quad}}(\mathbf{u}, \mathbf{v}; \boldsymbol{x}) = f(\mathbf{u}_0, \mathbf{v}_0; \boldsymbol{x}) + \frac{1}{\sqrt{md}} \sum_{i=1}^{m} (\mathbf{u}_i - \mathbf{u}_{0,i})^T \boldsymbol{x} \mathbb{1}_{\{\mathbf{u}_{0,i}^T \boldsymbol{x} \geq 0\}} v_{0,i} + \frac{1}{\sqrt{m}} \sum_{i=1}^{m} (v_i - v_{0,i}) \sigma\left(\frac{1}{\sqrt{d}} \mathbf{u}_{0,i}^T \boldsymbol{x}\right)$$

$$+ \frac{1}{\sqrt{md}} \sum_{i=1}^{m} (\mathbf{u}_i - \mathbf{u}_{0,i})^T \boldsymbol{x} \mathbb{1}_{\{\mathbf{u}_{0,i}^T \boldsymbol{x} \geq 0\}} (v_i - v_{0,i}). \tag{17}$$

## B    DERIVATION OF DYNAMICS EQUATIONS

For simplicity of notation, we denote $f_{\text{quad}}$ by $g$. Note that at initialization, the first and second derivatives of $f$ with respect to parameters are the same as those of $g$.

### B.1    SINGLE TRAINING EXAMPLE

The NQM can be equivalently written as:

$$g(\mathbf{u}, \mathbf{v}; \boldsymbol{x}) = g(\mathbf{u}_0, \mathbf{v}_0; \boldsymbol{x}) + \left\langle \mathbf{u} - \mathbf{u}_0, \nabla_{\mathbf{u}} g(\mathbf{u}, \mathbf{v}; \boldsymbol{x}) \Big|_{\mathbf{u}=\mathbf{u}_0, \mathbf{v}=\mathbf{v}_0} \right\rangle + \left\langle \mathbf{v} - \mathbf{v}_0, \nabla_{\mathbf{v}} g(\mathbf{u}, \mathbf{v}; \boldsymbol{x}) \Big|_{\mathbf{u}=\mathbf{u}_0, \mathbf{v}=\mathbf{v}_0} \right\rangle$$

$$+ \left\langle \mathbf{u} - \mathbf{u}_0, \frac{\partial^2 g(\mathbf{u}, \mathbf{v}; \boldsymbol{x})}{\partial \mathbf{u} \partial \mathbf{v}} \Big|_{\mathbf{u}=\mathbf{u}_0, \mathbf{v}=\mathbf{v}_0} (\mathbf{v} - \mathbf{v}_0) \right\rangle,$$

since $\frac{\partial^2 g}{\partial \mathbf{u}^2} = 0$ and $\frac{\partial^2 g}{\partial \mathbf{v}^2} = 0$.

And the tangent kernel $\lambda(\mathbf{u}, \mathbf{v}; \boldsymbol{x})$ takes the form

$$\lambda(\mathbf{u}, \mathbf{v}; \boldsymbol{x}) = \left\| \nabla_{\mathbf{u}} g(\mathbf{u}, \mathbf{v}; \boldsymbol{x}) \Big|_{\mathbf{u}=\mathbf{u}_0, \mathbf{v}=\mathbf{v}_0} + \frac{\partial^2 g(\mathbf{u}, \mathbf{v}; \boldsymbol{x})}{\partial \mathbf{u} \partial \mathbf{v}} \Big|_{\mathbf{u}=\mathbf{u}_0} (\mathbf{v} - \mathbf{v}_0) \right\|_F^2$$

$$+ \left\| \nabla_{\mathbf{v}} g(\mathbf{u}, \mathbf{v}; \boldsymbol{x}) \Big|_{\mathbf{u}=\mathbf{u}_0, \mathbf{v}=\mathbf{v}_0} + (\mathbf{u} - \mathbf{u}_0)^T \frac{\partial^2 g(\mathbf{u}, \mathbf{v}; \boldsymbol{x})}{\partial \mathbf{u} \partial \mathbf{v}} \Big|_{\mathbf{u}=\mathbf{u}_0, \mathbf{v}=\mathbf{v}_0} \right\|^2.$$

Here

$$\nabla_{\mathbf{u}_i} g(\mathbf{u}, \mathbf{v}; \boldsymbol{x}) \Big|_{\mathbf{u}=\mathbf{u}_0, \mathbf{v}=\mathbf{v}_0} = \frac{1}{\sqrt{md}} \sum_{i=1}^{m} v_{0,i} \mathbb{1}_{\{\mathbf{u}_{0,i}^T \boldsymbol{x} \geq 0\}} \boldsymbol{x}, \quad \forall i \in [m],$$

$$\nabla_{\mathbf{v}} g(\mathbf{u}, \mathbf{v}; \boldsymbol{x}) \Big|_{\mathbf{u}=\mathbf{u}_0, \mathbf{v}=\mathbf{v}_0} = \frac{1}{\sqrt{md}} \sigma\left(\mathbf{u}_0^T \boldsymbol{x}\right).$$

In the following, we will consider the dynamics of $g$ and $\lambda$ with GD, hence for simplicity of notations, we denote

$$\nabla_{\mathbf{u}} g(0) := \nabla_{\mathbf{u}} g(\mathbf{u}, \mathbf{v}; \boldsymbol{x})\Big|_{\mathbf{u}=\mathbf{u}_0, \mathbf{v}=\mathbf{v}_0},$$

$$\nabla_{\mathbf{v}} g(0) := \nabla_{\mathbf{v}} g(\mathbf{u}, \mathbf{v}; \boldsymbol{x})\Big|_{\mathbf{u}=\mathbf{u}_0, \mathbf{v}=\mathbf{v}_0},$$

$$\frac{\partial^2 g(0)}{\partial \mathbf{u} \partial \mathbf{v}} := \frac{\partial^2 g(\mathbf{u}, \mathbf{v}; \boldsymbol{x})}{\partial \mathbf{u} \partial \mathbf{v}}\Big|_{\mathbf{u}=\mathbf{u}_0, \mathbf{v}=\mathbf{v}_0}.$$

By gradient descent with learning rate $\eta$, at iteration $t$, we have the update equations for weights $\mathbf{u}$ and $\mathbf{v}$:

$$\mathbf{u}(t+1) = \mathbf{u}(t) - \eta(g(t) - y)\left(\nabla_{\mathbf{u}} g(0) + \frac{\partial^2 g(0)}{\partial \mathbf{u} \partial \mathbf{v}}(\mathbf{v}(t) - \mathbf{v}(0))\right),$$

$$\mathbf{v}(t+1) = \mathbf{v}(t) - \eta(g(t) - y)\left(\nabla_{\mathbf{v}} g(0) + (\mathbf{u}(t) - \mathbf{u}(0))^T \frac{\partial^2 g(0)}{\partial \mathbf{u} \partial \mathbf{v}}\right).$$

Then we plug them in the expression of $\lambda(t+1)$ and we get

$$\lambda(t+1) = \left\|\nabla_{\mathbf{u}} g(0) + \frac{\partial^2 g(0)}{\partial \mathbf{u} \partial \mathbf{v}}(\mathbf{v}(t+1) - \mathbf{v}(0))\right\|_F^2 + \left\|\nabla_{\mathbf{v}} g(0) + (\mathbf{u}(t+1) - \mathbf{u}(0))^T \frac{\partial^2 g(0)}{\partial \mathbf{u} \partial \mathbf{v}}\right\|^2$$

$$= \left\|\nabla_{\mathbf{u}} g(0) + \frac{\partial^2 g(0)}{\partial \mathbf{u} \partial \mathbf{v}}\left(\mathbf{v}(t) - \eta(g(t) - y)\left(\nabla_{\mathbf{v}} g(0) + (\mathbf{u}(t) - \mathbf{u}(0))^T \frac{\partial^2 g(0)}{\partial \mathbf{u} \partial \mathbf{v}}\right) - \mathbf{v}(0)\right)\right\|_F^2$$

$$+ \left\|\nabla_{\mathbf{v}} g(0) + \left(\mathbf{u}(t) - \eta(g(t) - y)\left(\nabla_{\mathbf{u}} g(0) + \frac{\partial^2 g(0)}{\partial \mathbf{u} \partial \mathbf{v}}(\mathbf{v}(t) - \mathbf{v}(0))\right) - \mathbf{u}(0)\right)^T \frac{\partial^2 g(0)}{\partial \mathbf{u} \partial \mathbf{v}}\right\|^2$$

$$= \lambda(t) + \eta^2(g(t) - y)^2 \left\|\frac{\partial^2 g(0)}{\partial \mathbf{u} \partial \mathbf{v}}\left(\nabla_{\mathbf{v}} g(0) + (\mathbf{u}(t) - \mathbf{u}(0))^T \frac{\partial^2 g(0)}{\partial \mathbf{u} \partial \mathbf{v}}\right)\right\|_F^2$$

$$+ \eta^2(g(t) - y)^2 \left\|\left(\nabla_{\mathbf{u}} g(0) + \frac{\partial^2 g(0)}{\partial \mathbf{u} \partial \mathbf{v}}(\mathbf{v}(t) - \mathbf{v}(0))\right)^T \frac{\partial^2 g(0)}{\partial \mathbf{u} \partial \mathbf{v}}\right\|^2$$

$$- 2\eta(g(t) - y)\left\langle\nabla_{\mathbf{u}} g(0) + \frac{\partial^2 g(0)}{\partial \mathbf{u} \partial \mathbf{v}}(\mathbf{v}(t) - \mathbf{v}(0)), \frac{\partial^2 g(0)}{\partial \mathbf{u} \partial \mathbf{v}}\left(\nabla_{\mathbf{v}} g(0) + (\mathbf{u}(t) - \mathbf{u}(0))^T \frac{\partial^2 g(0)}{\partial \mathbf{u} \partial \mathbf{v}}\right)\right\rangle$$

$$- 2\eta(g(t) - y)\left\langle\nabla_{\mathbf{v}} g(0) + (\mathbf{u}(t) - \mathbf{u}(0))^T \frac{\partial^2 g(0)}{\partial \mathbf{u} \partial \mathbf{v}}, \left(\nabla_{\mathbf{u}} g(0) + \frac{\partial^2 g(0)}{\partial \mathbf{u} \partial \mathbf{v}}(\mathbf{v}(t) - \mathbf{v}(0))\right)^T \frac{\partial^2 g(0)}{\partial \mathbf{u} \partial \mathbf{v}}\right\rangle.$$

Due to the structure of $\frac{\partial^2 g(0)}{\partial \mathbf{u} \partial \mathbf{v}}$, we have

$$\left\|\frac{\partial^2 g(0)}{\partial \mathbf{u} \partial \mathbf{v}}\left(\nabla_{\mathbf{v}} g(0) + (\mathbf{u}(t) - \mathbf{u}(0))^T \frac{\partial^2 g(0)}{\partial \mathbf{u} \partial \mathbf{v}}\right)\right\|_F^2 = \frac{\|\boldsymbol{x}\|^2}{md}\left\|\nabla_{\mathbf{v}} g(0) + (\mathbf{u}(t) - \mathbf{u}(0))^T \frac{\partial^2 g(0)}{\partial \mathbf{u} \partial \mathbf{v}}\right\|^2$$

$$= \frac{\|\boldsymbol{x}\|^2}{md}\|\nabla_{\mathbf{v}} g(t)\|^2,$$

and

$$\left\|\left(\nabla_{\mathbf{u}} g(0) + \frac{\partial^2 g(0)}{\partial \mathbf{u} \partial \mathbf{v}}(\mathbf{v}(t) - \mathbf{v}(0))\right)^T \frac{\partial^2 g(0)}{\partial \mathbf{u} \partial \mathbf{v}}\right\|^2 = \frac{\|\boldsymbol{x}\|^2}{md}\left\|\nabla_{\mathbf{u}} g(0) + \frac{\partial^2 g(0)}{\partial \mathbf{u} \partial \mathbf{v}}(\mathbf{v}(t) - \mathbf{v}(0))\right\|_F^2$$

$$= \frac{\|\boldsymbol{x}\|^2}{md}\|\nabla_{\mathbf{u}} g(t)\|_F^2.$$

Furthermore,

$$\left\langle \nabla_{\mathbf{u}}g(0) + \frac{\partial^2 g(0)}{\partial \mathbf{u} \partial \mathbf{v}}(\mathbf{v}(t) - \mathbf{v}(0)), \frac{\partial^2 g(0)}{\partial \mathbf{u} \partial \mathbf{v}}\left(\nabla_{\mathbf{v}}g(0) + (\mathbf{u}(t) - \mathbf{u}(0))^T \frac{\partial^2 g(0)}{\partial \mathbf{u} \partial \mathbf{v}}\right) \right\rangle$$

$$= \frac{\|\boldsymbol{x}\|^2}{md} \langle \mathbf{v}(t) - \mathbf{v}(0), \nabla_{\mathbf{v}}g(0) \rangle + \frac{\|\boldsymbol{x}\|^2}{md} \langle \nabla_{\mathbf{u}}g(0), \mathbf{u}(t) - \mathbf{u}(0) \rangle + \left\langle \nabla_{\mathbf{u}}g(0), \frac{\partial^2 g(0)}{\partial \mathbf{u} \partial \mathbf{v}} \nabla_{\mathbf{v}}g(0) \right\rangle$$

$$+ \left\langle \frac{\partial^2 g(0)}{\partial \mathbf{u} \partial \mathbf{v}}(\mathbf{v}(t) - \mathbf{v}(0)), \frac{\partial^2 g(0)}{\partial \mathbf{u} \partial \mathbf{v}}(\mathbf{u}(t) - \mathbf{u}(0))^T \frac{\partial^2 g(0)}{\partial \mathbf{u} \partial \mathbf{v}} \right\rangle$$

$$= \frac{\|\boldsymbol{x}\|^2}{md} \langle \mathbf{v}(t) - \mathbf{v}(0), \nabla_{\mathbf{v}}g(0) \rangle + \frac{\|\boldsymbol{x}\|^2}{md} \langle \nabla_{\mathbf{u}}g(0), \mathbf{u}(t) - \mathbf{u}(0) \rangle + g(0) + \frac{\|\boldsymbol{x}\|^2}{md} \left\langle \mathbf{v}(t) - \mathbf{v}(0), \frac{\partial^2 g(0)}{\partial \mathbf{u} \partial \mathbf{v}}(\mathbf{u}(t) - \mathbf{u}(0))^T \right\rangle$$

$$= g(t)\|\boldsymbol{x}\|^2/md.$$

Similarly, we have

$$\left\langle \nabla_{\mathbf{v}}g(0) + (\mathbf{u}(t) - \mathbf{u}(0))^T \frac{\partial^2 g(0)}{\partial \mathbf{u} \partial \mathbf{v}}, \left(\nabla_{\mathbf{u}}g(0) + \frac{\partial^2 g(0)}{\partial \mathbf{u} \partial \mathbf{v}}(\mathbf{v}(t) - \mathbf{v}(0))\right)^T \frac{\partial^2 g(0)}{\partial \mathbf{u} \partial \mathbf{v}} \right\rangle = g(t)\|\boldsymbol{x}\|^2/md.$$

As a result,

$$\lambda(t+1) = \lambda(t) + \frac{\|\boldsymbol{x}\|^2}{md}\eta^2(g(t) - y)^2\lambda(t) - \frac{4\|\boldsymbol{x}\|^2}{md}\eta(g(t) - y)g(t)$$

$$= \lambda(t) + \eta\frac{\|\boldsymbol{x}\|^2}{md}(g(t) - y)^2\left(\eta\lambda(t) - 4\frac{g(t)}{g(t) - y}\right).$$

For $g$, we plug the update equations for $\mathbf{u}$ and $\mathbf{v}$ in the expression of $g(t+1)$ and we can get

$$g(t+1) = g(0) + \langle \mathbf{u}(t+1) - \mathbf{u}(0), \nabla_{\mathbf{u}}g(0) \rangle + \langle \mathbf{v}(t+1) - \mathbf{v}(0), \nabla_{\mathbf{v}}g(0) \rangle$$

$$+ \left\langle \mathbf{u}(t+1) - \mathbf{u}(0), \frac{\partial^2 g(0)}{\partial \mathbf{u} \partial \mathbf{v}}(\mathbf{v}(t+1) - \mathbf{v}(0)) \right\rangle$$

$$= g(0) + \left\langle \mathbf{u}(t) - \eta(g(t) - y)\left(\nabla_{\mathbf{u}}g(0) + \frac{\partial^2 g(0)}{\partial \mathbf{u} \partial \mathbf{v}}(\mathbf{v}(t) - \mathbf{v}(0))\right) - \mathbf{u}(0), \nabla_{\mathbf{u}}g(0) \right\rangle$$

$$+ \left\langle \mathbf{v}(t) - \eta(g(t) - y)\left(\nabla_{\mathbf{v}}g(0) + (\mathbf{u}(t) - \mathbf{u}(0))^T \frac{\partial^2 g(0)}{\partial \mathbf{u} \partial \mathbf{v}}\right) - \mathbf{v}(0), \nabla_{\mathbf{v}}g(0) \right\rangle$$

$$+ \left\langle \mathbf{u}(t) - \eta(g(t) - y)\left(\nabla_{\mathbf{u}}g(0) + \frac{\partial^2 g(0)}{\partial \mathbf{u} \partial \mathbf{v}}(\mathbf{v}(t) - \mathbf{v}(0))\right) - \mathbf{u}(0), \right.$$

$$\left. \frac{\partial^2 g(0)}{\partial \mathbf{u} \partial \mathbf{v}}\left(\mathbf{v}(t) - \eta(g(t) - y)\left(\nabla_{\mathbf{v}}g(0) + (\mathbf{u}(t) - \mathbf{u}(0))^T \frac{\partial^2 g(0)}{\partial \mathbf{u} \partial \mathbf{v}}\right) - \mathbf{v}(0)\right) \right\rangle$$

$$= g(t) - \eta(g(t) - y)\left\langle \nabla_{\mathbf{u}}g(0) + \frac{\partial^2 g(0)}{\partial \mathbf{u} \partial \mathbf{v}}(\mathbf{v}(t) - \mathbf{v}(0)), \nabla_{\mathbf{u}}g(0) \right\rangle$$

$$- \eta(g(t) - y)\left\langle \nabla_{\mathbf{v}}g(0) + (\mathbf{u}(t) - \mathbf{u}(0))^T \frac{\partial^2 g(0)}{\partial \mathbf{u} \partial \mathbf{v}}, \nabla_{\mathbf{v}}g(0) \right\rangle$$

$$+ \eta^2(g(t) - y)^2\left\langle \nabla_{\mathbf{u}}g(0) + \frac{\partial^2 g(0)}{\partial \mathbf{u} \partial \mathbf{v}}(\mathbf{v}(t) - \mathbf{v}(0)), \frac{\partial^2 g(0)}{\partial \mathbf{u} \partial \mathbf{v}}\left(\nabla_{\mathbf{v}}g(0) + (\mathbf{u}(t) - \mathbf{u}(0))^T \frac{\partial^2 g(0)}{\partial \mathbf{u} \partial \mathbf{v}}\right) \right\rangle$$

$$- \eta(g(t) - y)\left\langle \mathbf{u}(t) - \mathbf{u}(0), \frac{\partial^2 g(0)}{\partial \mathbf{u} \partial \mathbf{v}}\left(\nabla_{\mathbf{v}}g(0) + (\mathbf{u}(t) - \mathbf{u}(0))^T \frac{\partial^2 g(0)}{\partial \mathbf{u} \partial \mathbf{v}}\right) \right\rangle$$

$$- \eta(g(t) - y)\left\langle \nabla_{\mathbf{u}}g(0) + \frac{\partial^2 g(0)}{\partial \mathbf{u} \partial \mathbf{v}}(\mathbf{v}(t) - \mathbf{v}(0)), \frac{\partial^2 g(0)}{\partial \mathbf{u} \partial \mathbf{v}}(\mathbf{v}(t) - \mathbf{v}(0)) \right\rangle$$

$$= g(t) - \eta(g(t) - y)\lambda(t)$$

$$+ \eta^2(g(t) - y)^2\left\langle \nabla_{\mathbf{u}}g(0) + \frac{\partial^2 g(0)}{\partial \mathbf{u} \partial \mathbf{v}}(\mathbf{v}(t) - \mathbf{v}(0)), \frac{\partial^2 g(0)}{\partial \mathbf{u} \partial \mathbf{v}}\left(\nabla_{\mathbf{v}}g(0) + (\mathbf{u}(t) - \mathbf{u}(0))^T \frac{\partial^2 g(0)}{\partial \mathbf{u} \partial \mathbf{v}}\right) \right\rangle$$

$$= g(t) - \eta(g(t) - y)\lambda(t) + \frac{\|\boldsymbol{x}\|^2}{md}\eta^2(g(t) - y)^2 g(t)$$

Therefore,

$$g(t+1) - y = \left(1 - \eta\lambda(t) + \frac{\|x\|^2}{md}\eta^2(g(t) - y)g(t)\right)(g(t) - y).$$

## B.2 Multiple training examples

We follow the similar notation on the first and second order derivative of $g$ with Appendix B.1. Specifically, for $k \in [n]$, we denote

$$\nabla_{\mathbf{u}}g_k(0) := \nabla_{\mathbf{u}}g(\mathbf{u}, \mathbf{v}; x_k)\Big|_{\mathbf{u}=\mathbf{u}_0, \mathbf{v}=\mathbf{v}_0},$$

$$\nabla_{\mathbf{v}}g_k(0) := \nabla_{\mathbf{v}}g(\mathbf{u}, \mathbf{v}; x_k)\Big|_{\mathbf{u}=\mathbf{u}_0, \mathbf{v}=\mathbf{v}_0},$$

$$\frac{\partial^2 g_k(0)}{\partial\mathbf{u}\partial\mathbf{v}} := \frac{\partial^2 g(\mathbf{u}, \mathbf{v}; x_k)}{\partial\mathbf{u}\partial\mathbf{v}}\Big|_{\mathbf{u}=\mathbf{u}_0, \mathbf{v}=\mathbf{v}_0}.$$

By GD with learning rate $\eta$, we have the update equations for weights $\mathbf{u}$ and $\mathbf{v}$ at iteration $t$:

$$\mathbf{u}(t+1) = \mathbf{u}(t) - \eta\sum_{k=1}^{n}(g_k(t) - y_k)\left(\nabla_{\mathbf{u}}g_k(0) + \frac{\partial^2 g_k(0)}{\partial\mathbf{u}\partial\mathbf{v}}(\mathbf{v}(t) - \mathbf{v}(0))\right),$$

$$\mathbf{v}(t+1) = \mathbf{v}(t) - \eta\sum_{k=1}^{n}(g_k(t) - y_k)\left(\nabla_{\mathbf{v}}g_k(0) + (\mathbf{u}(t) - \mathbf{u}(0))^T\frac{\partial^2 g_k(0)}{\partial\mathbf{u}\partial\mathbf{v}}\right).$$

We consider the evolution of $K(t)$ first.

$$K_{i,j}(t+1) = \left\langle\nabla_{\mathbf{u}}g_i(0) + \frac{\partial^2 g_i(0)}{\partial\mathbf{u}\partial\mathbf{v}}(\mathbf{v}(t+1) - \mathbf{v}(0)), \nabla_{\mathbf{u}}g_j(0) + \frac{\partial^2 g_j(0)}{\partial\mathbf{u}\partial\mathbf{v}}(\mathbf{v}(t+1) - \mathbf{v}(0))\right\rangle$$

$$+ \left\langle\nabla_{\mathbf{v}}g_i(0) + (\mathbf{u}(t+1) - \mathbf{u}(0))^T\frac{\partial^2 g_i(0)}{\partial\mathbf{u}\partial\mathbf{v}}, \nabla_{\mathbf{v}}g_j(0) + (\mathbf{u}(t+1) - \mathbf{u}(0))^T\frac{\partial^2 g_j(0)}{\partial\mathbf{u}\partial\mathbf{v}}\right\rangle$$

$$= K_{i,j}(t) - \left\langle\eta\frac{\partial^2 g_i(0)}{\partial\mathbf{u}\partial\mathbf{v}}\sum_{k=1}^{n}(g_k(t) - y_k)\left(\nabla_{\mathbf{v}}g_k(0) + (\mathbf{u}(t) - \mathbf{u}(0))^T\frac{\partial^2 g_k(0)}{\partial\mathbf{u}\partial\mathbf{v}}\right),\right.$$

$$\left.\nabla_{\mathbf{u}}g_j(0) + \frac{\partial^2 g_j(0)}{\partial\mathbf{u}\partial\mathbf{v}}(\mathbf{v}(t) - \mathbf{v}(0))\right\rangle$$

$$- \left\langle\eta\frac{\partial^2 g_j(0)}{\partial\mathbf{u}\partial\mathbf{v}}\sum_{k=1}^{n}(g_k(t) - y_k)\left(\nabla_{\mathbf{v}}g_k(0) + (\mathbf{u}(t) - \mathbf{u}(0))^T\frac{\partial^2 g_k(0)}{\partial\mathbf{u}\partial\mathbf{v}}\right), \nabla_{\mathbf{u}}g_i(0) + \frac{\partial^2 g_i(0)}{\partial\mathbf{u}\partial\mathbf{v}}(\mathbf{v}(t) - \mathbf{v}(0))\right\rangle$$

$$+ \left\langle\eta\frac{\partial^2 g_i(0)}{\partial\mathbf{u}\partial\mathbf{v}}\sum_{k=1}^{n}(g_k(t) - y_k)\left(\nabla_{\mathbf{v}}g_k(0) + (\mathbf{u}(t) - \mathbf{u}(0))^T\frac{\partial^2 g_k(0)}{\partial\mathbf{u}\partial\mathbf{v}}\right),\right.$$

$$\left.\eta\frac{\partial^2 g_j(0)}{\partial\mathbf{u}\partial\mathbf{v}}\sum_{k=1}^{n}(g_k(t) - y_k)\left(\nabla_{\mathbf{v}}g_k(0) + (\mathbf{u}(t) - \mathbf{u}(0))^T\frac{\partial^2 g_k(0)}{\partial\mathbf{u}\partial\mathbf{v}}\right)\right\rangle$$

$$- \left\langle\eta\frac{\partial^2 g_j(0)}{\partial\mathbf{u}\partial\mathbf{v}}\sum_{k=1}^{n}(g_k(t) - y_k)\left(\nabla_{\mathbf{u}}g_k(0) + \frac{\partial^2 g_k(0)}{\partial\mathbf{u}\partial\mathbf{v}}(\mathbf{v}(t) - \mathbf{v}(0))\right), \nabla_{\mathbf{v}}g_i(0) + (\mathbf{u}(t) - \mathbf{u}(0))^T\frac{\partial^2 g_i(0)}{\partial\mathbf{u}\partial\mathbf{v}}\right\rangle$$

$$- \left\langle\eta\frac{\partial^2 g_i(0)}{\partial\mathbf{u}\partial\mathbf{v}}\sum_{k=1}^{n}(g_k(t) - y_k)\left(\nabla_{\mathbf{u}}g_k(0) + \frac{\partial^2 g_k(0)}{\partial\mathbf{u}\partial\mathbf{v}}(\mathbf{v}(t) - \mathbf{v}(0))\right), \nabla_{\mathbf{v}}g_j(0) + (\mathbf{u}(t) - \mathbf{u}(0))^T\frac{\partial^2 g_j(0)}{\partial\mathbf{u}\partial\mathbf{v}}\right\rangle$$

$$+ \left\langle\eta\frac{\partial^2 g_i(0)}{\partial\mathbf{u}\partial\mathbf{v}}\sum_{k=1}^{n}(g_k(t) - y_k)\left(\nabla_{\mathbf{u}}g_k(0) + \frac{\partial^2 g_k(0)}{\partial\mathbf{u}\partial\mathbf{v}}(\mathbf{v}(t) - \mathbf{v}(0))\right),\right.$$

$$\left.\eta\frac{\partial^2 g_j(0)}{\partial\mathbf{u}\partial\mathbf{v}}\sum_{k=1}^{n}(g_k(t) - y_k)\left(\nabla_{\mathbf{u}}g_k(0) + \frac{\partial^2 g_k(0)}{\partial\mathbf{u}\partial\mathbf{v}}(\mathbf{v}(t) - \mathbf{v}(0))\right)\right\rangle.$$

We separate the data into two sets according to their sign:

$$\mathcal{S}_+ := \{i : x_i \geq 0, i \in [n]\}, \quad \mathcal{S}_- := \{i : x_i < 0, i \in [n]\}.$$

We consider two scenarios: (1) $x_i$ and $x_j$ have different signs; (2) $x_i$ and $x_j$ have the same sign.

**(1)** With simple calculation, we get if $x_i$ and $x_j$ have different signs, i.e., $i \in \mathcal{S}_+, j \in \mathcal{S}_-$ or $i \in \mathcal{S}_-, j \in \mathcal{S}_+$,

$$\frac{\partial^2 g_i(0)}{\partial \mathbf{u} \partial \mathbf{v}} \frac{\partial^2 g_j(0)}{\partial \mathbf{u} \partial \mathbf{v}} = 0, \quad \frac{\partial^2 g_i(0)}{\partial \mathbf{u} \partial \mathbf{v}} \nabla_{\mathbf{u}} g_j(0) = 0, \quad \frac{\partial^2 g_i(0)}{\partial \mathbf{u} \partial \mathbf{v}} \nabla_{\mathbf{v}} g_j(0) = 0.$$

Without lose of generality, we assume $i \in \mathcal{S}_+, j \in \mathcal{S}_-$. Then we have

$$K_{i,j}(t+1) = K_{i,j}(t).$$

**(2)** If $x_i$ and $x_j$ have the same sign, i.e., $i, j \in \mathcal{S}_+$ or $i, j \in \mathcal{S}_-$,

$$\frac{\partial^2 g_i(0)}{\partial \mathbf{u} \partial \mathbf{v}} \frac{\partial^2 g_j(0)}{\partial \mathbf{u} \partial \mathbf{v}} = \frac{1}{\sqrt{m}} \frac{\partial^2 g_i(0)}{\partial \mathbf{u} \partial \mathbf{v}} x_j, \quad \frac{\partial^2 g_i(0)}{\partial \mathbf{u} \partial \mathbf{v}} \nabla_{\mathbf{u}} g_j(0) = \frac{1}{\sqrt{m}} \nabla_{\mathbf{u}} g_i(0) x_j, \quad \frac{\partial^2 g_i(0)}{\partial \mathbf{u} \partial \mathbf{v}} \nabla_{\mathbf{v}} g_j(0) = \frac{1}{\sqrt{m}} \nabla_{\mathbf{v}} g_i(0) x_j.$$

For $i, j \in \mathcal{S}_+$, we have

$$K_{i,j}(t+1) = K_{i,j}(t) - \frac{2\eta}{\sqrt{m}} \sum_{k \in \mathcal{S}_+} (g_k(t) - y_k) x_i \left\langle \nabla_{\mathbf{v}} g_k(0) + (\mathbf{u}(t) - \mathbf{u}(0))^T \frac{\partial^2 g_k(0)}{\partial \mathbf{u} \partial \mathbf{v}}, \right.$$

$$\left. \nabla_{\mathbf{u}} g_j(0) + \frac{\partial^2 g_j(0)}{\partial \mathbf{u} \partial \mathbf{v}} (\mathbf{v}(t) - \mathbf{v}(0)) \right\rangle$$

$$- \frac{2\eta}{\sqrt{m}} \sum_{k \in \mathcal{S}_+} (g_k(t) - y_k) x_i \left\langle \nabla_{\mathbf{u}} g_k(0) + \frac{\partial^2 g_k(0)}{\partial \mathbf{u} \partial \mathbf{v}} (\mathbf{v}(t) - \mathbf{v}(0)), \nabla_{\mathbf{v}} g_j(0) + (\mathbf{u}(t) - \mathbf{u}(0))^T \frac{\partial^2 g_j(0)}{\partial \mathbf{u} \partial \mathbf{v}} \right\rangle$$

$$+ \frac{\eta^2}{m} x_i x_j \left\| \sum_{k \in \mathcal{S}_+} (g_k(t) - y_k) \left( \nabla_{\mathbf{v}} g_k(0) + (\mathbf{u}(t) - \mathbf{u}(0))^T \frac{\partial^2 g_k(0)}{\partial \mathbf{u} \partial \mathbf{v}} \right) \right\|^2$$

$$+ \frac{\eta^2}{m} x_i x_j \left\| \sum_{k \in \mathcal{S}_+} (g_k(t) - y_k) \left( \nabla_{\mathbf{u}} g_k(0) + \frac{\partial^2 g_k(0)}{\partial \mathbf{u} \partial \mathbf{v}} (\mathbf{v}(t) - \mathbf{v}(0)) \right) \right\|^2$$

$$= K_{i,j}(t) - \frac{4\eta}{m} x_i x_j \sum_{k \in \mathcal{S}_+} (g_k(t) - y_k) g_k(t) + \frac{\eta^2}{m} x_i x_j \left( (\mathbf{g}(t) - \mathbf{y}) \odot \boldsymbol{m}_+ \right)^T K(t) \left( (\mathbf{g}(t) - \mathbf{y}) \odot \boldsymbol{m}_+ \right)$$

$$= K_{i,j}(t) - \frac{4\eta}{m} x_i x_j \left( (\mathbf{g}(t) - \mathbf{y}) \odot \boldsymbol{m}_+ \right)^T \left( \mathbf{g}(t) \odot \boldsymbol{m}_+ \right)$$

$$+ \frac{\eta^2}{m} x_i x_j \left( (\mathbf{g}(t) - \mathbf{y}) \odot \boldsymbol{m}_+ \right)^T K(t) \left( (\mathbf{g}(t) - \mathbf{y}) \odot \boldsymbol{m}_+ \right).$$

Similarly, for $i, j \in \mathcal{S}_-$, we have

$$K_{i,j}(t+1) = K_{i,j}(t) - \frac{4\eta}{m} x_i x_j \left( (\mathbf{g}(t) - \mathbf{y}) \odot \boldsymbol{m}_- \right)^T \left( \mathbf{g}(t) \odot \boldsymbol{m}_- \right)$$

$$+ \frac{\eta^2}{m} x_i x_j \left( (\mathbf{g}(t) - \mathbf{y}) \odot \boldsymbol{m}_- \right)^T K(t) \left( (\mathbf{g}(t) - \mathbf{y}) \odot \boldsymbol{m}_- \right).$$

Combining the results together, we have

$$K(t+1) = K(t) + \frac{\eta^2}{m} \left( (\mathbf{g}(t) - \mathbf{y}) \odot \boldsymbol{m}_+ \right)^T K(t) \left( (\mathbf{g}(t) - \mathbf{y}) \odot \boldsymbol{m}_+ \right) \boldsymbol{p}_1 \boldsymbol{p}_1^T$$

$$+ \frac{\eta^2}{m} \left( (\mathbf{g}(t) - \mathbf{y}) \odot \boldsymbol{m}_- \right)^T K(t) \left( (\mathbf{g}(t) - \mathbf{y}) \odot \boldsymbol{m}_- \right) \boldsymbol{p}_2 \boldsymbol{p}_2^T$$

$$- \frac{4\eta}{m} \left( (\mathbf{g}(t) - \mathbf{y}) \odot \boldsymbol{m}_+ \right)^T \left( \mathbf{g}(t) \odot \boldsymbol{m}_+ \right) \boldsymbol{p}_1 \boldsymbol{p}_1^T$$

$$- \frac{4\eta}{m} \left( (\mathbf{g}(t) - \mathbf{y}) \odot \boldsymbol{m}_- \right)^T \left( \mathbf{g}(t) \odot \boldsymbol{m}_- \right) \boldsymbol{p}_2 \boldsymbol{p}_2^T.$$

Now we derive the evolution of $\mathbf{g}(t) - \mathbf{y}$. Suppose $i \in \mathcal{S}_+$. Then we have

$$
\begin{aligned}
g_i(t+1) &= g_i(0) + \langle \mathbf{u}(t+1) - \mathbf{u}(0), \nabla_{\mathbf{u}} g_i(0) \rangle + \langle \mathbf{v}(t+1) - \mathbf{v}(0), \nabla_{\mathbf{v}} g_i(0) \rangle \\
&\quad + \left\langle \mathbf{u}(t+1) - \mathbf{u}(0), \frac{\partial^2 g_i(0)}{\partial \mathbf{u} \partial \mathbf{v}} (\mathbf{v}(t+1) - \mathbf{v}(0)) \right\rangle \\
&= g_i(t) - \eta \left\langle \sum_{k=1}^n (g_k(t) - y_k) \left( \nabla_{\mathbf{u}} g_k(0) + \frac{\partial^2 g_k(0)}{\partial \mathbf{u} \partial \mathbf{v}} (\mathbf{v}(t) - \mathbf{v}(0)) \right), \nabla_{\mathbf{u}} g_i(0) \right\rangle \\
&\quad - \eta \left\langle \sum_{k=1}^n (g_k(t) - y_k) \left( \nabla_{\mathbf{v}} g_k(0) + (\mathbf{u}(t) - \mathbf{u}(0))^T \frac{\partial^2 g_k(0)}{\partial \mathbf{u} \partial \mathbf{v}} \right), \nabla_{\mathbf{v}} g_i(0) \right\rangle \\
&\quad - \eta \left\langle \sum_{k=1}^n (g_k(t) - y_k) \left( \nabla_{\mathbf{u}} g_k(0) + \frac{\partial^2 g_k(0)}{\partial \mathbf{u} \partial \mathbf{v}} (\mathbf{v}(t) - \mathbf{v}(0)) \right), \frac{\partial^2 g_i(0)}{\partial \mathbf{u} \partial \mathbf{v}} (\mathbf{v}(t) - \mathbf{v}(0) \right\rangle \\
&\quad - \eta \left\langle \sum_{k=1}^n (g_k(t) - y_k) \left( \nabla_{\mathbf{v}} g_k(0) + (\mathbf{u}(t) - \mathbf{u}(0))^T \frac{\partial^2 g_k(0)}{\partial \mathbf{u} \partial \mathbf{v}} \right), (\mathbf{u}(t) - \mathbf{u}(0)^T \frac{\partial^2 g_i(0)}{\partial \mathbf{u} \partial \mathbf{v}} \right\rangle \\
&\quad + \eta^2 \left\langle \sum_{k=1}^n (g_k(t) - y_k) \left( \nabla_{\mathbf{u}} g_k(0) + \frac{\partial^2 g_k(0)}{\partial \mathbf{u} \partial \mathbf{v}} (\mathbf{v}(t) - \mathbf{v}(0)) \right), \right. \\
&\qquad\qquad \left. \frac{\partial^2 g_i(0)}{\partial \mathbf{u} \partial \mathbf{v}} \sum_{k=1}^n (g_k(t) - y_k) \left( \nabla_{\mathbf{v}} g_k(0) + (\mathbf{u}(t) - \mathbf{u}(0))^T \frac{\partial^2 g_k(0)}{\partial \mathbf{u} \partial \mathbf{v}} \right) \right\rangle \\
&= g_i(t) - \eta \sum_{k \in \mathcal{S}_+} (g_k(t) - y_k) K_{k,i}(t) + \frac{\eta^2}{m} \sum_{k \in \mathcal{S}_+} \sum_{j \in \mathcal{S}_+} (g_k(t) - y_k)(g_j(t) - y_j) g_j(t) x_k x_i.
\end{aligned}
$$

Similarly, for $i \in \mathcal{S}_-$, we have

$$
g_i(t+1) = g_i(t) - \eta \sum_{k \in \mathcal{S}_-} (g_k(t) - y_k) K_{k,i}(t) + \frac{\eta^2}{m} \sum_{k \in \mathcal{S}_- f} \sum_{j \in \mathcal{S}_-} (g_k(t) - y_k)(g_j(t) - y_j) g_j(t) x_k x_i.
$$

Combining the results together, we have

$$
\begin{aligned}
\mathbf{g}(t+1) - \mathbf{y} &= \left( I - \eta K(t) + \frac{\eta^2}{m} ((\mathbf{g}(t) - \mathbf{y}) \odot \boldsymbol{m}_+)^T (\mathbf{g}(t) \odot \boldsymbol{m}_+) \boldsymbol{p}_1 \boldsymbol{p}_1^T \right. \\
&\quad \left. + \frac{\eta^2}{m} ((\mathbf{g}(t) - \mathbf{y}) \odot \boldsymbol{m}_-)^T (\mathbf{g}(t) \odot \boldsymbol{m}_-) \boldsymbol{p}_2 \boldsymbol{p}_2^T \right) (\mathbf{g}(t) - \mathbf{y}).
\end{aligned}
$$

## C  OPTIMIZATION WITH SUB-CRITICAL LEARNING RATES

**Theorem 4.** *Consider training the NQM Eq. ([10](#)), with squared loss on a single training example by GD. With a sub-critical learning rate $\eta \in [\epsilon, \frac{2-\epsilon}{\lambda_0}]$ with $\epsilon = \Theta\left( \frac{\log m}{\sqrt{m}} \right)$, the loss decreases exponentially with*

$$
\mathcal{L}(t+1) \leq \left( 1 - \delta + O\left( \frac{1}{m\delta} \right) \right)^2 \mathcal{L}(t) = (1 - \delta + o(\delta))^2 \mathcal{L}(t),
$$

*where $\delta = \min(\eta \lambda_0, 2 - \eta \lambda_0)$.*

*Furthermore, $\sup_t |\lambda(t) - \lambda(0)| = O\left( \frac{1}{m\delta} \right)$.*

We use the following transformation of the variables to simplify notations.

$$
u(t) = \frac{\|\boldsymbol{x}\|^2}{md} \eta^2 (g(t) - y)^2, \quad w(t) = \frac{\|\boldsymbol{x}\|^2}{md} \eta^2 (g(t) - y) y, \quad v(t) = \eta \lambda(t).
$$

Then the Eq. (11) and Eq. (12) are reduced to

$$u(t + 1) = (1 - v(t) + u(t) + w(t))^2 u(t) := \kappa(t)u(t)$$
$$v(t + 1) = v(t) - u(t)(4 - v(t)) - 4w(t).$$

At initialization, since $|g(0)| = O(1)$, we have $u(0) \leq C_u/m$ for some constant $C_u > 0$. As $|w(t)| = \frac{C\sqrt{u(t)}}{\sqrt{m}}$. where $C := \frac{\eta\|\boldsymbol{x}\|\|y\|}{\sqrt{d}} > 0$, we have $|w(0)| \leq C_u C/m^{3/2}$. Note that by definition, for all $t \geq 0$, $u(t) \geq 0$ we have $v(t) \geq 0$ since $\lambda(t)$ is the tangent kernel for a single training example. From the definition of $\delta$, we can infer that $\delta < 1$.

In the following, we will show that if $v(0) \in [\epsilon, 2 - \epsilon]$ with $\epsilon = \Theta\left(\frac{\log m}{\sqrt{m}}\right)$, then there exist constant $C_\kappa, C_v > 0$ such that for all $t \geq 0$,

$$\kappa(t) \leq \left(1 - \delta + \frac{C_\kappa}{m\delta}\right)^2 < 1, |v(t) - v(0)| \leq \frac{C_v}{m\delta},$$

if $C_\kappa \geq 9C_u + (C_u + C_u C)\delta$ and $m$ satisfies

$$m > \max\left\{\frac{12C_\kappa}{\delta^2}, \frac{\sqrt{6}C_\kappa}{\delta^{3/2}}, C^2\right\}.$$

Given the condition on $m$, we have $(1 - \delta + \frac{C_\kappa}{m\delta})^2 < 1$.

We will prove the result by induction. When $t = 0$, we have

$$\kappa(0) = (1 - v(0) + u(0) + w(0))^2 < \left(1 - \delta + \frac{C_u}{m} + \frac{C_u C}{m^{3/2}}\right)^2 < \left(1 - \delta + \frac{C_\kappa}{m\delta}\right)^2,$$

where we use the assumption $C_\kappa \geq (C_u + C_u C)\delta$.

Therefore the result holds at $t = 0$.

Suppose when $t = T$ the results hold. Then at $t = T + 1$, by the inductive hypothesis that $u(t)$ decreases exponentially with $\kappa(t) < \left(1 - \delta + \frac{C_\kappa}{m\delta}\right)^2$, we can bound the change of $v(T + 1)$ from $v(0)$:

$$|v(T + 1) - v(0)| = \sum_{t=0}^{T} |u(t)(4 - v(t)) + 4w(t)|$$
$$\leq \sum_{t=0}^{T} |u(t)||4 - v(t)| + \sum_{t=0}^{T} 4|w(t)|$$
$$\leq \max_{t \in [0,T]} |v(t) - 4| \cdot \frac{u(0) - u(T)\max_{t \in [0,T]} \kappa(t)}{1 - \max_{t \in [0,T]} \kappa(t)} + \frac{w(0) - w(T)\max_{t \in [0,T]} \sqrt{\kappa(t)}}{1 - \max_{t \in [0,T]} \sqrt{\kappa(t)}}$$
$$\leq 4 \cdot \frac{u(0)}{1 - \max_{t \in [0,T]} \kappa(t)} + \frac{w(0)}{1 - \max_{t \in [0,T]} \sqrt{\kappa(t)}}$$
$$\leq 4 \cdot \frac{C_u/m}{\delta/2} + \frac{C_u C/m^{3/2}}{\delta/2}$$
$$\leq \frac{9C_u}{m\delta}.$$

For the summation of the "geometric sequence" i.e., $\{u(0), u(1), \cdots, u(T)\}$ where $u(0)$ and $u(T)$ have the determined order but the ratio has an upper bound, we use the maximum ratio, i.e., $\max \kappa(t)$ in the denominator to upper bound the summation.

For $1 - \max_{t \in [0,T]} \kappa(t)$, we use the bound that

$$
\begin{aligned}
1 - \left(1 - \delta + \frac{C_\kappa}{m\delta}\right)^2 &= 2\delta - \delta^2 - \frac{C_\kappa^2}{m^2\delta^2} - \frac{2C_\kappa}{m\delta} + \frac{2C_\kappa}{m} \\
&\geq \delta - \frac{\delta}{6} - \frac{\delta}{6} - \frac{\delta}{6} \\
&\geq \frac{\delta}{2},
\end{aligned}
$$

where we use the assumption on $m$.

Furthermore,

$$
\begin{aligned}
\kappa(T+1) &= (1 - v(T+1) + u(T+1) + w(T+1))^2 \\
&= (1 - v(0) + v(0) - v(T+1) + u(T+1) + w(T+1))^2 \\
&\leq \left(1 - \delta + \frac{9C_u}{m\delta} + \frac{C_u}{m} + \frac{C_u C}{m^{3/2}}\right)^2 \\
&\leq \left(1 - \delta + \frac{C_\kappa}{m\delta}\right)^2.
\end{aligned}
$$

Here we use the assumption $C_\kappa \geq 9C_u + (C_u + C_u C)\delta$.

Therefore, we finish the inductive step hence finishing the proof.

## D   PROOF OF LEMMA 1

We present the formal statement of Lemma 1:

**Lemma 1.**  *Consider constants $C_u, C_u', C_v, C_\kappa, C_\epsilon > 0$ which satisfies $C_\kappa \geq 28C_u' + C_u'\delta + C\sqrt{C_u'}$ where $C = \eta\|\boldsymbol{x}\|\|y\|/\sqrt{d}$, and $C_v \geq 28C_u'$. If $m$ satisfies*

$$
m \geq \max\left\{\frac{C_\kappa \delta}{C_u(C+1)}, \left(\frac{2C_u + 4C\sqrt{C_u}}{C_\kappa C_\epsilon}\right)^2, \frac{576C^2}{C_u'\delta^2}, \exp(C_v\delta), \exp(4C_\kappa)\right\},
$$

*then with high probability over random initialization of the weights, the following holds: for $T > 0$ such that $\sup_{t \in [0,T]} u(t) \leq \frac{C_u'\delta^2}{\log m}$, $u(t)$ increases exponentially with ratio $\inf_{t \in [0,T]} \kappa(t) \geq \left(1 + \delta - \frac{C_\kappa \delta}{\log m}\right)^2 > 1$ and $\sup_{t \in [0,T]} |v(t) - v(0)| \leq \frac{C_v\delta}{\log m}$.*

*Proof.*  Due to the random initialization of the weights, we have with probability, there exists constant $C_u > 0$ such that $|u(0)| \leq C_u/m$. As $|w(t)| = \frac{C\sqrt{u(t)}}{\sqrt{m}}$, where $C := \frac{\eta\|\boldsymbol{x}\|\|y\|}{\sqrt{d}} > 0$, we have $|w(0)| \leq \frac{C\sqrt{C_u}}{m^{3/2}}$.

We prove the results by induction.

Recall that $\delta := \eta\lambda_0 - 2 \in [\epsilon, 2 - \epsilon]$ where $\epsilon \in [C_\epsilon \log m/\sqrt{m}, C_\epsilon' \log m/\sqrt{m}]$ for some constant $0 < C_\epsilon < C_\epsilon'$.

When $t = 0$, as $v(0) = \eta\lambda_0 = \delta + 2$, we have

$$
\begin{aligned}
\kappa(0) &= (1 - v(0) + u(0) + w(0))^2 \\
&= (1 - (\delta + 2) + u(0) + w(0))^2 \\
&= (1 + \delta - u(0) - w(0))^2.
\end{aligned}
$$

Based on the condition on $m$ that $m \geq \frac{C_\kappa \delta}{C_u(C+1)}$, we have,

$$(1 + \delta - u(0) - w(0))^2 \geq \left(1 + \delta - \frac{C_u}{m} - \frac{C_u C}{m^{3/2}}\right)^2$$

$$\geq \left(1 + \delta - \frac{C_u}{m} - \frac{C_u C}{m}\right)^2$$

$$\geq \left(1 + \delta - \frac{C_\kappa \delta}{\log m}\right)^2.$$

And by the condition $m \geq \left(\frac{2C_u + 4c\sqrt{C_u}}{C_\kappa C_\epsilon}\right)^2$, we get

$$|v(1) - v(0)| \leq |u(0)||2 - \delta| + 4|w(0)| \leq 2C_u/m + \frac{4C\sqrt{C_u}}{m^{3/2}} \leq C_\kappa \delta / \log m.$$

Therefore the results hold at $t = 0$.

Suppose when $t = T'$ the results hold. Then at $t = T' + 1$, by the inductive hypothesis that $u(t)$ increases exponentially with a rate at least $\left(1 + \delta - \frac{C_\kappa \delta}{\log m}\right)^2$ from $u(0) \leq C_u/m$ to $u(T') \leq \frac{C'_u \delta^2}{\log m}$, we can bound the change of $v(t)$:

$$|v(T' + 1) - v(0)| = \left|\sum_{t=1}^{T'} u(t)(v(t) - 4) + 4w(t)\right|$$

$$\leq \max_{t \in [0,T']} |v(t) - 4| \sum_{t=1}^{T'} u(t) + 4\sum_{t=1}^{T'} |w(t)|$$

$$\leq \max_{t \in [0,T']} |v(t) - 4| \frac{u(T') \min_{t \in [0,T']} \kappa(t)}{\min_{t \in [0,T']} \kappa(t) - 1} + 4\frac{|w(T')| \min_{t \in [0,T']} \sqrt{\kappa(t)}}{\min_{t \in [0,T']} \sqrt{\kappa(t)} - 1}$$

$$\leq \left(\max_{t \in [0,T']} |v(t) - v(0)| + |v(0) - 4|\right) \frac{\left(\frac{C'_u \delta^2}{\log m}\right) \cdot (1 + \delta)^2}{\left(1 + \delta - \left(\frac{C_\kappa \delta}{\log m}\right)\right)^2 - 1}$$

$$+ \frac{4\left(\frac{C\sqrt{C'_u}\delta}{\sqrt{m}\log m}\right) \cdot (1 + \delta)}{\left(1 + \delta - \left(\frac{C_\kappa \delta}{\log m}\right)\right) - 1}$$

$$\leq \left(2 - \delta + \frac{C_v \delta}{\log m}\right) \cdot \left(\frac{9C'_u \delta}{\log m}\right) + \frac{24C\sqrt{C'_u}}{\sqrt{m}\log m}$$

$$\leq \frac{28C'_u \delta}{\log m}. \tag{18}$$

Here are the techniques we used for the above inequalities: for the summation of the "geometric sequence" i.e., $\{u(0), u(1), \cdots, u(T')\}$ where $u(0)$ and $u(T')$ have the determined order but the ratio has a lower bound, we use the smallest ratio, i.e., $\inf \kappa(t)$ to upper bound the summation. Specifically, we apply the following inequality to bound the summation:

$$\sum_{t=1}^{T'} u(t) \leq \sum_{t=1}^{T'} \frac{u(T')}{\left(\min_{t \in [0,T']} \kappa(t)\right)^{t-1}} = u(T') \sum_{t=1}^{T'} \frac{1}{\left(\min_{t \in [0,T']} \kappa(t)\right)^{t-1}} \leq u(T') \frac{\min_{t \in [0,T']} \kappa(t)}{\min_{t \in [0,T']} \kappa(t) - 1}.$$

Additionally, sine $m \geq \exp(4C_\kappa \delta)$, we used the inequality

$$\left(1 + \delta - \frac{C_\kappa \delta}{\log m}\right)^2 - 1 = \left(1 - \frac{C_\kappa \delta}{\log m}\right)^2 \delta^2 + 2\left(1 - \frac{C_\kappa \delta}{\log m}\right)\delta$$

$$\geq 2\left(1 - \frac{C_\kappa \delta}{\log m}\right)\delta \geq \delta,$$

and $\left(1 + \delta - \left(\frac{C_\kappa \delta}{\log m}\right)\right) - 1 \geq \frac{\delta}{2}$ to bound the denominator of the summation of the geometric sequence.

And we further used the inequality $0 < \delta < 2$ and $\frac{24C\sqrt{C'_u}}{\sqrt{m}\log m} \leq \frac{C'_u \delta}{\log m}$ by the condition on $m$ to get the final upper bound.

Consequently, by the assumption $C_v \geq 28C'_u\delta$, we have $|v(T'+1) - v(0)| \leq \frac{C_v \delta}{\log m}$.

Now we bound the ratio $\kappa(T'+1)$. By our assumption, $u(T'+1) \leq u(T) \leq \frac{C'_u \delta^2}{\log m}$, and we can similarly bound $|w(T'+1)| \leq \frac{C\sqrt{C'_u}\delta}{\sqrt{m}\log m}$ as $|w(T'+1)| = \frac{C\sqrt{u(T'+1)}}{\sqrt{m}}$.

And the rate $\kappa(T'+1)$ satisfies

$$\kappa(T'+1) = (1 - v(T'+1) + u(T'+1) + w(T'+1))^2$$
$$= (1 - v(0) + v(0) - v(T'+1) + u(T'+1) + w(T'+1))^2$$
$$= (1 + \delta + v(T'+1) - v(0) - u(T'+1) - w(T'+1))^2.$$

Note that $|v(T'+1) - v(0)| \leq \frac{28C'_u\delta}{\log m}$ by Eq. (18). By the assumption that $m \geq \exp(4C_\kappa)$ and $C_\kappa \geq 28C'_u + C'_u\delta + C\sqrt{C'_u}$, we have $\delta > |v(T'+1) - v(0)| + u(T'+1) + |w(T'+1)|$.

Consequently, we can get

$$\kappa(T'+1) = (1 + \delta + v(T'+1) - v(0) - u(T'+1) - w(T'+1))^2$$
$$\geq (1 + \delta - |v(T'+1) - v(0)| - u(T'+1) - |w(T'+1)|)^2$$
$$\geq \left(1 + \delta - \frac{28C'_u\delta}{\log m} - \frac{C'_u\delta^2}{\log m} - \frac{C\sqrt{C'_u}\delta}{\sqrt{m}\log m}\right)^2$$
$$\geq \left(1 + \delta - \frac{C_\kappa \delta}{\log m}\right)^2.$$

Since $m \geq \exp(4C_\kappa)$, we have $\left(1 + \delta - \frac{C_\kappa \delta}{\log m}\right)^2 \geq \left(1 + \frac{3}{4}\delta\right)^2 > 1$.

Then we finish the inductive step hence finishing the proof.

$\square$

# E    PROOF OF LEMMA 2

A formal statement of Lemma 2 is as follows:

**Lemma 2:**   Under the condition of Lemma 1, if we further assume that $m$ satisfies

$$m > \max\left\{\exp(2C_v\delta), \frac{256C^2}{(C_\epsilon - C'_v)^2{C'_u}^2}, \exp\left(5(C'_u + 4C\sqrt{C'_u})\right), \exp\left(\frac{C'_u(C_\epsilon - 2C'_v) - 8C\sqrt{C'_u}}{20CC'_u}\right)\right\},$$

where $C'_v := 18C'_u + 2C_v$, and $C_v \geq 4C\sqrt{C'_u}$, $C_\epsilon > 2C'_v$, then with high probability over random initialization of the weights, the following holds: there exists $T^* > 0$ such that $u(T^*) = O\left(\frac{1}{m}\right)$.

*Proof.* The main idea of the proof is the following: as $u(t)$ increases, $v(t)$ decreases since $u(t)(4 - v(t)) \gg w(t) = \Theta(\sqrt{u(t)}/\sqrt{m})$ in Eq. (14) and $u(t)(4 - v(t)) < 0$. Furthermore, the increase of $u(t)$ speeds up the decrease of $v(t)$. However, $v(t)$ cannot decrease infinitely as $v(t) \geq 0$ by definition. Therefore, $u(t)$ has to stop increasing at some point and decrease to a small value.

We first show that by the choice of the learning rate that $4 - v(0) \geq \epsilon$ where $\epsilon = \Theta\left(\frac{\log m}{\sqrt{m}}\right)$, we will have $4 - v(t) > 0$ for all $t$ in the increasing phase. Recall that $\delta := \eta\lambda_0 - 2$.

**Proposition 1.** *Under the condition in Lemma 1, if we further assume $m > \exp\left(\frac{48C\sqrt{C_u'}}{C_\epsilon}\right)^{2/3}$, then for $T > 0$ such that $\sup_{t \in [0,T]} u(t) \leq \frac{C_u'\delta^2}{\log m}$, we have $v(T) < 4 - \frac{C_\epsilon \log m}{2\sqrt{m}}$.*

See the proof in Appendix H.1

Given the constant $C_u'$ in Lemma 1, we define the end of the increasing phase by $T_1$, i.e.,

$$T_1 := \sup\left\{t : u(t) \leq \frac{C_u'\delta^2}{\log m}\right\}. \tag{19}$$

We further show that there exists $T_2 \geq T_1$ such that $v(T_2) \leq 3$.

Note that we indeed can show that there exists $T_2$ such that $v(T_2) < \overline{C}$ where $\overline{C} \in (2, 4)$ is a constant independent of $m$. Here for the simplicity of the presentation, we take $C$ as 3. Furthermore, we note that $T_1, T_2$ depends on $m$.

Before that, we present a useful result that controls the decrease of $v(t)$:

**Proposition 2.** *For $t$ such that $v(t) < 4$, if $u(t) > \frac{4C}{m(4-v(t))^2}$, then $v(t+1) < v(t)$.*

See the proof in Appendix H.2.

Now we are ready to show the existence of $T_2$ such that $v(T_2) \leq 3$.

**Proposition 3.** *Under the condition of Lemma 1, if we further assume that $m$ satisfies*

$$m > \max\left\{\exp\left(\frac{768^2C^2}{C_u'C_\epsilon^2}\right), \exp\left(2C_u' + C_\epsilon\right), \exp\left(\frac{48C\sqrt{C_u'}}{C_\epsilon}\right)^{2/3}, \frac{16C^2}{C_u'^2}\right\},$$

*and $C_u' \geq 4C^2$, there exists $T_2 \geq T_1$ such that $v(T_2) \leq 3$.*

See the proof in Appendix H.3

Since $v(T_2) < 3$ hence $4 - v(T_2) \geq 1$. Simply using Proposition 2, we get

**Proposition 4.** *$v(t)$ keeps decreasing after $T_2$ until $u(t) = O\left(\frac{1}{m}\right)$.*

By definition $v(t) = \eta\lambda(t)$ where $\lambda(t) \geq 0$, $v(t)$ will not keep decreasing for $t \to \infty$ hence there exists $T^*$ such that $u(T^*) = O\left(\frac{1}{m}\right)$. And it indicates that the loss will decrease to the order of $O(1)$.

## F  PROOF OF THEOREM 2

We compute the steady-state equilibria of Eq. (13) and (14). By letting $u(t+1) = u(t)$ and $v(t+1) = v(t)$, we have the steady-state equilibria $(u^*, v^*)$ satisfy one of the following:

(1) $u^* = 0$, $v^* \in \mathbb{R}$;

(2) $|1 - v^* + u^* + w^*| = 1$, $u^*(4 - v^*) + 4w^* = 0$.

As $w(t)^2 = \frac{C^2 u(t)}{m}$ where $C := \frac{\eta\|x\|\|y\|}{\sqrt{d}} > 0$, we write $w$ as a function of $u$ for simplicity, hence $w^* = w(u^*)$.

As the dynamics equations are non-linear, we analyze the local stability of the steady-state equilibria. We consider the Jacobian matrix of the dynamical systems:

$$J(u, v) = \begin{bmatrix} 2(1 - v + u + w)(1 + \frac{dw}{du})u + (1 - v + u + w)^2 & -2(1 - v + u + w)u \\ v - 4 - 4\frac{dw}{du} & 1 + u \end{bmatrix}.$$

We analyze the stability of two equilibria separately.

For Scenario (1), we evaluate $J(u, v)$ at the steady-state equilibrium $(u^*, v^*)$ then we get

$$J(u^*, v^*) = \begin{bmatrix} (1 - v^*)^2 & 0 \\ v^* - 4 - 4\frac{dw}{du} & 1 \end{bmatrix}.$$

We get the two eigenvalues of $J(u^*, v^*)$ are 1 and $(1 - v^*)^2$. We will show the Lyapunov stability of the equilibrium $(u^*, v^*)$. Specifically, we apply Theorem 1.2 in Bof et al. (2018). We find the domain

$$D = \{(u, v) : u \le C_1, |v - v^*| \le \min(|C_2 - v^*|, |2 - C_2 - v^*|\},$$

where $C_1 = \Theta(1/m)$ and $C_2 = \Theta(1/\sqrt{m})$, and the Lyapunov function $V(u, v) = u + (v - v^*)^2$. It is not hard to verify that $V$ is locally Lipschitz in D as $V$ is continuous in a compact domain. Furthermore, we can see that $(u^*, v^*)$ with $u^* = 0, v^* \in [\epsilon, 2 - \epsilon]$ where $\epsilon = \Theta(\log m/\sqrt{m})$ satisfies the condtions Eq. (3,4) in Theorem 1.2 in Bof et al. (2018). Therefore, $(u^*, v^*)$ with $u^* = 0$ and $v^* \in [\epsilon, 2 - \epsilon]$ is a stable equilibrium point.

For Scenario (2), we again evaluate $J(u, v)$ at the steady-state equilibrium $(u^*, v^*)$ then we get

$$J(u^*, v^*) = \begin{bmatrix} -2u^* + \frac{C\sqrt{u^*}}{\sqrt{m}} + 1 & 2u^* \\ -\frac{2C}{\sqrt{mu^*}} & 1 + u^* \end{bmatrix},$$

where we replace $v^*$ by $4 + 4w^*/u^*$ based on the second equality in Scenario (2). Note that $u^*(4 - v^*) > 0$ since $v < 4$ during the whole training process, therefore we have $w^* < 0$ to achieve the equilibrium.

We can compute the eigenvalue of $J(u^*, v^*)$ then we get

$$\lambda_J = 1 + \frac{C}{2\sqrt{m}}\sqrt{u^*} - \frac{u^*}{2} \pm \frac{1}{2}(u^*)^{1/4}\sqrt{16\frac{C}{\sqrt{m}} - \frac{C^2\sqrt{u^*}}{m} + 6\frac{Cu^*}{\sqrt{m}} - 9(u^*)^{3/2}}i.$$

Note that when Scenario (2) holds, there are only two possible cases

(2.1) $u^* = \Theta(1/m), |w^*| = \Theta(1/m)$ and $v^* = \Theta(1)$;

(2.2) $u^* = \Theta(1/m), |w^*| = \Theta(1/m)$ and $v^* = \Theta(1/m)$.

For (2.1), by the first equality $v^* = 2 - u^* + w^* \in (1, 2)$. Then plugging $v^*$ into the second equality yields $u^* \in \left(\frac{4}{3}\frac{C^2}{m}, 2\frac{C^2}{m}\right)$.

For (2.2), by the second equality that $u^*(4 - v^*) + 4w^* = 0$, we have $u^* = \frac{C^2}{m} + o(1/m)$.

By computing the modulo of $\lambda_J$, we have

$$|\lambda_J| = 1 + \frac{5C}{\sqrt{m}}\sqrt{u^*} - u^* + o\left(\frac{1}{m}\right).$$

Therefore, for both (2.1) and (2.2) we have $|\lambda_J| > 1$ which indicates $(u^*, v^*)$ is unstable.

$\square$

## G  OPTIMIZATION WITH $\eta > \eta_{\max}$

**Theorem 5.** *Consider training the NQM Eq. (10), with squared loss on a single training example by GD. If the learning rate satisfies $\eta \in \left[\frac{4+\epsilon}{\lambda_0}, \infty\right)$ with $\epsilon = \Theta\left(\frac{\log m}{\sqrt{m}}\right)$, then GD diverges.*

*Proof.* We similarly use the transformation transformation of the variables to simplify notations.

$$u(t) = \frac{\|\boldsymbol{x}\|^2}{md}\eta^2(g(t) - y)^2, \quad w(t) = \frac{\|\boldsymbol{x}\|^2}{md}\eta^2(g(t) - y)y, \quad v(t) = \eta\lambda(t).$$

Then the Eq. (11) and Eq. (12) are reduced to

$$u(t + 1) = (1 - v(t) + u(t) + w(t))^2 u(t) := \kappa(t)u(t)$$
$$v(t + 1) = v(t) - u(t)(4 - v(t)) - 4w(t).$$

We similarly consider the interval $[0, T]$ such that $\sup_{t \in [0,T]} u(t) = O\left(\frac{1}{\log m}\right)$. By Lemma 1, in $[0, T]$, $u(t)$ increases exponentially with a rate $\sup_{t \in [0,T]} \kappa(t) > 9$. We assume $|w(t)| > |u(t)(4 - v(t))|$ for all $t \in [0, T]$, which is the worst case as $v(t)$ will increase the least. By Lemma 1, we have $\sum_{t=0}^{T} |w(t)| = O\left(\frac{1}{\sqrt{m \log m}}\right)$, which is less than $\epsilon$. Therefore, we have $v(T) > 4$.

Then at the end of the increasing phase, we have $|u(T_1)(4 - v(T_1))| = \Omega(1/\sqrt{m})$ is of a greater order than $|w(T_1)| = O(1/\sqrt{m \log m})$, hence $v(t)$ will increase at $T_1$. Note that $\kappa(T_1) = (1 - 4 + o(1))^2 = 9 + o(1)$, hence $u((t)$ also increases at $T_1$.

It is not hard to see that $v(t)$ will keep increasing unless $u(t)$ decreases to a smaller order. Specifically, if $|u(t)(4 - v(t))| = |4w(t)|$, it requires $u(t)$ to be of the order at least $O(1/\sqrt{\log m})$ (by letting $\epsilon u(t) = \Theta(w(t)) = \Theta(\sqrt{u(t)/m})$), which will not happen as $\kappa(t) = (1 - v(t) + o(1))^2 > 1$ and it contradicts the decrease of $u(t)$.

Therefore, both $u(t)$ and $v(t)$ keep increasing which leads to the divergence of GD.

$\square$

# H    PROOF OF PROPOSITIONS

## H.1    PROOF OF PROPOSITION 1

*Proof.* Note that $4 - v(0) = 2 - \delta \geq \frac{C_\epsilon \log m}{\sqrt{m}}$ by definition, where $C_\epsilon > 0$ is a constant. To show $4 - v(T) > \frac{C_\epsilon \log m}{2\sqrt{m}}$, a sufficient condition is $v(T) - v(0) < \frac{C_\epsilon \log m}{2\sqrt{m}}$.

Specifically, we will prove for $T > 0$ such that $\sup_{t \in [0,T]} u(t) \leq \frac{C'_u \delta^2}{\log m}$, the following holds:

$$v(T) - v(0) < 4 \sum_{t=0}^{T} |w(t)| \leq \frac{24C\sqrt{C'_u}}{\sqrt{m \log m}},$$

where $C, C'_u$ are the same constants defined in Lemma 1. Then by the condition that $m > \exp\left(\left(\frac{48C\sqrt{C'_u}}{C_\epsilon}\right)^{2/3}\right)$, we have $v(T) - v(0) < \frac{C_\epsilon \log m}{2\sqrt{m}}$.

We will prove the result by induction.

When $T = 0$, the result holds trivially.

Suppose $T = T'$ the result holds. When $T = T' + 1$, since $v(T') - v(0) < 0$, we have $v(T') < 4$. Therefore, by the update equation of $v(t)$ Eq. (14), we have

$$v(T' + 1) = v(T') - u(T')(4 - v(T')) - 4w(T')$$
$$\leq v(T') - 4w(T')$$
$$\leq v(T') + 4|w(T')|.$$

Then $v(T' + 1) - v(0) = v(T' + 1) - v(T') + v(T') - v(0) \leq \sum_{t=0}^{T'+1} |w(t)|$.

By Lemma 1, we have $\sum_{t=0}^{T'+1} |w(t)| \leq \frac{24C\sqrt{C'_u}}{\sqrt{m \log m}}$. Indeed, this inequality holds for any $T' + 1$ such that $\sup_{t \in [0, T'+1]} u(t) \leq \frac{C'_u \delta^2}{\log m}$.

Therefore, we finish the inductive step hence finish the proof.

$\square$

## H.2 PROOF OF PROPOSITION 2

*Proof.* A sufficient condition for $v(t)$ to decrease is

$$u(t)(4 - v(t)) > 4|w(t)| = \frac{4C\sqrt{u(t)}}{\sqrt{m}}.$$

If $u(t) > \frac{4C}{m(4-v(t))^2}$, then the above condition is satisfied. $\square$

## H.3 PROOF OF PROPOSITION 3

*Proof.* Note that for $t \in [0, T_1]$, the change of $v(t)$ satisfies $\sup_t |v(t) - v(0)| \leq \frac{C_v \delta}{\log m}$ by Lemma 1.

For $\delta < 1 - \frac{C_v \delta}{\log m}$, i.e., $v(0) < 3 - \frac{C_v \delta}{\log m}$, we have $v(T_1) < v(0) + |v(T_1) - v(0)| = 2 + \delta + |v(0) - v(T_1)| < 3$. Therefore, the existence of $T_2$ can be guaranteed by simply letting $T_2 = T_1$.

For $\delta \geq 1 - \frac{C_v \delta}{\log m}$, i.e., $v(0) \geq 3 - \frac{C_v \delta}{\log m}$, we will show there exists $T_2 \geq T_1$ which depends on $m$ such that $v(T_2) < 3$.

We prove the existence of $T_2$ by contradiction. Suppose that for all $t \geq T_1 + 1$ we have $v(t) \geq 3$.

For the simple case that if all $u(t) > \frac{4C}{m(4-v(t))^2}$, then by Proposition 2, $v(t)$ keeps decreasing which will ultimately lead to $v(t) < 3$.

Suppose there is an iteration $t \geq T_1 + 1$ such that $u(t) \leq \frac{4C}{m(4-v(t))^2}$. The following Proposition guarantees that $v(t)$ will decrease to a smaller value after $t$ once such $t$ occurs. Therefore, we can find $T_2$.

**Proposition 5.** *Under the condition of Lemma 1, suppose $m$ further satisfies*

$$m > \max\left\{ \exp\left(\frac{768^2 C^2}{C'_u C_\epsilon^2}\right), \frac{16C^2}{C'^2_u}, \exp\left(2C'_u + C_\epsilon\right) \right\},$$

*where $C'_u \geq 4C^2$.*

*Then if there is $T \geq 0$ such that $u(T) \leq \frac{4C}{m(4-v(T))^2}$ and $v(T) > 3$, we have $v(T'+2) < v(T)$ and $u(T'+1) > \frac{4C}{m(4-v(T'))^2}$, where $T'$ is the end of the increasing phase starting from $T$.*

See the proof in Appendix H.4.

$\square$

## H.4 PROOF OF PROPOSITION 5

*Proof.* Since $u(T) \leq \frac{C'_u}{\log m}$ by the assumption that $m > \frac{16C^2}{C'^2_u}$, the training dynamics falls into the increasing phase from $T$. We denote the end of the increasing phase starting from $T$ by $T'$, i.e.,

$$T' = \sup\left\{ t : u(t) \leq \frac{C'_u \delta^2}{\log m}, t \geq T \right\}.$$

We will prove the result by induction.

Suppose $T'$ is the end of the first increasing phase, i.e., $T' = T_1$. By proposition 1, $v(T_1) < 4 - \frac{C_\epsilon \log m}{2\sqrt{m}}$. And the magnitude of $u(T_1 + 1)$ can be lower bounded by

$$u(T_1 + 1) \geq \frac{C'_u}{4 \log m}, \tag{20}$$

where we use $\delta \geq \frac{1}{2}$ by the assumption on $m$ that and plug $\delta$ in $\frac{C_u' \delta^2}{\log m}$.

Note that when $m > \exp(28C_u')$, $\delta \geq \frac{1}{2}$ is necessary for $v(T_1) > 3$ as $|v(T_1) - v(0)| \leq \frac{28C_u' \delta}{\log m}$ by Inequality (18).

Furthermore, with the above bound on $u(T_1 + 1)$, we have

$$
\begin{aligned}
v(T_1 + 1) &= v(T_1) - u(T_1)(4 - v(T_1)) - 4w(T_1) \\
&\leq 4 - \frac{C_\epsilon \log m}{2\sqrt{m}} + \frac{C_u'}{4\log m} \frac{C_\epsilon \log m}{2\sqrt{m}} + 4|w(T_1)| \\
&\leq 4 - \frac{C_\epsilon \log m}{2\sqrt{m}} + \frac{C_u' C_\epsilon}{8\sqrt{m}} + \frac{2\sqrt{C_u'}}{\sqrt{m}\log m} \\
&\leq 4 - \frac{C_\epsilon \log m}{4\sqrt{m}}.
\end{aligned}
\tag{21}
$$

Consequently, when $C_u' \geq 4C^2$ and $m > \exp(2C_u' + C_\epsilon)$, we have

$$
\begin{aligned}
u(T_1 + 1) &= \kappa(T_1)u(T_1) = (1 - v(T_1) + u(T_1) + w(T_1))^2 u(T_1) \\
&\leq \left(1 - 4 + \frac{C_\epsilon \log m}{4\sqrt{m}} + \frac{C_u'}{4\log m} + \frac{C\sqrt{C_u'}}{2\sqrt{m}\log m}\right)^2 \frac{C_u'}{4\log m} \\
&\leq \frac{9C_u'}{4\log m}.
\end{aligned}
$$

Therefore, at $T_1 + 1$, we have

$$
\begin{aligned}
v(T_1 + 1) - v(T_1 + 2) &\geq u(T_1 + 1)(4 - v(T_1 + 1)) - \frac{4C\sqrt{u(T_1 + 1)}}{\sqrt{m}} \\
&\geq \frac{C_u'}{4\log m} \frac{C_\epsilon \log m}{4\sqrt{m}} - \frac{8C\sqrt{C_u'}}{\sqrt{m}\log m} \\
&= \frac{C_u' C_\epsilon}{16\sqrt{m}} - \frac{6C\sqrt{C_u'}}{\sqrt{m}\log m} \\
&\geq \frac{C_u' C_\epsilon}{32\sqrt{m}},
\end{aligned}
$$

where we use the assumption that $m > \exp\left(\frac{256C^2}{9C_u' C_\epsilon^2}\right)$.

Note that the increase is caused by the term $w(t)$ since for all $t \in [T, T_1 + 1]$ we have $u(t)(4 - v(t)) < 0$. Then we have the maximum increase during $[T, T_1 + 1]$ be bounded by

$$
v(T_1 + 1) - v(T) \leq \sum_{t=T}^{T_1 + 1} 4|w(t)| \leq \frac{24C\sqrt{C_u'}}{\sqrt{m}\log m},
$$

where we use Eq. (18) in the proof of Lemma 1.

By the assumption on $m$ that $m > \exp\left(\frac{768^2 C^2}{C_u' C_\epsilon^2}\right)$, we have $v(T_1 + 2) < v(T)$.

If there is $\widetilde{T} > 0$ such that $u(\widetilde{T}) \leq \frac{4C}{m(4 - v(\widetilde{T}))^2}$ while $v(\widetilde{T}) > 3$, there is another increasing phase. Since $v(\widetilde{T}) < v(T)$, we can apply the same analysis under the same condition to show $v(\widetilde{T}_1 + 2) < v(\widetilde{T})$, where $\widetilde{T}_1$ is the end of the increasing phase starting from $\widetilde{T}$. Therefore, we finish the inductive step hence finish the proof.

$\square$

## H.5 Proof of Proposition 7

**Restate Proposition 7:** *For any* $\mathbf{u}, \mathbf{v} \in \mathbb{R}^m$, $\mathrm{rank}(K) \leq 2$. *Furthermore,* $\boldsymbol{p}_1, \boldsymbol{p}_2$ *are eigenvectors of* $K$, *where* $p_{1,i} = x_i \mathbb{1}_{\{i \in \mathcal{S}_+\}}$, $p_{2,i} = x_i \mathbb{1}_{\{i \in \mathcal{S}_-\}}$, *for* $i \in [n]$.

*Proof.* By Definition 1,

$$K_{i,j} = \frac{1}{m} \sum_{k=1}^m (v_k^2 + u_k^2) x_i x_j \mathbb{1}_{\{u_k x_i \geq 0\}} \mathbb{1}_{\{u_k x_j \geq 0\}}, \quad i, j \in [n].$$

By definition of eigenvector, we can see

$$\sum_{j=1}^n K_{i,j} p_{1,j} = \frac{1}{m} \sum_{j=1}^n \sum_{k=1}^m (v_k^2 + u_k^2) x_i x_j^2 \mathbb{1}_{\{u_k x_i \geq 0\}} \mathbb{1}_{\{u_k x_j \geq 0\}} \mathbb{1}_{\{j \in \mathcal{S}_+\}}$$

$$= \sum_{j=1}^n x_j^2 \mathbb{1}_{\{j \in \mathcal{S}_+\}} \frac{1}{m} \sum_{k=1}^m (v_k^2 + u_k^2) x_i \mathbb{1}_{\{u_k x_i \geq 0\}} \mathbb{1}_{\{u_k x_j \geq 0\}}$$

$$= x_i \mathbb{1}_{\{x_i \in S_+\}} \sum_{j=1}^n x_j^2 \mathbb{1}_{\{j \in \mathcal{S}_+\}} \frac{1}{m} \sum_{k=1}^m (v_k^2 + u_k^2) \mathbb{1}_{\{u_k x_j \geq 0\}},$$

where we use the fact that if $x_i x_j < 0$, $K_{i,j} = 0$.

As $p_{1,i} = x_i \mathbb{1}_{\{x_i \in S_+\}}$ and $\sum_{j=1}^n x_j^2 \mathbb{1}_{\{j \in \mathcal{S}_+\}} \frac{1}{m} \sum_{k=1}^m (v_k^2 + u_k^2) \mathbb{1}_{\{u_k x_j \geq 0\}}$ does not depend on $i$, we can see $\boldsymbol{p}_1$ is an eigenvector of $K$ with corresponding eigenvalue $\lambda_1 = \sum_{j=1}^n x_j^2 \mathbb{1}_{\{j \in \mathcal{S}_+\}} \frac{1}{m} \sum_{k=1}^m (v_k^2 + u_k^2) \mathbb{1}_{\{u_k x_j \geq 0\}}$.

The same analysis can be applied to show $\boldsymbol{p}_2$ is another eigenvector of $K$ with corresponding $\lambda_2 = \sum_{j=1}^n x_j^2 \mathbb{1}_{\{j \in \mathcal{S}_-\}} \frac{1}{m} \sum_{k=1}^m (v_k^2 + u_k^2) \mathbb{1}_{\{u_k x_j \geq 0\}}$.

For the rank of $K$, it is not hard to verify that $K = \lambda_1 \boldsymbol{p}_1 \boldsymbol{p}_1^T + \lambda_2 \boldsymbol{p}_2 \boldsymbol{p}_2^T$ hence the rank of $K$ is at most 2. $\qquad\square$

## I  Scale of the tangent kernel for single training example

**Proposition 6** (Scale of tangent kernel)**.** *For any* $\delta \in (0, 1)$, *if* $m \geq c' \log(4/\delta)$ *where* $c'$ *is an absolute constant, with probability at least* $1 - \delta$, $\|\boldsymbol{x}\|^2/(2d) \leq \lambda(0) \leq 3\|\boldsymbol{x}\|^2/(2d)$.

*Proof.* Note that when $t = 0$,

$$\lambda(0) = \frac{1}{md} \sum_{i=1}^m \left( \mathbf{u}_{0,i}^T \boldsymbol{x} \mathbb{1}_{\left\{\mathbf{u}_{0,i}^T \boldsymbol{x} \geq 0\right\}} \right)^2 + \frac{1}{md} \sum_{i=1}^m (v_{0,i})^2 \|\boldsymbol{x}\|^2 \left( \mathbb{1}_{\left\{\mathbf{u}_{0,i}^T \boldsymbol{x} \geq 0\right\}} \right)^2.$$

According to NTK initialization, for each $i \in [m]$, $v_{0,i} \sim \mathcal{N}(0, 1)$ and $\mathbf{u}_{0,i} \sim \mathcal{N}(0, I)$. We consider the random variable

$$\zeta_i := \mathbf{u}_{0,i}^T \boldsymbol{x} \mathbb{1}_{\left\{\mathbf{u}_{0,i}^T \boldsymbol{x} \geq 0\right\}}, \quad \xi_i := v_{0,i} \mathbb{1}_{\left\{\mathbf{u}_{0,i}^T \boldsymbol{x} \geq 0\right\}}.$$

it is not hard to see that $\zeta_i$ and $\xi_i$ are sub-guassian since $\mathbf{u}_{0,i}^T \boldsymbol{x}$ and $v_{0,i}$ are sub-gaussian. Specifically, for any $t \geq 0$,

$$\mathbb{P}\{|\zeta_i| \geq t\} \leq \mathbb{P}\{|\mathbf{u}_{0,i}^T \boldsymbol{x}| \geq t\} \leq 2 \exp\left(-t^2/(2\|\boldsymbol{x}\|^2)\right),$$

$$\mathbb{P}\{|\xi_i| \geq t\} \leq \mathbb{P}\{|v_{0,i}| \geq t\} \leq 2 \exp\left(-t^2/2\right),$$

where the second inequality comes from the definition of sub-gaussian variables.

Since $\xi_i$ is sub-gaussian, by definition, $\xi^2$ is sub-exponential, and its sub-exponential norm is bounded:

$$\|\xi_i^2\|_{\psi_1} \leq \|\xi_i\|_{\psi_2}^2 \leq C,$$

where $C > 0$ is a absolute constant. Similarly we have $\|\zeta_i\|_{\psi_2}^2 \le C\|\boldsymbol{x}\|^2$.

By Bernstein's inequality, for every $t \ge 0$, we have

$$\mathbb{P}\left\{\left|\sum_{i=1}^m \xi_i^2 - \frac{m}{2}\right| \ge t\right\} \le 2\exp\left(-c\min\left(\frac{t^2}{\sum_{i=1}^m \|\xi_i^2\|_{\psi_1}^2}, \frac{t}{\max_i \|\xi_i^2\|_{\psi_1}}\right)\right),$$

where $c > 0$ is an absolute constant.

Letting $t = m/4$, we have with probability at least $1 - 2\exp(-m/c')$,

$$\frac{m}{4} \le \sum_{i=1}^m \xi_i^2 \le \frac{3m}{4},$$

where $c' = c/(4C)$.

Similarity, we have with probability at least $1 - 2\exp(-m/c')$,

$$\frac{m}{4}\|\boldsymbol{x}\|^2 \le \sum_{i=1}^m \zeta_i^2 \le \frac{3m}{4}\|\boldsymbol{x}\|^2.$$

As a result, using union bound, we have probability at least $1 - 4\exp(-m/c')$,

$$\frac{\|\boldsymbol{x}\|^2}{2d} \le \lambda(0) \le \frac{3\|\boldsymbol{x}\|^2}{2d}.$$

$\square$

## J    SCALE OF THE TANGENT KERNEL FOR MULTIPLE TRAINING EXAMPLES

*Proof.* As shown in Proposition 7, $\boldsymbol{p}_1$ and $\boldsymbol{p}_2$ are eigenvectors of $K$, hence we have two eigenvalues:

$$\lambda_1(0) = \frac{\boldsymbol{p}_1^T K(0)\boldsymbol{p}_1}{\|\boldsymbol{p}_1\|^2}, \quad \lambda_2(0) = \frac{\boldsymbol{p}_2^T K(0)\boldsymbol{p}_2}{\|\boldsymbol{p}_2\|^2}.$$

Take $\lambda_1(0)$ as an example:

$$\lambda_1(0)\|\boldsymbol{p}_1\|^2 = \sum_{i,j=1}^n x_i x_j \mathbb{1}_{\{x_i \ge 0\}} \mathbb{1}_{\{x_j \ge 0\}} \sum_{k=1}^m (u_{0,k}^2 + v_{0,k}^2) x_i x_j \mathbb{1}_{\{u_{0,k}x_i \ge 0\}} \mathbb{1}_{\{u_{0,k}x_j \ge 0\}}$$

$$= \sum_{k=1}^m (u_{0,k}^2 + v_{0,k}^2) \left(\mathbb{1}_{\{u_{0,k} \ge 0\}}\right)^2 \sum_{i,j=1}^n x_i^2 x_j^2 \mathbb{1}_{\{x_i \ge 0\}} \mathbb{1}_{\{x_j \ge 0\}}.$$

Similar to the proof of Proposition 6, we consider $\xi_k := v_{0,k}\mathbb{1}_{\{u_{0,k} \ge 0\}}$ which is a sub-gaussian random variable. Hence $\xi_k^2$ is sub-exponential so that $\|\xi_k^2\|_{\psi_1} \le C$ where $C > 0$ is an absolute constant. By Bernstein's inequality, for every $t \ge 0$, we have

$$\mathbb{P}\left\{\left|\sum_{i=1}^m \xi_i^2 - \frac{m}{2}\right| \ge t\right\} \le 2\exp\left(-c\min\left(\frac{t^2}{\sum_{i=1}^m \|\xi_i^2\|_{\psi_1}^2}, \frac{t}{\max_i \|\xi_i^2\|_{\psi_1}}\right)\right),$$

where $c > 0$ is an absolute constant.

Letting $t = m/4$, we have with probability at least $1 - 2\exp(-m/c')$,

$$\frac{m}{4} \le \sum_{i=1}^m \xi_i^2 \le \frac{3m}{4},$$

where $c' = c/(4C)$.

The same analysis applies to $\zeta_k := u_{0,k}\mathbb{1}_{\{u_{0,k}\geq 0\}}$ as well and we have with probability at least $1 - 2\exp\left(-m/c'\right)$,

$$\frac{m}{4} \leq \sum_{i=1}^{m} \zeta_i^2 \leq \frac{3m}{4}.$$

As a result, we have probability at least $1 - 4\exp\left(-m/c'\right)$,

$$\lambda_1(0)\|\boldsymbol{p}_1\|^2 = \frac{1}{m}\sum_{i=k}^{m}(u_{0,k}^2 + v_{0,k}^2)\left(\mathbb{1}_{\{u_k(0)\geq 0\}}\right)^2 \sum_{i,j=1}^{n} x_i^2 x_j^2 \mathbb{1}_{\{x_i\geq 0\}}\mathbb{1}_{\{x_j\geq 0\}}$$

$$\in \left[\frac{1}{2}\sum_{i,j=1}^{n} x_i^2 x_j^2 \mathbb{1}_{\{x_i\geq 0\}}\mathbb{1}_{\{x_j\geq 0\}}, \frac{3}{2}\sum_{i,j=1}^{n} x_i^2 x_j^2 \mathbb{1}_{\{x_i\geq 0\}}\mathbb{1}_{\{x_j\geq 0\}}\right].$$

Applying the same analysis to $\lambda_2(0)$, we have with probability $1 - 4\exp\left(-m/c'\right)$,

$$\lambda_2(0)\|\boldsymbol{p}_2\|^2 = \frac{1}{m}\sum_{i=k}^{m}(u_{0,k}^2 + v_{0,k}^2)\left(\mathbb{1}_{\{u_k(0)\leq 0\}}\right)^2 \sum_{i,j=1}^{n} x_i^2 x_j^2 \mathbb{1}_{\{x_i\leq 0\}}\mathbb{1}_{\{x_j\leq 0\}}$$

$$\in \left[\frac{1}{2}\sum_{i,j=1}^{n} x_i^2 x_j^2 \mathbb{1}_{\{x_i\leq 0\}}\mathbb{1}_{\{x_j\leq 0\}}, \frac{3}{2}\sum_{i,j=1}^{n} x_i^2 x_j^2 \mathbb{1}_{\{x_i\leq 0\}}\mathbb{1}_{\{x_j\leq 0\}}\right].$$

The largest eigenvalue is $\max\{\lambda_1(0), \lambda_2(0)\}$. Combining the results together, we have with probability at least $1 - 4\exp\left(-m/c'\right)$,

$$\frac{1}{2}M \leq \|K(0)\| \leq \frac{3}{2}M,$$

where $M = \max\left\{\frac{\sum_{i,j=1}^{n} x_i^2 x_j^2 \mathbb{1}\{x_i\geq 0\}\mathbb{1}\{x_j\geq 0\}}{\sum_{i=1}^{n} x_i^2 \mathbb{1}\{x_i\geq 0\}}, \frac{\sum_{i,j=1}^{n} x_i^2 x_j^2 \mathbb{1}\{x_i\leq 0\}\mathbb{1}\{x_j\leq 0\}}{\sum_{i=1}^{n} x_i^2 \mathbb{1}\{x_i\leq 0\}}\right\}.$ $\qquad\square$

## K  ANALYSIS ON OPTIMIZATION DYNAMICS FOR MULTIPLE TRAINING EXAMPLES

In this section, we discuss the optimization dynamics for multiple training examples. We will see that by confining the dynamics into each eigendirection of the tangent kernel, the training dynamics is similar to that for a single training example.

Since $x_i$ is a scalar for all $i \in [n]$, with the homogeneity of ReLU activation function, we can compute the exact eigenvectors of $K(t)$ for all $t \geq 0$. To that end, we group the data into two sets $\mathcal{S}_+$ and $\mathcal{S}_-$ according to their sign:

$$\mathcal{S}_+ := \{i : x_i \geq 0, i \in [n]\}, \quad \mathcal{S}_- := \{i : x_i < 0, i \in [n]\}.$$

Now we have the proposition for the tangent kernel $K$(the proof is deferred to Appendix H.5):

**Proposition 7** (Eigenvectors and low rank structure of $K$). *For any* $\mathbf{u}, \mathbf{v} \in \mathbb{R}^m$, $\text{rank}(K) \leq 2$. *Furthermore,* $\boldsymbol{p}_1$, $\boldsymbol{p}_2$ *are eigenvectors of $K$, where* $p_{1,i} = x_i \mathbb{1}_{\{i\in\mathcal{S}_+\}}$, $p_{2,i} = x_i \mathbb{1}_{\{i\in\mathcal{S}_-\}}$, *for* $i \in [n]$.

Note that when all $x_i$ are of the same sign, $\text{rank}(K) = 1$ and $K$ only has one eigenvector (either $\boldsymbol{p}_1$ or $\boldsymbol{p}_2$ depending on the sign). It is in fact a simpler setting since we only need to consider one direction, whose analysis is covered by the one for $\text{rank}(K) = 2$. Therefore, in the following we will assume $\text{rank}(K) = 2$. We denote two eigenvalues of $K(t)$ by $\lambda_1(t)$ and $\lambda_2(t)$ corresponding to $\boldsymbol{p}_1$ and $\boldsymbol{p}_2$ respectively, i.e., $K(t)\boldsymbol{p}_1 = \lambda_1(t)\boldsymbol{p}_1$, $K(t)\boldsymbol{p}_2 = \lambda_2(t)\boldsymbol{p}_2$. Without loss of generality, we assume $\lambda_1(0) \geq \lambda_2(0)$.

By Eq. (5), the tangent kernel $K$ at step $t$ is defined as:

$$K_{i,j}(t) = \langle \nabla_{\mathbf{v}} g_i(t), \nabla_{\mathbf{v}} g_j(t) \rangle + \langle \nabla_{\mathbf{u}} g_i(t), \nabla_{\mathbf{u}} g_j(t) \rangle$$

$$= \frac{1}{m} \sum_{k=1}^{m} \left( (u_k(t))^2 + (v_k(t))^2 \right) x_i x_j \mathbb{1}_{\{u_k(0)x_i \geq 0\}} \mathbb{1}_{\{u_k(0)x_j \geq 0\}}, \quad \forall i, j \in [n].$$

Similar to single example case, the largest eigenvalue of of tangent kernel is bounded from 0:

**Proposition 8.** *For any $\delta \in (0, 1)$, if $m \geq c' \log(4/\delta)$ where $c'$ is an absolute constant, with probability at least $1 - \delta$, $M/2 \leq \lambda_{\max}(K(0)) \leq 3M/2$ where $M =$*
$\max \left\{ \frac{\sum_{i,j=1}^{n} x_i^2 x_j^2 \mathbb{1}_{\{x_i \geq 0\}} \mathbb{1}_{\{x_j \geq 0\}}}{\sum_{i=1}^{n} x_i^2 \mathbb{1}_{\{x_i \geq 0\}}}, \frac{\sum_{i,j=1}^{n} x_i^2 x_j^2 \mathbb{1}_{\{x_i \leq 0\}} \mathbb{1}_{\{x_j \leq 0\}}}{\sum_{i=1}^{n} x_i^2 \mathbb{1}_{\{x_i \leq 0\}}} \right\}$.

The proof can be found in Appendix J.

For the simplicity of notation, given $\boldsymbol{p}, \boldsymbol{m} \in \mathbb{R}^n$, we define the matrices $K_{\boldsymbol{p},\boldsymbol{m}}$ and $Q_{\boldsymbol{p},\boldsymbol{m}}$:

$$K_{\boldsymbol{p},\boldsymbol{m}}(t) := ((\mathbf{g}(t) - \mathbf{y}) \odot \boldsymbol{m})^T K(t) ((\mathbf{g}(t) - \mathbf{y}) \odot \boldsymbol{m}) \boldsymbol{p}\boldsymbol{p}^T,$$

$$Q_{\boldsymbol{p},\boldsymbol{m}}(t) := ((\mathbf{g}(t) - \mathbf{y}) \odot \boldsymbol{m})^T (\mathbf{g}(t) \odot \boldsymbol{m}) \boldsymbol{p}\boldsymbol{p}^T$$

It is not hard to see that for all $t$, $K_{\boldsymbol{p},\boldsymbol{m}}$ and $Q_{\boldsymbol{p},\boldsymbol{m}}$ are rank-1 matrices. Specially, $\boldsymbol{p}$ is the only eigenvector of $K_{\boldsymbol{p},\boldsymbol{m}}$ and $Q_{\boldsymbol{p},\boldsymbol{m}}$.

With the above notations, we can write the update equations for $\mathbf{g}(t) - \mathbf{y}$ and $K(t)$ during gradient descent with learning rate $\eta$:

**Dynamics equations.**

$$\mathbf{g}(t+1) - \mathbf{y} = \left( I - \eta K(t) + \underbrace{\frac{\eta^2}{m} \left( Q_{\boldsymbol{p}_1, \boldsymbol{m}_+}(t) + Q_{\boldsymbol{p}_2, \boldsymbol{m}_-}(t) \right)}_{R_{\mathbf{g}}(t)} \right) (\mathbf{g}(t) - \mathbf{y}), \qquad (22)$$

$$K(t+1) = K(t) + \underbrace{\frac{\eta^2}{m} \left( K_{\boldsymbol{p}_1, \boldsymbol{m}_+}(t) + K_{\boldsymbol{p}_2, \boldsymbol{m}_-}(t) \right) - \frac{4\eta}{m} \left( Q_{\boldsymbol{p}_1, \boldsymbol{m}_+}(t) + Q_{\boldsymbol{p}_2, \boldsymbol{m}_-}(t) \right)}_{R_K(t)}, \qquad (23)$$

where $\boldsymbol{m}_+, \boldsymbol{m}_- \in \mathbb{R}^n$ are mask vectors:

$$m_{+,i} = \mathbb{1}_{\{i \in \mathcal{S}_+\}}, \quad m_{-,i} = \mathbb{1}_{\{i \in \mathcal{S}_-\}}.$$

Now we are ready to discuss different three optimization dynamics for multiple training examples case, similar to the single training example case in the following.

**Monotonic convergence: sub-critical learning rates ($\eta < 2/\lambda_1(0)$).** We use the key observation that when $\|\mathbf{g}(t)\|$ is small, i.e., $O(1)$, and $\|K(t)\|$ is bounded, then $\|R_{\mathbf{g}}(t)\|$ and $\|R_K(t)\|$ are of the order $o(1)$. Then the dynamics equations approximately reduce to the ones of linear dynamics for multiple training examples:

$$\mathbf{g}(t+1) - \mathbf{y} = (I - \eta K(t) + o(1)) (\mathbf{g}(t) - \mathbf{y}),$$
$$K(t+1) = K(t) + o(1).$$

At initialization, $\|\mathbf{g}(0)\| = O(1)$ with high probability over random initialization. By the choice of the learning rate, we will have for all $t \geq 0$, $\|I - \eta K(t)\| < 2$, hence $\|\mathbf{g}(t) - \mathbf{y}\|$ decreases exponentially. The cumulative change on the norm of tangent kernel is $o(1)$ since $\|R_K(t)\| = O(1/m)$ and the loss decreases exponentially hence $\sum \|R_K(t)\| = O(1/m) \cdot \log O(1) = o(1)$.

**Catapult convergence: super-critical learning rates ($2/\lambda_1(0) < \eta < \min\{2/\lambda_2(0), 4/\lambda_1(0)\}$).** We summarize the catapult dynamics in the following:

**Restate Theorem 3** (Catapult dynamics on multiple training examples). Supposing Assumption 1 holds, consider training the NQM Eq. (10) with squared loss on multiple training examples by GD. Then,

1. with $\eta \in \left[\frac{2+\epsilon}{\lambda_1(0)}, \frac{2-\epsilon}{\lambda_2(0)}\right]$, the catapult only occurs in eigendirection $\boldsymbol{p}_1$: $\Pi_1 \mathcal{L}$ increases to the order of $\Omega\left(\frac{m(\eta-2/\lambda_1(0))^2}{\log m}\right)$ then decreases to $O(1)$;

2. with $\eta \in \left[\frac{2+\epsilon}{\lambda_2(0)}, \frac{4-\epsilon}{\lambda_1(0)}\right]$, the catapult occurs in both eigendirections $\boldsymbol{p}_1$ and $\boldsymbol{p}_2$: $\Pi_i \mathcal{L}$ for $i = 1, 2$ increases to the order of $\Omega\left(\frac{m(\eta-2/\lambda_i(0))^2}{\log m}\right)$ then decreases to $O(1)$,

where $\epsilon = \Theta\left(\frac{\log m}{\sqrt{m}}\right)$.

The proof can be found in Appendix L.

For the remaining eigendirections $\boldsymbol{p}_3, \cdots, \boldsymbol{p}_n$, i.e., the basis of the subspace orthogonal to $\boldsymbol{p}_1$ and $\boldsymbol{p}_2$, we can show that the loss projected to this subspace does not change during training in the following proposition. It follows from the fact that $K$, $R_{\mathbf{g}}(t)$ and $R_K(t)$ are orthogonal to $\boldsymbol{p}_i \boldsymbol{p}_i^T$ for $i = 3, \cdots, n$.

**Proposition 9.** $\forall t \geq 0$, $\Pi_i \mathcal{L}(t) = \Pi_i \mathcal{L}(0)$ for $i = 3, \cdots, n$.

Once the catapult finishes as the loss decreases to the order of $O(1)$, we generally have $\eta > 2/\lambda_1$ and $\eta > 2/\lambda_2$. Therefore the training dynamics fall into linear dynamics, and we can use the same analysis for sub-critical learning rates for the remaining training dynamics.

**Divergence:** $(\eta > \eta_{\max} = 4/\lambda_1(0))$. Similar to the increasing phase in the catapult convergence, initially $\|\mathbf{g}(t) - \mathbf{y}\|$ increases in direction $\boldsymbol{p}_1$ and $\boldsymbol{p}_2$ since linear dynamics dominate and the learning rate is chosen to be larger than $\eta_{\mathrm{crit}}$. Also, we approximately have $\eta > 4/\lambda_1(t)$ at the end of the increasing phase, by a similar analysis for the catapult convergence. We consider the evolution of $K(t)$ in the direction $\boldsymbol{p}_1$. Note that when $\|\mathbf{g}(t)\|$ increases to the order of $\Theta(\sqrt{m})$, $\mathbf{g}(t) \odot \boldsymbol{m}_+$ will be aligned with $\boldsymbol{p}_1$, hence with simple calculation, we approximately have

$$\boldsymbol{p}_1^T R_K(t) \boldsymbol{p}_1 \approx \frac{\|\mathbf{g}(t)\|^2 \|\boldsymbol{p}_1\|^2}{m} \eta(\lambda_1(t) - 4\eta) > 0.$$

Therefore, $\lambda_1(t)$ increases since $\boldsymbol{p}_1^T K(t+1) \boldsymbol{p}_1 = \boldsymbol{p}_1^T K(t) \boldsymbol{p}_1 + \boldsymbol{p}_1^T R_K(t) \boldsymbol{p}_1 > \boldsymbol{p}_1^T K(t) \boldsymbol{p}_1$. As a result, $\|I - \eta K(t) + R_{\mathbf{g}}(t)\|$ becomes even larger which makes $\|\mathbf{g}(t) - \mathbf{y}\|$ grows faster, and ultimately leads to divergence of the optimization.

## L  Proof of Theorem 3

As the tangent kernel $K$ has rank 2 by Proposition 7, the update of weight parameters $\mathbf{w}$ is in a subspace with dimension 2. Specifically,

$$\mathbf{w}(t+1) = \mathbf{w}(t) - \eta \frac{\partial \mathbf{g}}{\partial \mathbf{w}} \frac{\partial \mathcal{L}}{\partial \mathbf{g}}(t),$$

where $\partial \mathbf{g}/\partial \mathbf{w}$ has rank 2. Therefore, to understand the whole training dynamics, it is sufficient to analyze the dynamics of the loss in eigendirection $\boldsymbol{p}_1$ and $\boldsymbol{p}_2$.

We will analyze the dynamics of the loss $\mathcal{L}$ and the tangent kernel $K$ confined to $\boldsymbol{p}_1$ and $\boldsymbol{p}_2$. It turns out that the dynamics in each eigen direction is almost independent on the other hence can be reduced to the same training dynamics for a single training example.

We start with eigendirection $\boldsymbol{p}_1$. For dynamics equations Eq. (22) and (23), we consider the training dynamics confined to direction $\boldsymbol{p}_1$ and we have

$$\Pi_1 \mathcal{L}(t) = \left(1 - \eta \lambda_1(t) + \boldsymbol{p}_1^T R_{\mathbf{g}}(t) \boldsymbol{p}_1\right)^2 \Pi_1 \mathcal{L}(t) := \kappa_1(t) \Pi_1 \mathcal{L}(t),$$
$$\lambda_1(t+1) = \lambda_1(t) + \boldsymbol{p}_1^T R_K(t) \boldsymbol{p}_1,$$

where we use the notation $\Pi_1 \mathcal{L}(t) = \frac{1}{2} \langle \mathbf{g}(t) - \mathbf{y}, \boldsymbol{p}_1 \rangle^2$.

We further expand $\boldsymbol{p}_1^T R_{\mathbf{g}}(t) \boldsymbol{p}_1$ and $\boldsymbol{p}_1^T R_K(t) \boldsymbol{p}_1$ and we have

$$\boldsymbol{p}_1^T R_{\mathbf{g}}(t) \boldsymbol{p}_1 = \frac{2\eta^2}{m} \Pi_1 \mathcal{L}(t) + \frac{\eta^2}{m} \langle (\mathbf{g}(t) - \mathbf{y}) \odot \boldsymbol{m}_+, \mathbf{y} \odot \boldsymbol{m}_+ \rangle,$$

$$\boldsymbol{p}_1^T R_K(t) \boldsymbol{p}_1 = \frac{2\eta}{m} \Pi_1 \mathcal{L}(t)(\eta \lambda_1(t) - 4) - \frac{4\eta}{m} \langle (\mathbf{g}(t) - \mathbf{y}) \odot \boldsymbol{m}_+, \mathbf{y} \odot \boldsymbol{m}_+ \rangle.$$

Analogous to the transformation for Eq. (11) and (12) as we have done in the proof of Theorem 1, we let

$$u_1(t) = \frac{2\eta^2}{m} \Pi_1 \mathcal{L}(t), \quad w_1(t) = \frac{\eta^2}{m} \langle (\mathbf{g}(t) - \mathbf{y}) \odot \boldsymbol{m}_+, \mathbf{y} \odot \boldsymbol{m}_+ \rangle, \quad v_1(t) = \eta \lambda_1(t).$$

Then the dynamic equations can be written as:

$$u_1(t+1) = (1 - v_1(t) + u_1(t) + w_1(t))^2 u_1(t), \tag{24}$$
$$v_1(t+1) = v_1(t) - u_1(t)(4 - v_1(t)) - 4w_1(t). \tag{25}$$

Note that at initialization, $\|\mathbf{g}(t)\| = O(\sqrt{1})$ with high probability, hence we have $u_1(0) = O\left(\frac{1}{m}\right)$ and $w_1(0) = O\left(\frac{1}{m}\right)$ (we omit the factor $n$ as $n$ is a constant). Furthermore, $|w_1(t)| = \Theta\left(\frac{\sqrt{u_1(t)}}{\sqrt{m}}\right)$. Therefore, both the dynamic equations and the initial condition are exactly the same with the ones for a single training example (Eq. (13) and (14)). Then we can follow the same idea of the proof of Theorem 1 to show the catapult in eigendirection $\boldsymbol{p}_1$.

Similarly, when we consider the training dynamics confined to $\boldsymbol{p}_2$, we have

$$u_2(t+1) = (1 - v_2(t) + u_2(t) + w_2(t))^2 u_2(t), \tag{26}$$
$$v_2(t+1) = v_2(t) - u_2(t)(4 - v_2(t)) - 4w_2(t), \tag{27}$$

where

$$u_2(t) = \frac{2\eta^2}{m} \Pi_2 \mathcal{L}(t), \quad w_2(t) = \frac{\eta^2}{m} \langle (\mathbf{g}(t) - \mathbf{y}) \odot \boldsymbol{m}_-, \mathbf{y} \odot \boldsymbol{m}_- \rangle, \quad v_2(t) = \eta \lambda_2(t).$$

Then the same analysis with Theorem 1 can be used to show the catapult in direction $\boldsymbol{p}_2$.

Note that when $2/\lambda_2(0) > 4/\lambda_1(0)$, the learning rate is only allowed to be less than $4/\lambda_1(0)$ otherwise GD will diverge, therefore, there will be no catapult in direction $\boldsymbol{p}_2$.

# M    SPECIAL CASE OF QUADRATIC MODELS WHEN $\phi(\boldsymbol{x}) = 0$

In this section we will show under some special settings, the catapult phase phenomenon also happens and how two layer linear neural networks fit in our quadratic model.

We consider one training example $(\boldsymbol{x}, y)$ with label $y = 0$ and assume the initial tangent kernel $\lambda(0) = \Omega(1)$. Letting the feature vector $\phi(\boldsymbol{x}) = 0$, the quadratic model Eq.(3) becomes:

$$g(\mathbf{w}) = \frac{1}{2} \gamma \mathbf{w}^T \Sigma(\boldsymbol{x}) \mathbf{w}.$$

For this quadratic model, we have the following proposition:

**Proposition 10.** *With learning rate $\frac{2}{\lambda(0)} < \eta < \frac{4}{\lambda(0)}$, if $\Sigma(\boldsymbol{x})^2 = \|\boldsymbol{x}\|^2 \cdot I$, $g(\mathbf{w})$ exhibits catapult phase.*

*Proof.* With simple computation, we get

$$g(t+1) = \left(1 - \eta\lambda(t) + \gamma\eta^2 \|\boldsymbol{x}\|^2 (g(t))^2\right) g(t),$$
$$\lambda(t+1) = \lambda(t) - \gamma\|\boldsymbol{x}\|^2 (g(t))^2 (4 - \eta\lambda(t)).$$

We note that the evolution of $g$ and $\lambda$ is almost the same with Eq. (11) and Eq. (12) if we regard $\gamma = 1/m$. Hence we can apply the same analysis to show the catapult phase phenomenon.    $\square$

It is worth pointing out that the two-layer linear neural network with input $\boldsymbol{x} \in \mathbb{R}^d$ analyzed in Lewkowycz et al. (2020) that

$$f(\mathbf{U}, \mathbf{v}; x) = \frac{1}{\sqrt{m}} \mathbf{v}^T \mathbf{U} \boldsymbol{x},$$

where $\mathbf{v} \in \mathbb{R}^m, \mathbf{U} \in \mathbb{R}^{m \times d}$ is a special case of our model with $\mathbf{w} = \left[\text{Vec}(\mathbf{U})^T, \mathbf{v}^T\right]^T$, $\gamma = 1/\sqrt{m}$ and

$$\Sigma = \begin{pmatrix} 0 & I_m \otimes \boldsymbol{x} \\ I_m \otimes \boldsymbol{x}^T & 0 \end{pmatrix} \in \mathbb{R}^{md+m}.$$

# N    EXPERIMENTAL SETTINGS AND ADDITIONAL RESULTS

## N.1    VERIFICATION OF NON-LINEAR TRAINING DYNAMICS OF NQMS, I.E., FIGURE 3

We train the NQM which approximates the two-layer fully-connected neural network with ReLU activation function on 128 data points where each input is drawn i.i.d. from $\mathcal{N}(-2, 1)$ if the label is $-1$ or $\mathcal{N}(2, 1)$ if the label is 1. The network width is $5,000$.

## N.2    EXPERIMENTS FOR TRAINING DYNAMICS OF WIDE NEURAL NETWORKS WITH MULTIPLE EXAMPLES.

We train a two-layer fully-connected neural network with ReLU activation function on 128 data points where each input is drawn i.i.d. from $\mathcal{N}(-2, 1)$ if the label is $-1$ or $\mathcal{N}(2, 1)$ if the label is 1. The network width is $5,000$. See the results in Figure 5.

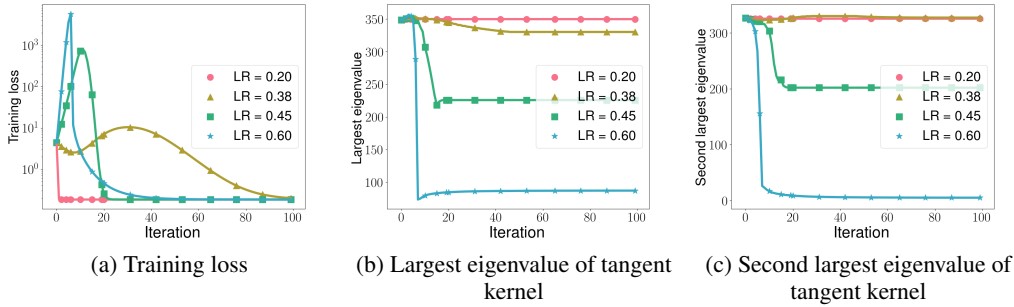

(a) Training loss     (b) Largest eigenvalue of tangent kernel     (c) Second largest eigenvalue of tangent kernel

Figure 5: **Training dynamics of wide neural networks for multiple examples case with different learning rates.** Compared to the training dynamics of NQMs, i.e., Figure 3, the behaviour of of top eigenvalues is almost the same with different learning rates: when $\eta < 0.37$, the kernel is nearly constant; when $0.37 < \eta < 0.39$, only $\lambda_1(t)$ decreases; when $0.39 < \eta < \eta_{\max}$, both $\lambda_1(t)$ and $\lambda_2(t)$ decreases. See the experiment setting in Appendix N.2.

## N.3    TRAINING DYNAMICS CONFINED TO TOP EIGENSPACE OF THE TANGENT KERNEL

We consider the corresponding dynamics equations (15) and (16) for neural networks:

$$\mathbf{f}(t+1) - \mathbf{y} = (I - \eta K(t) + R_{\mathbf{f}}(t))\,(\mathbf{f}(t) - \mathbf{y}), \tag{28}$$
$$K(t+1) = K(t) - R_K(t). \tag{29}$$

Note that for NQMs, $R_{\mathbf{f}}(t)$ and $R_K(t)$ have closed-form expressions but generally for neural networks they do not have.

We consider the training dynamics confined to the top eigenvector of the tangent kernel $\boldsymbol{p}_1(t)$:

$$\langle \boldsymbol{p}_1(t), \mathbf{f}(t+1) - \mathbf{y} \rangle = \left(I - \eta\lambda_1(t) + \boldsymbol{p}_1(t)^T R_{\mathbf{f}}(t)\boldsymbol{p}_1(t)\right) \langle \boldsymbol{p}_1(t), \mathbf{f}(t) - \mathbf{y} \rangle,$$
$$\boldsymbol{p}_1(t)^T K(t+1)\boldsymbol{p}_1(t) = \lambda_1(t) - \boldsymbol{p}_1(t)^T R_K(t)\boldsymbol{p}_1(t).$$

We conduct experiments to show that $\boldsymbol{p}_1(t)^T R_{\mathbf{f}}(t)\boldsymbol{p}_1(t)$ and $\boldsymbol{p}_1(t)^T R_K(t)\boldsymbol{p}_1(t)$ scale with the loss and remain positive when the loss is large. Furthermore, the loss confined to $\boldsymbol{p}_1$ can almost capture the spike in the training loss.

In the experiments, we train a two-layer FC and CNN with width 2048 and 1024 respectively on 128 points from CIFAR-2 (2 class subset of CIFAR-10) and SVHN-2 (2 class subset from SVHN-10). The results for NQM can be seen in Figure 6 and for neural networks can be seen in Figure 7.

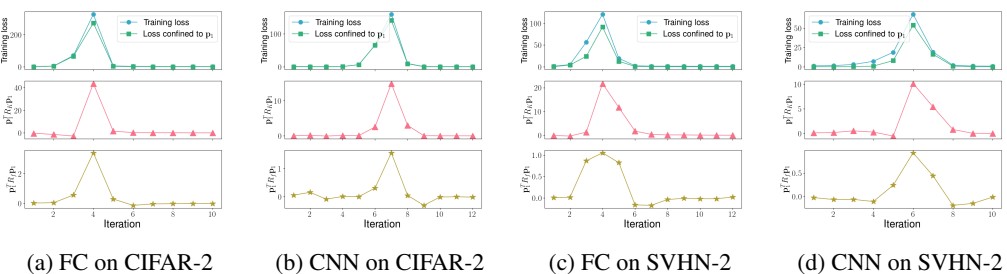

| (a) FC on CIFAR-2 | (b) CNN on CIFAR-2 | (c) FC on SVHN-2 | (d) CNN on SVHN-2 |

Figure 6: **Training dynamics confined to the top eigenspace of the tangent kernel for NQMs.**

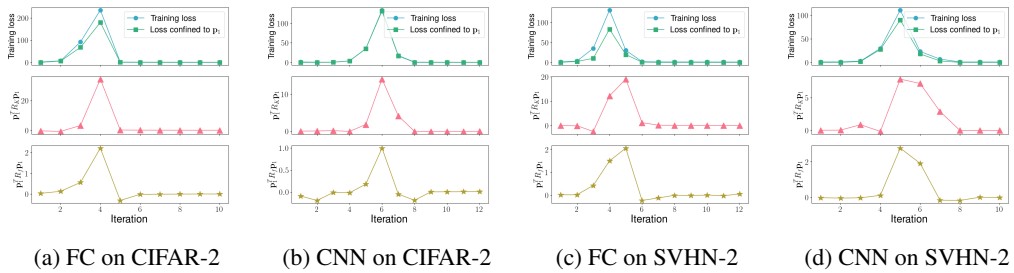

| (a) FC on CIFAR-2 | (b) CNN on CIFAR-2 | (c) FC on SVHN-2 | (d) CNN on SVHN-2 |

Figure 7: **Training dynamics confined to the top eigenspace of the tangent kernel for wide neural networks.**

### N.4 TRAINING DYNAMICS OF GENERAL QUADRATIC MODELS AND NEURAL NETWORKS.

As discussed at the end of Section 3, a more general quadratic model can exhibit the catapult phase phenomenon. Specifically, we consider a general quadratic model:

$$g(\mathbf{w}; \boldsymbol{x}) = \mathbf{w}^T \phi(\boldsymbol{x}) + \frac{1}{2}\gamma \mathbf{w}^T \Sigma(\boldsymbol{x})\mathbf{w}.$$

We will train the general quadratic model with different learning rates, and different $\gamma$ respectively, to see how the catapult phase phenomenon depends on these two factors. For comparison, we also implement the experiments for neural networks. See the experiment setting in the following:

**General quadratic models.** We set the dimension of the input $d = 100$. We let the feature vector $\phi(\boldsymbol{x}) = \boldsymbol{x}/\|\boldsymbol{x}\|$ where $x_i \sim \mathcal{N}(0,1)$ i.i.d. for each $i \in [d]$. We let $\Sigma$ be a diagonal matrix with $\Sigma_{i,i} \in \{-1, 1\}$ randomly and independently. The weight parameters $\mathbf{w}$ are initialized by $\mathcal{N}(0, I_d)$. Unless stated otherwise, $\gamma = 10^{-3}$, and the learning rate is set to be 2.8.

**Neural networks.** We train a two-layer fully-connected neural networks with ReLU activation function on 20 data points of CIFAR-2. Unless stated otherwise, the network width is $10^4$, and the learning rate is set to be 2.8.

See the results in Figure 8.

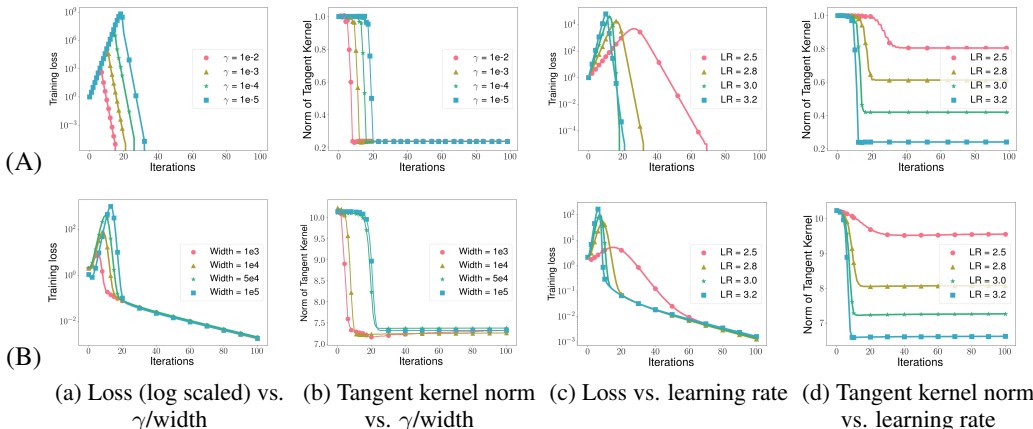

(a) Loss (log scaled) vs. γ/width  (b) Tangent kernel norm vs. γ/width  (c) Loss vs. learning rate  (d) Tangent kernel norm vs. learning rate

Figure 8: **General quadratic models have similar training dynamics with neural networks when trained with super-critical learning rates.** Panel (A): experiments on general quadratic models. Smaller $\gamma$ or larger learning rates lead to larger training loss at the peak. Larger learning rates make tangent kernel decrease more. Panel (B): experiments on two-layer neural networks. Larger width (corresponding to smaller $\gamma$) and larger learning rates have similar effect on the training loss at the peak and decrease of tangent kernel norm with quadratic models. Note that width or $\gamma$ seems to have no effect on the tangent kernel norm at convergence.

## N.5 TEST PERFORMANCE OF $f$, $f_{\text{lin}}$ AND $f_{\text{quad}}$, I.E., FIGURE 2(B) AND FIGURE 4

For the architectures of two-layer fully connected neural network and two-layer convolutional neural network, we set the width to be $5,000$ and $1,000$ respectively. Specific to Figure 2(b), we use the architecture of a two-layer fully connected neural network.

Due to the large number of parameters in NQMs, we choose a small subset of all the datasets. We use the first class (airplanes) and third class (birds) of CIFAR-10, which we call CIFAR-2, and select $256$ data points out of it as the training set. We use the number 0 and 2 of SVHN, and select $256$ data points as the training set. We select $128$, $256$, $128$ data points out of MNIST, FSDD and AG NEWS dataset respectively as the training sets. The size of testing set is $2,000$ for all. When implementing SGD, we choose batch size to be $32$.

For each setting, we report the average result of 5 independent runs.

## N.6 TEST PERFORMANCE OF $f$, $f_{\text{lin}}$ AND $f_{\text{quad}}$ IN TERMS OF ACCURACY

In this section, we report the best test accuracy for $f$, $f_{\text{lin}}$ and $f_{\text{quad}}$ corresponding to the best test loss in Figure 4. We use the same setting as in Appendix N.5.

## N.7 TEST PERFORMANCE OF $f$, $f_{\text{lin}}$ AND $f_{\text{quad}}$ WITH ARCHITECTURE OF 3-LAYER FC

In this section, we extend our results for shallow neural networks discussed in Section 4 to 3-layer fully connected neural networks. In the same way, we compare the test performance of three models, $f$, $f_{\text{lin}}$ and $f_{\text{quad}}$ upon varying learning rate. We observe the same phenomenon for 3-layer ReLU activated FC with shallow neural networks. See Figure 12 and 13.

We use the first class (airplanes) and third class (birds) of CIFAR-10, which we call CIFAR-2, and select 100 data points out of it as the training set. We use the number 0 and 2 of SVHN, and select 100 data points as the training set. We select 100 data points out of AG NEWS dataset as the training set. For the speech data set FSDD, we select 100 data points in class 1 and 3 as the training set. The size of testing set is 500 for all.

For each setting, we report the average result of 5 independent runs.

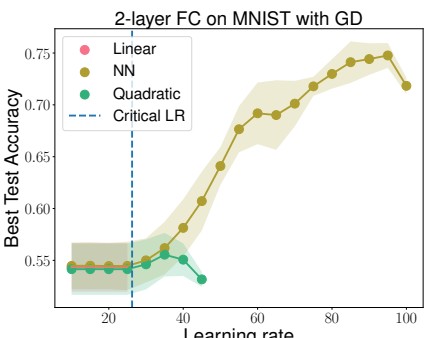 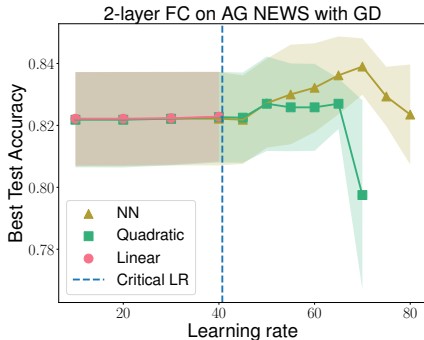

Figure 9: **Best test accuracy plotted against different learning rates for** $f_{\text{quad}}$, $f$, **and** $f_{\text{lin}}$**.** Left panel: 2-layer FC on MNIST trained with GD. Right panel: 2-layer FC on AG NEWS trained with GD.

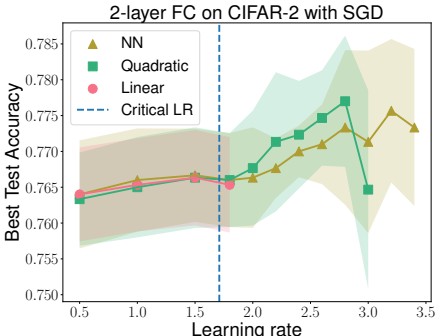 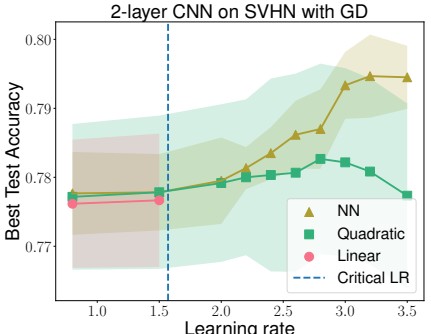

Figure 10: **Best test accuracy plotted against different learning rates for** $f_{\text{quad}}$, $f$, **and** $f_{\text{lin}}$**.** Left panel: 2-layer FC on CIFAR-2 trained with SGD. Right panel: 2-layer CNN on SVHN trained with GD.

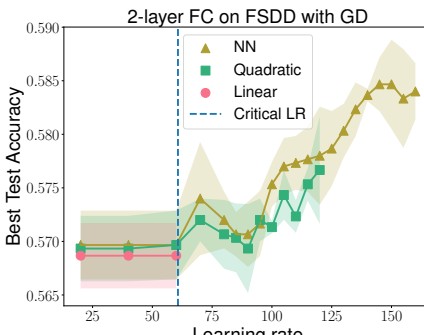 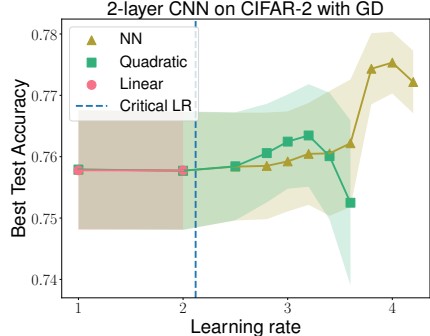

Figure 11: **Best test accuracy plotted against different learning rates for** $f_{\text{quad}}$, $f$, **and** $f_{\text{lin}}$**.** Left panel: 2-layer FC on FSDD trained with GD. Right panel: 2-layer CNN on CIFAR-2 trained with GD.

## N.8 TEST PERFORMANCE WITH TANH AND SWISH ACTIVATION FUNCTIONS

We replace ReLU by Tanh and Swish activation functions to train the models with the same setting as Figure 4. We observe the same phenomenon as we describe in Section 4.

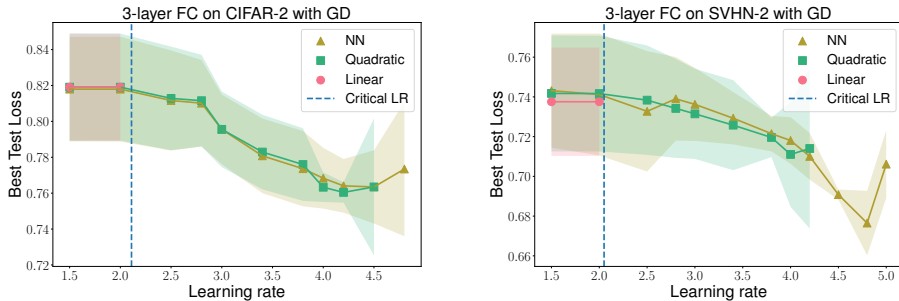

Figure 12: **Best test accuracy plotted against different learning rates for** $f_{\text{quad}}$**,** $f$**, and** $f_{\text{lin}}$**.** Left panel: 3-layer FC on CIFAR-2 trained with GD. Right panel: 3-layer FC on SVHN-2 trained with GD.

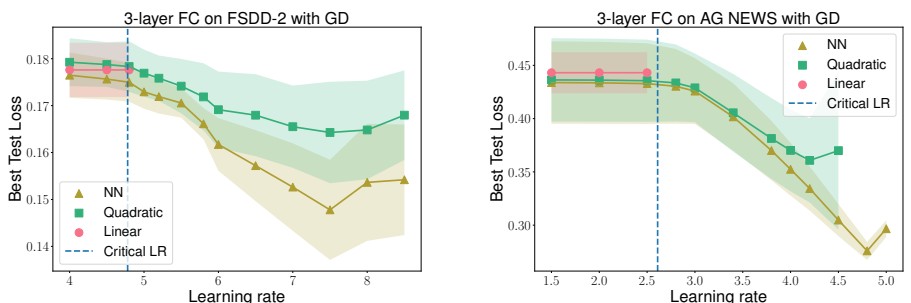

Figure 13: **Best test accuracy plotted against different learning rates for** $f_{\text{quad}}$**,** $f$**, and** $f_{\text{lin}}$**.** Left panel: 3-layer FC on FSDD-2 trained with GD. Right panel: 3-layer FC on AG NEWS trained with GD.

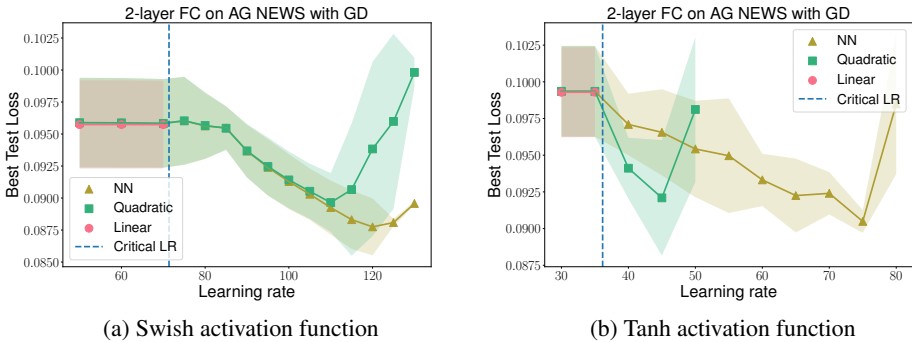

(a) Swish activation function        (b) Tanh activation function

Figure 14: **Best test loss plotted against different learning rates for** $f_{\text{quad}}$**,** $f$**, and** $f_{\text{lin}}$**.** We choose 2-layer FC as the architecture and train the models on AG NEWS with GD.

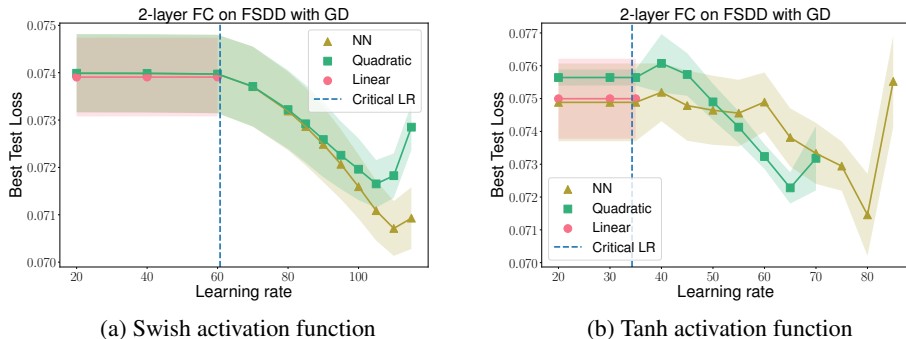

(a) Swish activation function

(b) Tanh activation function

Figure 15: **Best test loss plotted against different learning rates for** $f_{\text{quad}}$**,** $f$**, and** $f_{\text{lin}}$**.** We choose 2-layer FC as the architecture and train the models on FSDD with GD.

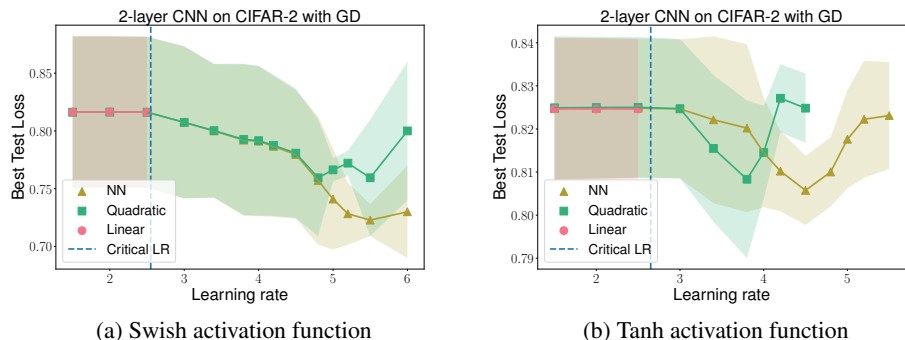

(a) Swish activation function

(b) Tanh activation function

Figure 16: **Best test loss plotted against different learning rates for** $f_{\text{quad}}$**,** $f$**, and** $f_{\text{lin}}$**.** We choose 2-layer CNN as the architecture and train the models on CIFAR-2 with GD.

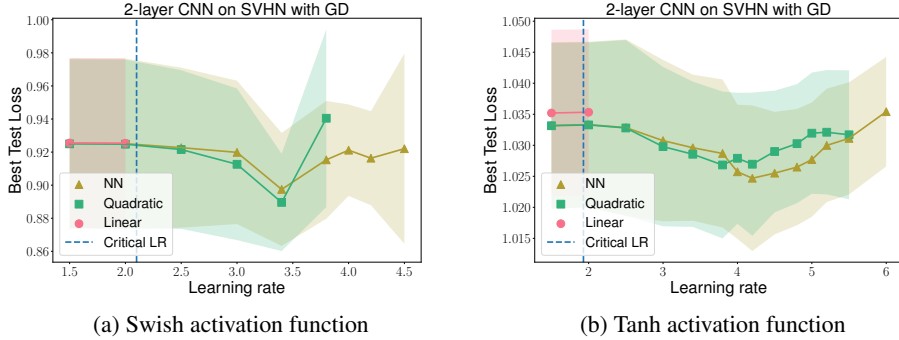

(a) Swish activation function

(b) Tanh activation function

Figure 17: **Best test loss plotted against different learning rates for** $f_{\text{quad}}$**,** $f$**, and** $f_{\text{lin}}$**.** We choose 2-layer CNN as the architecture and train the models on SVHN with GD.

