# OpenReview forum: "Quadratic models for understanding catapult dynamics of neural networks"
_ICLR.cc/2024/Conference — ICLR 2024 poster_

### Official Review · Reviewer_5b1y · 2023-10-23

**Soundness:** 3 good
**Presentation:** 2 fair
**Contribution:** 1 poor
**Rating:** 5
**Confidence:** 3

**Summary:**

This work shows that quadratic models exhibit the catapult phase of neural networks.

**Strengths:**

The strength of the work lies in the soundness of the theory and that the topics covered by the paper are all pertinent problems to the contemporary deep learning theory.

**Weaknesses:**

There are quite a few fatal weaknesses in my opinion.

1. The paper covered too many topics that it feels that it does not achieve any point satisfactorily. For example, the paper feels mistitled. The main focus of the paper is the catapult mechanism -- which does appear in deep learning but cannot represent all types of "neural network dynamics." In my opinion, the catapult mechanism is a rather special / specific type of dynamics and the title is an overclaim. If the authors change the title to a more proper one, the paper could be much easier to evaluate.

2. Lack of discussion of a highly relevant problem. Essentially, within the framework of quadratic models, it appears to me that the catapult mechanism is nothing but what the academia traditionally calls "chaos." For example, consider a width-1 quadratic model, and compare the dynamics of GD to that of a logistic map -- the dynamics is essentially identitical -- the loss of local stability leads to chaos in the logistic map, and the same here in the quadratic model. The authors need to discuss this point in my opinion.

**Questions:**

See weakness


------
I feel more positive about the paper with the new title. However, I still feel that the paper needs to clarify why or why not chaos is present in the model and clarify its connection to the studies in this line.

---

> ### Author Response · Authors · 2023-11-19
>
> We thank the reviewer for the comments. We will address your concerns and questions below.
>
> *W1: The paper covered too many topics that it feels that it does not achieve any point satisfactorily. For example, the paper feels mistitled. The main focus of the paper is the catapult mechanism -- which does appear in deep learning but cannot represent all types of "neural network dynamics." In my opinion, the catapult mechanism is a rather special / specific type of dynamics and the title is an overclaim. If the authors change the title to a more proper one, the paper could be much easier to evaluate.*
>
> We feel the title should not preclude the evaluation of the paper.  Nevertheless, we have changed the title to "Quadratic models for understanding catapult dynamics of neural networks" to emphasize connections to catapult dynamics. We hope this addresses your concern and will result in a more favorable evaluation and score.
>
> *W2: Lack of discussion of a highly relevant problem. Essentially, within the framework of quadratic models, it appears to me that the catapult mechanism is nothing but what the academia traditionally calls "chaos." For example, consider a width-1 quadratic model, and compare the dynamics of GD to that of a logistic map -- the dynamics is essentially identitical -- the loss of local stability leads to chaos in the logistic map, and the same here in the quadratic model. The authors need to discuss this point in my opinion.*
>
> The catapult dynamics considered in our paper is not chaotic, and as such does not seem to be connected with the dynamics of the logistic map.  Would you be able to share a reference so we understand your point better?

---

> > ### Comment · Reviewer_5b1y · 2023-11-23
> > **Reply**
> >
> > The new title feels much better.
> >
> > For chaos, consider the simplest type of quadratic model with width 1: $f(x)=w^2$ where $w$ is a trainable parameter, on a simple MSE loss: $L= (w^2 -1)^2$. This problem is not completely identical to logistic map, but very similar to it.
> >
> > One way to see chaos is this: expand the loss function around the local minimum $w=1$, we obtain the dynamics of the quantity $q_t:=w_t-1$:
> > $$\Delta q_t = - \eta (4q_t + 8 q_t^2) + O(q_t^3),$$
> > which implies that
> > $$q_{t+1} = (1 - 4 \eta) q_t + 8 \eta q_t^2  + O(q_t^3).$$
> > Now, compare this dynamics with the dynamics of the logistic map, they are not identical, but qualitatively the same. At a small $\eta$ the dynamics is approximated by gradient flow, and there is no jump. When $\eta> 1/2$, we have the onset of chaos (due to the loss of stability), and one begins to observe limit cycles and bifurcations (if one plot the distribution of $q_t$). When $\eta$ is larger than $1$ (I did not check this value, but I think it is a close estimate), the dynamics diverges due to global instability. Compare what I said with figure 1b in the paper
> >
> > In fact, in this type of deterministic dynamics on a nonquadratic energy function, chaos almost always appears when the local stability in a local minimum is lost, which is given by the $\eta_{crit}$ in the theory proposed by the authors.

---

> > > ### Author Response · Authors · 2023-11-23
> > >
> > > We note that the catapult dynamic considered in our work is not chaotic. See, e.g., X. Chen, et al. "From Stability to Chaos: Analyzing Gradient Descent Dynamics in Quadratic Regression"  arXiv:2310.01687 (2023) for a nice discussion (we will add the ref. to the paper).
> > >
> > > We hope we have addressed your concerns. If so, please consider raising your score.

---

> > > > ### Comment · Reviewer_5b1y · 2023-11-23
> > > > **Reply**
> > > >
> > > > Thanks for the nice reference.
> > > >
> > > > This raises another question. Naively, I expect the problem studied in the present manuscript has a chaos phase. This reference (arXiv:2310.01687) also shows that there is a chaos phase in the related problems. However, the authors claim that there is no chaos in the problem in the manuscript. Why is this case? I think the authors ought to study, discuss the reason and compare with this reference they mentioned.

---

### Official Review · Reviewer_JHXp · 2023-10-31

**Soundness:** 3 good
**Presentation:** 3 good
**Contribution:** 3 good
**Rating:** 8
**Confidence:** 3

**Summary:**

I was asked to review this paper at the last minute, so although I did not get the chance to work carefully through the proofs, I can comment on the broad content and contributions

The paper analyzes optimization and generalization properties of neural networks using quadratic models. It shows theoretically and empirically that quadratic models exhibit the "catapult phase" with large learning rates, explaining a property of neural networks not captured by linear models. The quadratic model and its (changing) tangent kernel is studied analytically for the case of a single training example and multiple uni-dimensional training samples. Three regimes are identified. When catapault effects occur, better generalization is observed in quadratic models. Experiments demonstrate quadratic models parallel neural networks better than linear models in generalization with large learning rates.

**Strengths:**

The paper rigorously analyzes the catapult phase in NQMs theoretically. Prior works have attempted similar analyses, but the presentation here is particularly readable. The experiments validating theory and demonstrating applicability are thorough. They successfully demonstrate that NQMs better capture neural network behavior than linear models. The experiments in the appendix in particular provides good additional support. The mathematical analysis appears solid.

**Weaknesses:**

The architectures and datasets used are still relatively limited. It would also be good to highlight what the incremental value of this paper is over prior work on quadratic models. A CNN and transformer experiment would be particularly welcome.

From a cursory literature review, I have found this paper which has relatively substantial overlap in topic, and I believe should at least be cited (Meltzer and Liu https://arxiv.org/abs/2301.07737). A more thorough set of references on recent work around the catapult effect and edge of stability would also benefit the paper. If these changes are incorporated and a more thorough related works section is added, I will likely raise my score.

From empirical work, I've also seen that quadratic model's validation curve only tracks the NN's curve in the early stages of training, but then diverges from it. It would be nice to give an example of the limitations of the quadratic model for understanding generalization. Being clear about potential limitations of quadratic models to fully model neural networks would be welcome.

**Questions:**

Have you analyzed how the Hessian evolves during catapult phase? This could shed some light onto the phase transition.

It may be interesting to consider explicitly varying the feature learning parameter (e.g. $\alpha$ in Chizat et al) and see how the quadratic model differs from the linear one. This would be related to going into $\mu$ parameterization as in Yang.

Can you relate observed properties of trained NQMs to improvements in generalization, perhaps through the lens of something like kernel-target alignment for the after-kernel of the quadratic model?

---

> ### Author Response · Authors · 2023-11-19
>
> We thank the reviewer for the positive feedback and insightful comments. We will address your concerns and questions below.
>
> *W1.a: The architectures and datasets used are still relatively limited... A CNN and transformer experiment would be particularly welcome.*
>
>  Indeed we have experiments with CNNs as shown in Figure 4 and 16.  We experimented with various datasets, including vision (CIFAR10, SVHN, MNIST), speech (FSDD) and language (AG NEWS). As a primarily theoretical paper, we believe this is a good scope of experiments.
>
> *W1.b:  It would also be good to highlight what the incremental value of this paper is over prior work on quadratic models.*
>
> The related paper [1] introduced and analyzed the NQM (called quadratic models in their book), but our work analyzed the training dynamics with large learning rates, which was not discussed in [1].
>
> 1. Roberts, Daniel A., Sho Yaida, and Boris Hanin. The principles of deep learning theory. Cambridge, MA, USA: Cambridge University Press, 2022.
>
> *W2: From a cursory literature review, I have found this paper which has relatively substantial overlap in topic, and I believe should at least be cited (Meltzer and Liu https://arxiv.org/abs/2301.07737). A more thorough set of references on recent work around the catapult effect and edge of stability would also benefit the paper.*
>
> Thanks for pointing out this related work. In fact, the first version of our work was prior to [Meltzer and Liu], and we weren't aware of it. We added a discussion in the updated draft.
>
> As for the Edge of Stability (EoS), we have not seen a clear connection between EoS and catapult dynamics. Although there seems to be some similarities between Edge of Stability (EoS) and catapult dynamics at first glance, we noticed a few key differences between the two: 1) the training loss can increase to a large value,  a order of $\Theta(m)$ with $m$ being the network width, in the catapult dynamics, while it does not observed to increase significantly in EoS, 2) there is a single loss spike in the catapult dynamics of full-batch gradient descent, while there are often multiple loss oscillations in EoS, 3) the catapult phase phenomenon often improve generalization across various tasks [Lewkowycz et al. 2020], while it is unclear if EoS has any effect on the generalization.
> The original EoS paper [1] conjectures that the training loss oscillations could be micro-catapults, however, there is no direct evidence for the connection between EoS and catapult dynamics.
>
> 1. J. Cohen,  et al. "Gradient Descent on Neural Networks Typically Occurs at the Edge of Stability." International Conference on Learning Representations. 2020.
>
> *W3: From empirical work, I've also seen that quadratic model's validation curve only tracks the NN's curve in the early stages of training, but then diverges from it. It would be nice to give an example of the limitations of the quadratic model for understanding generalization. Being clear about potential limitations of quadratic models to fully model neural networks would be welcome.*
>
> In our experiments, the generalization performance of neural networks seems to perform better than their corresponding NQMs when the learning rate is large, i.e., catapult occurs. One possible reason is that  NQMs are the second-order approximation of neural networks, while they can exhibit the catapult phase phenomenon the same as the neural networks, they may not be able to learn the same features as the neural networks. That is to say, to learn certain features, higher-order terms are needed. This is a complex question and a direction for future work.

---

> > ### Author Response · Authors · 2023-11-19
> >
> > *Q1: Have you analyzed how the Hessian evolves during catapult phase? This could shed some light onto the phase transition.*
> >
> > Indeed, the spectrum of the Hessian of the loss is very close to the spectrum of the tangent kernel in the kernel regime. This can be seen as follows:
> >
> > For MSE  $\mathcal{L}(W;X) = \frac{1}{n}\sum_{i=1}^n (f(W;x_i)-y_i)^2$, we can compute its $H_\mathcal{L}$ by the chain rule:
> > \begin{align*}
> >     H_\mathcal{L}(w) = \frac{2}{n}({{A}(w)}  +B(w)),
> > \end{align*}
> > where $A(w) = \sum_{i=1}^n \frac{\partial f(w;x_i)}{\partial w}^T \frac{\partial f(w;x_i)}{\partial w}$ and $B(w) =\sum_{i=1}^n  (f(w;x_i)-y_i)\frac{\partial^2 f(w;x_i)}{\partial w^2}$
> >
> > Note that the term $A(w)$ has the same spectrum as NTK, except zero eigenvalues. Term $\|B(w)\|$ is sufficiently small when the network has large width, due to "transition to linearity"[1]. Therefore, the spectrum of the tangent kernel is close to the spectrum of $H_\mathcal{L}(w)$.
> >
> > As the top eigenvalue of the tangent kernel decreases during the catapult, we generally would expect the top eigenvalue of the Hessian of the loss also decrease. As to the behavior of other eigenvalues, it is a complicated problem and is beyond the scope of our work.
> >
> > 1. C. Liu, et al. "On the linearity of large non-linear models: when and why the tangent kernel is constant." Advances in Neural Information Processing Systems 33 (2020): 15954-15964.
> >
> > *Q2: It may be interesting to consider explicitly varying the feature learning parameter (e.g. $\alpha$ in Chizat et al) and see how the quadratic model differs from the linear one. This would be related to going into $\mu$ parameterization as in Yang.*
> >
> > Thank you for your suggestion. This is definitely an interesting problem to look into.
> >
> > *Q3: Can you relate observed properties of trained NQMs to improvements in generalization, perhaps through the lens of something like kernel-target alignment for the after-kernel of the quadratic model?*
> >
> > We are aware that the kernel target alignment was shown to correlate with the generalization performance and is a useful feature learning measurement. We expect that the after kernel will lead to a larger kernel target alignment compared to the initial NTK, as discussed in [1]. Similarly, catapults seem to promote feature learning [2]. However, analyzing the generalization performance of the after-kernel is beyond the scope of our work.
> >
> >
> > 1. A. Atanasov, et al. "Neural Networks as Kernel Learners: The Silent Alignment Effect." International Conference on Learning Representations. 2021.
> >
> > 2. L. Zhu, et al. "Catapults in SGD: spikes in the training loss and their impact on generalization through feature learning." arXiv preprint arXiv:2306.04815 (2023).

---

> ### Comment · Reviewer_JHXp · 2023-12-04
> **Response to Official Comment by A**
>
> I thank the authors for their input. The distinction the authors' made in their rebuttal between studying EoS dynamics and catapult dynamics was helpful in being able to understand the relative novelty of this paper. The paper by Agarwala, Pedregosa, and Pennington: "Second-order regression models exhibit progressive sharpening to the edge of stability" tackles the EoS phenomenon for quadratic models. I'd encourage the authors to add a sentence of two about the distinction between these effects in the final draft to remove confusion between their contribution and earlier ones. Thanks also for including the discussion of the Meltzer and Liu paper. These additions will substantially improve the literature review.
> After some thought, I have decided to raise my score. This is due to the set of experiments pointed out to me, together with the clear introduction and general importance of the problem being studied. My confidence in my understanding of the proofs is still relatively low, but there are no glaring errors that I have found. Also, the empirical results give credence to the claims.

---

### Official Review · Reviewer_F9oB · 2023-11-01

**Soundness:** 3 good
**Presentation:** 3 good
**Contribution:** 3 good
**Rating:** 5
**Confidence:** 4

**Summary:**

This paper proposes using neural quadratic models -- second order taylor expansion of any function f around initial parameters $w_0$ -- as a way to study neural network dynamics. The authors start by explaining that linear dynamics perspective of neural networks falls short in explaining some of the behaviors in neural network dynamics; specifically, catapult phase of learning rate. They approximate a two layer neural network using its second order expansion. They first show that for a single training example, where tangent kernel reduces to a scalar, monotonic convergence, catapult phase, and divergence are separated based on learning rate and inverse of kernel value at random initialization. They further extend their analysis to uni-dimensional multiple example setting where the analysis is driven by the eigenvectors of the kernel matrix. Finally, the authors empirically show that for wide neural networks, catapults happen in the top eigenspace of the kernel -- similar to the multiple example setting. Experimental results suggest that catapult phase results in lower test error compared to error with sub-critical learning rate for the quadratic model, mimicing the dynamics of neural networks more closely.

**Strengths:**

The paper is written well and in general easy to follow. Empirical results on top eigenspace of tangent kernel for analyzing general wide neural networks is convincing.

**Weaknesses:**

Several assumptions about underlying the theory are not clear and different from the practice.

1. The neural network in Eq (9) is initialized as $u_i \sim \mathcal{N}(0, I_d)$ and $v_i \sim Unif(-1, 1)$. Can you explain why $v_i$ is initialized different from $u_i$, also different from the typical inverse fanin/fanout initialization in practice?

2. The assumption that width (m) is larger than data size (n) which is assumed to be a small constant is unrealistic. While in the limit where width goes to $\inf$, this would be the case but for any finite width network, this does not hold in practice. Is this assumption crucial, can you still assume that $n/m$ does not necessarily go to zero?

3. $p_1(t)$ is the top eigenvector of $K(t)$. Given that $p_1(t)$ is not necessarily equal to $p_1(t+1)$, it is not clear how you derived $\lambda_1(t+1)=\lambda_1(t)-p_1(t)^TR_K(t)p_1(t)$. Could you please explain in more detail?

4. In Eq(12), $R_\lambda(t)$ is defined without the minus sign. In the following paragraph, you mention that "$R_\lambda(t)$ stays positive and results in monotonic decrease of kernel" which makes sense since $\lambda(t)=\lambda(t-1)-R_\lambda(t)$. But in the next paragraph, you write down $\lambda(t)=\lambda(0)+\sum_{\tau=0}^{t} R_\lambda(\tau)$ which suggests that $R_\lambda(t)$ should include the minus sign. Please clarify.

5. Related to above, I think it should be $\lambda(t+1)=\lambda(0)+\sum_{\tau=0}^{t} R_\lambda(\tau)$ or $\lambda(t)=\lambda(0)+\sum_{\tau=0}^{t-1} R_\lambda(\tau)$

6.  You mention in the decreasing phase section in page 6 that decrease in $v(t)$ would cause a decrease in $\kappa(t)$. But in Eq (13), reducing $v(t)$ would increase inside of square which should lead to an increase in $\kappa(t)$; unless, $u(t)+w(t)<0$ which is not clear if it holds. Please clarify.

7. In Eq (10), $1/\sqrt(d)$ is missing from $\sigma(u_{0,i}^Tx)$. It is present in Appendix A.

8. Page 27 in the Appendix, it should be $\Pi_1 \mathcal{L}(t)=K_1(t)\Pi_1 \mathcal{L}(t-1)$

**Questions:**

Please see above for related questions as they are more meaningful within their respective contexts.

---

> ### Author Response · Authors · 2023-11-19
>
> We thank the reviewer for the insightful comments. We will address your concerns and questions below.
>
> *W1: The neural network in Eq (9) is initialized as $u_i \sim \mathcal{N}(0,I)$ and $v_i \sim Unif(-1,1)$. Can you explain why $v_i$ is initialized different from $u_i$, also different from the typical inverse fanin/fanout initialization in practice?*
>
> This initialization/parameterization strategy adopted in our paper is the so-called NTK parameterization [1]. It is standard in theoretically analyzing neural networks with large width, e.g., [1,2,3].  In the "typical inverse fan-in/fan-out initialization in practice", NTK scales as $O(m)$, which blows up for large width. However, in the NTK parameterization, NTK scales as $\Theta(1)$.
>
> The reason why we use the uniform distribution of $v_i$, instead of the normal distribution $\mathcal{N}(0,1)$, is to make sure $v_i$ is bounded, which simplifies the analysis. In the case of  $v_i \sim \mathcal{N}(0,1)$, similar results will hold with extra $log(m)$ factors in some $O$ terms, with a high probability that depends on $m$.
>
> Drawing $v_i$ from a bounded distribution is a common setting in literature, for example, [2,3].
>
> 1. A. Jacot, et al. “Neural tangent kernel: Convergence and
> generalization in neural networks”. NeurIPS, 2018.
>
> 2.  S. Du S., et al. "Gradient Descent Provably Optimizes Over-parameterized Neural Networks." International Conference on Learning Representations. 2018.
>
> 3. Z. Ji, and T. Matus. "Polylogarithmic width suffices for gradient descent to achieve arbitrarily small test error with shallow ReLU networks." International Conference on Learning Representations. 2019.
>
> *W2: The assumption that width ($m$) is larger than data size ($n$) which is assumed to be a small constant is unrealistic. While in the limit where width goes to $inf$, this would be the case but for any finite width network, this does not hold in practice. Is this assumption crucial, can you still assume that $n/m$ does not necessarily go to zero?*
>
> Having $m>n$, i.e., over-parameterization is a common setting in theoretical research of recent years. For example, [1] assumes $m = \Omega(n^6)$, [2] assumes $m = \Omega(n^4)$, [3] assumes $m = \Omega(n^{24})$. Many interesting findings and results in this large width setting can often be experimentally observed in practical settings. In practice, the catapult dynamics still occur when the width and the size of datasets are comparable. For example, in our experiment for FSDD dataset, the catapult still occurs when the width is $1,000$ and the size of the dataset is $256$.
>
> 1. S. Du S., et al. "Gradient Descent Provably Optimizes Over-parameterized Neural Networks." International Conference on Learning Representations. 2018.
> 2. S. Du S., et al. "Gradient descent finds global minima of deep neural networks." International conference on machine learning. PMLR, 2019.
> 3. Z. Allen-Zhu, et al. "A convergence theory for deep learning via over-parameterization." International conference on machine learning. PMLR, 2019.
>
> *W3: Given that $p_1(t)$ is not necessarily equal to $p_1(t+1)$, it is not clear how you derived $\lambda_1(t+1) = \lambda_1(t) - p_1(t)^T R_K(t) p_1(t)$...*
>
> We suppose you were referring to Appendix N.3. This is a typo: the equation should be $p_1(t)^T K(t+1)p_1(t) = \lambda_1(t) - p_1(t)^T R_K(t) p_1(t)$. The experiments in Appendix N.3 show that the training dynamics confined to the top eigenspace of the tangent kernel i.e., $p_1(t)$ can almost capture the catapult dynamics.  We have fixed it in the revision.
>
> *W4: In Eq(12), $R_\lambda(t)$ is defined without the minus sign....which suggests that $R_\lambda(t)$ should include the minus sign.*
>
> *W5: Related to above, I think it should be $\lambda(t) = \lambda(0) - \sum_{\tau = 0}^{T-1} R_\lambda(\tau)$ ...*
>
>
> Thanks for pointing it out. We have fixed the typo.
>
> *W6: You mention in the decreasing phase section in page 6 that decrease in $v(t)$ would cause a decrease in  $\kappa(t)$. But in Eq (13), reducing
> $v(t)$ would increase inside of square which should lead to an increase in  $\kappa(t)$; unless $u(t) + w(t)<0$ which is not clear if it holds. Please clarify.*
>
> Note that at initializaiton $v(0) > 2$ and $u(0) + w(0) = O(1/m)$. Therefore $\kappa(0) = (1-v(0) + O(1/m))^2 >1$ since $1-v(0) < -1$. As $v(t)$ decreases, $|1-v(t)|$ will also decrease therefore $\kappa(t)$ decreases.
>
> Additionally, the fact that $u(t) + w(t)>0$ and keeps increasing will further make $\kappa(t)$ decrease.
>
> *W7: In Eq.(10), $1/\sqrt{d}$ is missing from $\sigma(u_{0,i}^T x).$ It is present in Appendix A*
>
> In the theoretical analysis, we only consider ReLU activation function which is homogenous, hence we can extract the scaling factor $1/\sqrt{d}$ out of the function $\sigma(\cdot)$.
>
> *W8: Page 27 in the Appendix, it should be $\Pi_1 L_1(t) = K_1(t) \Pi_1 L(t-1$*
>
> We denote the scaling factor by $\kappa_1(t)$, consistent with the notation used in the single training sample scenario.

---

### Official Review · Reviewer_e4J5 · 2023-11-01

**Soundness:** 3 good
**Presentation:** 3 good
**Contribution:** 3 good
**Rating:** 8
**Confidence:** 1

**Summary:**

The paper studies the "catapult phase" during training models with large learning rates.
Importantly, it proposes Neural Quadratic Models (NQMs) as a tool to study it and proves that when large learning rates are used they exhibit a similar catapult phase as modern neural networks, which is not the case of other tools such as linear models, a popular theoretical tool to analysis the learning of neural networks.
The paper presents these findings with proofs and suggests that NQMs can be useful tools to analyze neural networks in the future.

**Strengths:**

**originality** The finding of the paper seems to be novel and the proposal of using NQMs can be new as well.

**quality** The finding and theory from the paper seems to be sound.

**clarity** The paper is overall well-written but can be hard to follow for people who are not very familiar with the field.

**significance** The contribution of the paper seems to be significant and can enable many future work on analysis with NQMs, which could lead to more useful findings than the catapult phase.

**Weaknesses:**

The proposed term Neural Quadratic Models seems to be unnecessary as for linear models we don't call them Neural Linear Models.

**Questions:**

Some recent findings such as the deep double descent are high-dependent on the size of the neural networks and other hyper-parameters.
Is the "catapult phase" here sensitive to other hyper-parameters apart from learning rates?
I don't see many conditions for the theorem presented in the paper.

---

> ### Author Response · Authors · 2023-11-19
>
> We thank the reviewer for the positive feedback and insightful comments. We will address your concerns and questions below.
>
> *W1: The proposed term Neural Quadratic Models seems to be unnecessary as for linear models we don't call them Neural Linear Models.*
>
> The term "quadratic model" can refer to any model that generically has a quadratic form. In this paper, we study the special instance: the quadratic model that best approximates the neural network. Hence, we think the term "Neural Quadratic Model” describes this specific model the best. This is similar to the situation of "Neural tangent kernel" (NTK). NTK is a specific kernel associated with neural networks}
>
> We believe it is important to highlight the close relation between the quadratic models and neural networks since they are the second-order Taylor expansion of the networks.
>
> Some works, e.g., [1] indeed refer to the linear approximation of neural networks as neural tangent models, to highlight their close relation.
>
> 1. B. Ghorbani, et al. (2019). Limitations of lazy training of two-layers neural network. NeurIPS.
>
> *Q1: Some recent findings such as the deep double descent are high-dependent on the size of the neural networks and other hyper-parameters. Is the "catapult phase" here sensitive to other hyper-parameters apart from learning rates? I don't see many conditions for the theorem presented in the paper.*
>
> The width $m$ of the network affects the magnitude of the catapult dynamics. As we have shown in the paper, the loss value at the peak of the catapult scales as $O(m/\log m)$. We don't observe dependence on other hyper-parameters.

---

> > ### Comment · Reviewer_e4J5 · 2023-11-22
> >
> > Thanks the authors for their clarification. I acknowledge that I've read the response.

---

### Official Review · Reviewer_Unww · 2023-11-09

**Soundness:** 2 fair
**Presentation:** 4 excellent
**Contribution:** 3 good
**Rating:** 6
**Confidence:** 1

**Summary:**

This paper studies a quadratic approximation of two-layer neural networks to understand optimization behaviours that cannot be captured by linear models. More precisely they theoretically evidence the so-called catapult dynamics for respectively one and multiple training points, ie they show the existence of two critical values for the learning rate which delimits respectively exponential convergence to the minimum, catapult dynamics (ie first increase of the loss then convergence to low loss) and finally divergence. Finally they provides experiments evidencing catapult dynamics as well as the fact that the quadratic approximation of a neural networks shows similar behaviours that the neural network above the critical learning rate.

**Strengths:**

This paper is well-written with clear figures and a good explanation of the setting and the results. This paper studies a very interesting phenomenon about the optimization of neural networks and has to deal with non-linear phenomena which are usually not well understood. They are able to evidence theoretically the catapult dynamics and the existence of two critical learning rates when dealing with neural quadratic models.

Empirical results shed light on the similarity of behaviours in terms of generalization between the NQM and the neural network function, evidencing the coherence of the quadratic model.

**Weaknesses:**

I find the proofs a bit hard to understand because of the use in the proofs of O,o, $\Omega$ notations. Hiding constants behind such notations makes the proofs a bit cloudy to me.

1)Especially I have some concerns about the proof of lemma 1 in appendix D: you use a proof by induction, and still use O,o notations. But summing $O$ terms remains $O$ only when the number of iterations is controlled. To still have $O$ in the end of the summation with respect to m, it should be checked that the summation index T is itself $O(1)$. However I am not sure such a result is proved (correct me if I'm wrong).
Especially it seems to me that it might not be the case by considering the fact that T must satisfy a relation of the form $(1+\delta)^T\sim \log(m)$ with $\delta \sim \frac{\log(m)}{\sqrt{m}}$. Indeed $u(t)$ goes from $O(1/m)$ to $O(\log(m)/m)$. In that case, $T\sim \frac{\log(\log(m))\sqrt{m}}{\log(m)}$ which is not $O(1)$.

2) Another ambiguity about the $O$ notation is for example when it is stated: $\kappa(0)>(1+\delta-O(1/m))^2>(1+\delta-O(\delta^2/\log(m)))^2$ in appendix D. In full generality it seems wrong for general sequences as it depends on the constants in the $O$ itself and their sign. For example $(1+\delta))^2<(1+\delta-(-3\delta^2/\log(m)))^2$. I think this statement in its current form would perhaps need an additional study of the constants, their sign or if they are zero.

3) The non-linearity that is used in this paper is a ReLU: it allows to compute easily the second derivative of the neural network function (it deletes the diagonal terms of the hessian of the neural network function). My only concern is regarding the generalization to non-linearities that are not piecewise linear and hence which make another term appear in the Hessian of the neural network function, which corresponds to the second derivative of the non-linearity (correct me if I'm wrong). Could the author comment about how to handle this and if they expect the analysis to be the same and the results to still hold?

**Questions:**

1) I am curious if you could clarify the links, if they exist, between catapult dynamics and recent works on edge of stability analysis of neural networks (cf the paper "Second-order regression models exhibit progressive sharpening to the edge of stability" that is not cited but seems to me very related to your setting). It seems interesting because both studies explore the influence of a non-linear quadratic term on the coupled dynamics of the sharpness and the learning rate.

2) It is written in the text that the results hold with high probability over initialization but I'm not sure if this is written in the statement of the theorems. Perhaps it could be added in the theorems themselves.

3) I find the results interesting and they provide good contribution. I would increase my score if the authors address my concerns about the clarity of $O,o,\Omega$ notations in the proof, especially the proof by induction.

**Details Of Ethics Concerns:**

This paper is mostly theoretical and does not present ethics concerns.

---

> ### Author Response · Authors · 2023-11-19
>
> We thank the reviewer for the insightful comments. We will address your concerns and questions below.
>
> *W1: I have some concerns about the proof of lemma 1 in appendix D ... But summing $O$ terms remains $O$ only when the number of iterations is controlled. To still have $O$ at the end of the summation with respect to $m$, it should be checked that the summation index $T$ is itself $O(1)$. However I am not sure such a result is proved (correct me if I'm wrong). Especially it seems to me that it might not be the case by considering the fact that T must satisfy a relation of the form $(1+\delta)^T \sim \log m$ with $\delta \sim \log m/\sqrt{m}$. Indeed $u(t)$ goes from $O(1/m)$ to $O(\log m /m)$. In that case, $T \sim \log (\log m)\sqrt{m}/\log m$ which is not $O(1)$.*
>
> In the proof of Lemma 1, $u(t)$ grows geometrically from $O(1/m)$ to $O(\log m /m)$ with rate $\kappa(t) \geq (1+\delta - O(\delta^2/\log m))^2>1$ for large enough $m$. Note that  the summation of a sequence that grows geometrically with a ratio $(1+\delta)^2>1$ can be controlled by the last term, i.e., $\sum_{t=1}^T u(t) = \Theta(u(T)/\delta)$.
> Hence, even if $T$ is not $O(1)$, our results still hold. We updated the draft to clarify this point.
>
> *W2: Another ambiguity about the $O$ notation is for example when it is stated: $\kappa(0)>(1+\delta - O(1/m))^2 > (1+\delta - O(\delta^2/\log m))^2$ in appendix D.  In full generality it seems wrong for general sequences as it depends on the constants in the $O$ itself and their sign. I think this statement in its current form would perhaps need an additional study of the constants, their sign or if they are zero.*
>
> We apologize for the confusion. In our analysis, the $O$-terms are always positive, and we explicitly write out their sign before $O$. For this inequality $\kappa(0)>(1+\delta - O(1/m))^2 > (1+\delta - O(\delta^2/\log m))^2$, we have $m$ large enough to make sure that each term within the brackets is positive. We have updated the draft to clarify this.
>
> *W3: My only concern is regarding the generalization to non-linearities that are not piecewise linear... Could the author comment about how to handle this and if they expect the analysis to be the same and the results to still hold?*
>
>  While our analysis is specific to ReLUs, our experiments for tanh and swish activation functions in Appendix N.8 also verify that our results extend to such smooth activation functions. In the literature, the analysis for smooth non-linear activation functions is often handled differently from ReLUs.
>
>
>
> In addition, ReLU is so widely used in both theory and practice that many important theoretical results are specific to neural networks with ReLU activations.  See for example:[1,2,3]
>
> 1. S. Du S., et al. "Gradient Descent Provably Optimizes Over-parameterized Neural Networks." International Conference on Learning Representations. 2018.
> 2. Z. Ji, and T. Matus. "Polylogarithmic width suffices for gradient descent to achieve arbitrarily small test error with shallow ReLU networks." International Conference on Learning Representations. 2019.
> 3. D. Zou, et al. (2020). Gradient descent optimizes over-parameterized deep ReLU networks. Machine learning, 109, 467-492.
>
> *Q1. I am curious if you could clarify the links, if they exist, between catapult dynamics and recent works on edge of stability analysis of neural networks...*
>
> Although there seem to be some similarities between Edge of Stability (EoS) and catapult dynamics at first glance, we noticed a few key differences between the two: 1) the training loss can increase to a large value, an order of $\Theta(m)$ with $m$ being the network width, in the catapult dynamics, while it is not observed to increase significantly in EoS, 2) there is a single loss spike in the catapult dynamics of full-batch gradient descent, while there are often multiple loss oscillations in EoS, 3) the catapult phase phenomenon often improve generalization across various tasks [Lewkowycz et al. 2020], while it is unclear if EoS has any effect on the generalization.
> The original EoS paper [1] conjectures that the training loss oscillations could be micro-catapults, however, there is no direct evidence for the connection between EoS and catapult dynamics.
>
> 1. J. Cohen, et al. "Gradient Descent on Neural Networks Typically Occurs at the Edge of Stability." International Conference on Learning Representations. 2020.
>
> *Q2: It is written in the text that the results hold with high probability over initialization but I'm not sure if this is written in the statement of the theorems. Perhaps it could be added in the theorems themselves.*
>
> Thanks for the suggestion. We have updated it in our revision.

---

> > ### Comment · Reviewer_Unww · 2023-11-19
> > **Response to the authors about remaining concerns**
> >
> > I thank the authors for the clarifications. However I feel that my concerns were not completely solved:
> >
> > W1: I get your explanation but I don't think that it answers my concern. My concern is about the method itself, which is an inductive proof by induction to show that $\kappa(t)=(1+\delta-O(\frac{\delta^2}{\log(m)})^2$.
> >
> > I agree both 1) that it is true at initialization 2) with your previous argument on the summation $\sum u(t)$ terms: you indeed get that the inductive property propagates: $\kappa(t+1)=(1+\delta-O(\frac{\delta^2}{\log(m)})^2$.
> > However my point is that it is valid only for $O(1)$ iterations because at each step you sum $O$ terms and group them inside one single $O$. You should ensure that the total number of $O$ terms that are summed is itself $O(1)$, i.e. that the number of iterations is itself $O(1)$, ie $T=O(1)$ which doesn't seem to be true.
> > More generally I don't think it is right to do a proof by induction with $O$ notations, without controlling the constants. For example, it is trivial that using such methods, one would have $O(m)=O(1)$ by induction (true at initialization and the property propagates one by one because $m+1=m+1=O(1)+O(1)=O(1)$). Therefore I remain unconvinced by using such a notation without properly controlling the constants.
> >
> > W2: I don't think this solves the problem: $(1+\delta-\frac{1}{m})^2<(1+\delta)^2$ seems to be a counter example.
> >
> > Finally I thank the authors for their answers to the other questions I had and would be thankful if they could clarify my remaining concerns.

---

> > > ### Author Response · Authors · 2023-11-20
> > >
> > > We thank you for the rely and insightful comments. We will address your concerns below.
> > >
> > > *W1:  However my point is that it is valid only for $O(1)$ iterations because at each step you sum $O$ terms and group them inside one single $O$ You should ensure that the total number of $O$ terms that are summed is itself $O(1)$, i.e. that the number of iterations is itself $O(1)$, ie $T = O(1)$ which doesn't seem to be true. More generally I don't think it is right to do a proof by induction with  $O$ notations, without controlling the constants....Therefore I remain unconvinced by using such a notation without properly controlling the constants.*
> > >
> > > We have revised the proof of Lemma 1 by using explicit constants to replace the $O,o,\Omega$ notation. Our results hold as the dependence on the induction step $t$ will be absorbed into the summation of the geometric sequence.
> > >
> > > The logic of Lemma 1 is as follows: we consider the interval $[0,T]$ where $T$ is the first time that $u(T)$ increased to $\Theta(\delta^2/\log m)$. To prove Lemma 1, we only need to show that the ratio $u(t+1)/u(t):=\kappa(t) >1$ for each $t\in [0,T]$ so that $u(t)$ keeps increasing until reaching to the order $\Theta(\delta^2/\log m)$. Note that, in this whole procedure,  the accumulation of $O$ terms across $T$ steps does not present a concern. To show $\kappa(t)>1$, we need to bound the change of $\kappa(t)$ from $\kappa(0)$ to be small, so that $\kappa(t)$ remains above $1$ by triangle inequality. Note that the change of $\kappa(t)$ from $\kappa(0)$ is controlled by  $u(t)$ and $v(t)$. As $u(t)$ increases geometrically and $u(t) = O(\delta^2/\log m)$ for all $t\in[0,T]$,  we can derive a bound of the sum of $u(t)$ independent of $t$. A similar bound can be done for $v(t)$. Therefore, we derive a bound for the change of $\kappa(t)$ from $\kappa(0)$ independent of $t$.   Consequently, even if the inductive step $t$ may depend on $m$, as it will not appear in the bound, our results hold.
> > >
> > > Please see more details in the revised proof.
> > >
> > > *W2: I don't think this solves the problem: $(1+\delta - 1/m)^2 <(1+\delta)^2$ seems to be a counter example.*
> > >
> > > We have revised the proof by using explicit constants, which solves this problem.

---

> > > > ### Author Response · Authors · 2023-11-20
> > > >
> > > > We thank the reviewer for the reply. Your concerns will be addressed below.
> > > >
> > > > *Comment 1: A minor concern is that I notice a change in the statement of Lemma 1, from a term $O(\frac{\delta^2}{\log m})$
> > > >  to  $O(\frac{\delta}{\log m})$. Could the authors explain more this change?*
> > > >
> > > >  We fixed a typo in the original draft. The typo was in the bound for the change of $v$, i.e., $|v(T'+1) -v(0)|$, where the summation of $u(t)$ has a dependence on $1/\delta$. Note that the dependence from $O(\frac{\delta^2}{\log m})$
> > > >  to  $O(\frac{\delta}{\log m})$ does not affect the result of $\kappa(t)>1$ as $\delta >\epsilon = \Theta(\log m/\sqrt{m})$.
> > > >
> > > > *Comment 2: In equation (13) there is written a definition $\kappa(t):=(1-v(t)+u(t)+w(t))^2$. However in the proof of lemma 1 it is written: $\kappa(0) = (1+\delta-u(0)-w(0))^2$. I would like to know the origin of the term $\delta$ and why there is not: $\kappa(0) = (1-\delta + u(0)+w(0))^2$. Similarly I do not understand at the end of the proof of lemma 1 why there is written $\kappa(T'+1) = (1-v(T'+1)+u(T'+1)+w(T'+1))^2 \geq (1+\delta - ...)^2$. Could the authors explicit which term becomes which term in the inequality?*
> > > >
> > > > We define $\delta := \eta\lambda_0 - 2$ (the definition can be found before the statement of Lemma 1 and we also update the draft to add the definition in the proof). Note that $\delta > 0$.  Since $v(0) = \eta\lambda_0 = \delta+2$, we can get,
> > > > \begin{align*}
> > > >     \kappa(0)  &= ({1-v(0)+u(0) + w(0)})^2\\
> > > >     &= ({ 1 - (\delta+2) +u(0) + w(0)})^2\\
> > > >     &= ({1+\delta-u(0) - w(0)})^2.
> > > > \end{align*}
> > > > For the bound on $\kappa(T'+1)$,  we can see
> > > > \begin{align*}
> > > >     \kappa(T'+1) &= (1-v(T'+1)+u(T'+1)+w(T'+1))^2\\
> > > >     &= (1 - v(0) + (v(0) - v(T'+1)) + u(T'+1) + w(T'+1))^2\\
> > > >     &= (1+\delta + v(T'+1) - v(0) - u(T'+1) - w(T'+1))^2.
> > > > \end{align*}
> > > >
> > > > Since we have the control over $|v(T'+1) - v(0)|$ and $|u(T'+1)|$ and $|w(T'+1)|$ where all of these terms will be much smaller than $\delta$ with sufficiently large $m$, we can lower bound $\kappa(T'+1)$:
> > > > \begin{align*}
> > > >     \kappa(T'+1) &=   (1+\delta + v(T'+1) - v(0) - u(T'+1) - w(T'+1))^2\\
> > > >     &\geq (1+\delta - |v(T'+1) - v(0)| - u(T'+1) - |w(T'+1)|)^2.
> > > > \end{align*}
> > > >
> > > > More specifically,  we get $|v(T'+1) - v(0)| \leq \frac{14C'_u\delta}{\log m}$ by Eq.(14). And we know $u(T'+1)\leq u(T) \leq C'_u\delta^2/\log m$ by the assumption. For $w(T'+1)$, as $|w(T'+1)| = C\sqrt{u(T'+1)}/\sqrt{m}$, we can bound it as well. The term $|v(T'+1) - v(0)| + u(T'+1) + |w(T'+1)|$ can be upper bounded by $\delta/4$ with sufficiently large $m$.
> > > >
> > > > We have revised the proof to clarify these steps. We also updated the constants for better clarification.  Please see more details in the updated draft.

---

> ### Comment · Reviewer_Unww · 2023-11-21
> **Response to the authors**
>
> I thank the authors for their response which I have the feeling make me now understand better theorem 4 and lemma 1.
>
> I thank them for having also changed the proof of theorem 4 with explicit constants which makes it more understandable in my point of view. However I have still concerns on the proof of proposition 3 still partly because of the the $O,o,\Theta,\Omega$ notations.
>
> Minor concerns:
> 1) there is a different definition of $\delta$ before the "increasing phase" in the main paper as in the proof of lemma 2 in the appendix. Could you clarify?
>
> 2) I get the intuition but do not understand the meaning of proposition 1,2,3 because I feel there lacks something of the form "for m large enough...". For example in proposition 3, you state "v(t) keeps increasing...". I feel it would be good to specify it. (or correct me if I am wrong)
>
> Bigger concerns:
> 3) I think that T_2 and T_1 depend both on m. I think it is important to recall it, or else I would be happy if the authors could explain me why it does not depend on m. Following this remark, I do not understand the sentence "Therefore, for $t\geq T_2$ such that u(t) is of the order greater than $O(1/m)$. Indeed, you cannot work with fixed $t$ if you set the condition $t\geq T_2$ with $t$ depending on $m$. Perhaps it would be good to specify this point.
>
> 4) I do not understand the proof of Proposition 2 in part H.2, starting "decreasing. Specifically...". Could your reformulate this part a bit? Especially: first what is meant by "order at least O(.)". Do you mean "order at least $\Omega$" or "order at most $O(.)$"? Second, are you sure about $O(1/\sqrt{log(m})$? it seems strange to me since this order is actually bigger than $1/\log(m)$ that you were discussing before. Shouldn't it be $O(1/log(m)^2)$
>
> 5) I will probably seem very attached to details but the proof by contradiction of proposition 2 is troubling me: you prove that the following statement is wrong: "$\forall t \geq T_1, v(t)=4-o(1)$". My concern is in terms of the mathematical statement of the contradiction of this. To me, the correct statement would be: "$\exists t \geq T_1, 4-v(t)$ is not $o(1)$" However, it seems not possible here since $T_1$ depends on $m$ and therefore $t$ cannot be defined.
> Furthermore I would like if the authors could clarify the following point: it seems to me that you use that the contradiction of being $o(1)$ is being $\Omega(1)$ but I think that it it is only true for a subsequence.
>
> 6) In the proof of proposition 2 it is written $u(T_1)=\Omega(\frac{1}{\log(m)})$. Could the authors explain more why it is true as it seems in contrast with the statement above proposition 2: $T_1=inf (t, u(t)=\Theta(\frac{\delta^2}{\log(m)}))$ with $\delta$ small.

---

> ### Author Response · Authors · 2023-11-22
>
> We thank the reviewer for the reply. We have revised the statements of propositions and proofs to get rid of $O,\Omega,o$ by using explicit constants. And we will address your specific concerns below.
>
> *Concern 1: there is a different definition of $\delta$
>  before the "increasing phase" in the main paper as in the proof of lemma 2 in the appendix. Could you clarify?*
>
>  This is a typo. We have fixed it.
>
> *Concern 2: I get the intuition but do not understand the meaning of proposition 1,2,3 because I feel there lacks something of the form "for m large enough...". For example in proposition 3, you state "v(t) keeps increasing...". I feel it would be good to specify it. (or correct me if I am wrong)*
>
> We have revised the statements of propositions and used explicit constants in the proof. Now there is an explicit lower bound for $m$ for the propositions to hold.
>
> *Concern 3: I think that $T_2$ and $T_1$ depend both on m. I think it is important to recall it, or else I would be happy if the authors could explain me why it does not depend on m. Following this remark, I do not understand the sentence "Therefore, for  $t\geq T_2$ such that $u(t)$ is of the order greater than  $O(1/m)$. Indeed, you cannot work with fixed $t$ if you set the condition $t \geq T_2$ with $t$ depending on $m$. Perhaps it would be good to specify this point.*
>
> We have clarified it in the proof that $T_1,T_2$ depend on $m$.
>
> We have revised the statements, and now the proposition becomes under certain conditions, there exits $T_2$ such that $v(T_2) <3$ (see Proposition 3 in the updated draft).
>
> *Concern 4: I do not understand the proof of Proposition 2 in part H.2, starting "decreasing. Specifically...". Could your reformulate this part a bit? Especially: first what is meant by "order at least $O()$". Do you mean "order at least $\Omega$
> " or "order at most $O$
> "? Second, are you sure about $O(1/\sqrt{\log m})$
> ? it seems strange to me since this order is actually bigger than $1/\log m$
>  that you were discussing before. Shouldn't it be $1/\log^2 m$?*
>
>  We have revised the proof by using explicit constants. Specifically, we added Proposition 2 to clarify when $u(t)$ is larger than $\frac{4C}{m (4-v(t))^2}$,  $v(t)$ will decrease.
>
> *Concern  5: I will probably seem very attached to details but the proof by contradiction of proposition 2 is troubling me: you prove that the following statement is wrong: "$\forall t\geq T_1, v(t) = 4-o(1)$
> ". My concern is in terms of the mathematical statement of the contradiction of this. To me, the correct statement would be: " $\exists t\geq T_1$,$ 4-v(t)$ is not $o(1)$
> " However, it seems not possible here since $T_1$
>  depends on $m$
>  and therefore $t$
>  cannot be defined. Furthermore, I would like if the authors could clarify the following point: it seems to me that you use that the contradiction of being
> $o(1)$ is being $\Omega(1)$
>  but I think that it it is only true for a subsequence.*
>
> We have revised the Proposition and used explicit constants in the proof. Now we show that there exists $T_2 \geq T_1$ ($T_2$ depends on $m$) such that $v(T_2)<3$. Note that this statement is more explicit than " $\exists t\geq T_1$,$ 4-v(t)$ is not $o(1)$".  The idea of the proof is similar: $v(t)$ will keep decreasing if $v(t)\geq 3$ therefore $v(t)$ will be less than $3$ at a certain iteration.
>
> *Concern 6: In the proof of proposition 2 it is written  $u(T_1) = \Omega(1/\log m)$
> . Could the authors explain more why it is true as it seems in contrast with the statement above proposition 2:
>  $T_1 = \inf_t\{u(t) = \Theta(\delta^2\log m)\}$
>  with $\delta$
>  small.*
>
>
> The proposition shows that there exits $T_2$ such that $v(T_2)$ is not so close to $4$. For the case where $\delta$ is small, i.e., $v(T_1)$ close to $2$, we can just let $T_2 = T_1$. For the case $v(T_1)$ is close to $4$, we have $\delta >1$. We have clarified it in the updated draft.

---

> > ### Comment · Reviewer_Unww · 2023-11-22
> >
> > I thank the authors for their answer and the improvement in the proofs that was done, by using explicit constants and getting rid of the (in my opinion) a bit "approximate" notations $O,o$... I unfortunately couldn't get through all the proofs by lack of time especially  of the result with multiple training points.
> >
> > I increased a bit my score because of the improvement in the proofs that was done and because I think that the results shown are interesting and hard, but decrease my confidence as I am unable to judge of the fully correctness of the remainding proofs.

---

### Meta-Review · Area_Chair_wQnH · 2023-12-05

**Metareview:**

This paper studies the catapult phenomenon for neural quadratic models. More precisely, the author show that the quadratic expansion of a Neural Network exhibit the ‘catapult phase’ which consist of an increase of the loss followed by a convergence of the loss when the learning rate is around 2/L (L being the largest eigenvalue of the Hessian at initialization).

The results are timely and interesting as they are able to model the optimization dynamics of high dimensional neural networks.

However, the proof are very complex and no reviewer (even after a long discussion) seem to be confident about the work.

**Justification For Why Not Higher Score:**

The correctness of the results is too uncertain.

**Justification For Why Not Lower Score:**

The results are novel and significant.

The authors engaged in the discussion and updated their proofs to clarify them.

---

### Decision · Program_Chairs · 2024-01-16

Accept (poster)